# Hand-like autonomous flying robot for airborne grasping and interaction

Yuze Wu [1,2,3,4], Fan Yang[2,4], Rui Jin[1,2], Yuhang Zhong[1,2], Junjie Wang[1,2], Xuankang Wu[2] & Fei Gao [1,2,3] ✉

Birds' extraordinary aerial agility and environmental interaction enable complex tasks such as mid-air hunting, perching, and nest-building, inspiring the development of advanced aerial robots with similar manipulation capabilities. However, existing platforms often face challenges such as large size, heavy payloads, end-effector torque interference, and limited functionality, severely restricting their practical deployment. Drawing inspiration from the biological, structural, and actuation characteristics of human hands, we propose a hand-like robot that integrates flight and grasping, demonstrating the synergistic advantages of compact structure, agile flight, and versatile manipulation. We propose an autonomous framework including efficient mission planning and multi-level adaptive control, enabling the robot to precisely and smoothly perform human-like grasping, opening doors, forest perching, object transport, and interactive tasks. Additionally, the framework supports human-robot collaboration, empowering individuals with mobility impairment to conduct remote transportation and airborne operations. Outdoor tests, which include perching in various scenarios, navigating confined spaces, and transporting payloads across challenging terrain, validate the proposed vehicle's potential in aerial delivery and manipulation tasks. These results demonstrate emerging possibilities for aerial operation, assistance, and delivery with integrated flight and manipulation abilities.

Through evolutionary processes, birds have developed a notable ability to use their forelimbs (wings) for aerial locomotion while employing their hindlimbs (talons) for interaction, allowing complex activities such as aerial hunting, grasping, perching, and nest building[1,2]. This biological characteristic has ignited numerous research interests to employ flying machines with manipulation capabilities[3–15], in order to interact with objects and humans in midair. Flying robots, as the most maneuverable robots, are highly anticipated to deeply participate in our social activities, especially safety-critical scenarios like earthquake rescue and high-risk inspections. In the past decade, flying robots are widely used in applications related to information acquisition, such as geographic surveying, aerial photography/ videography, inspection and monitoring. For instance, in hazardous

environments such as nuclear power plants or chemical facilities, close-range interactions including valve turning and button pushing are quite common. In search-and-rescue missions, quick catch and release are vital for supply delivery or collaborative transportation. In daily life, item distribution across the air, goods retrieval from human-unreachable areas, or even touch-range extending for the disabled, often occur in our imagined future house or factory. These cross-domain applications highlight the vast potential of aerial manipulation, motivating aerial robots from flying eyes to flying hands.

Existing research on aerial manipulation has made significant progress, while some fundamental limits restrict their further applicability and extensibility in real-world scenarios. Early research in this area primarily focuses on directly mounting robotic arms to

[1]Institute of Cyber-Systems and Control, College of Control Science and Engineering, Zhejiang University, Hangzhou, China. [2]Huzhou Institute, Zhejiang University, Huzhou, China. [3]Differential Robotics, Hangzhou, China. [4]These authors contributed equally: Yuze Wu, Fan Yang. ✉e-mail: fgaoaa@zju.edu.cn

drones[16–22], but their large size, heavy weight, and high energy consumption severely hurt their maneuverability and endurance, making them unsuitable for delicate or long-duration operations, especially in confined situations such as human-involved activities. Subsequent efforts seek to address these issues by optimizing end-effector designs, including simplifying actuators[23–36], developing novel drive mechanisms[37–40], and introducing soft grasping components[41–43]. Although these works shine in structural innovations, they introduce unavoidable control coupling problems, resulting in compromises in agility and stability of the robot. In response, researchers opt to leverage robots' own structures for aerial manipulation, works[44–51] try to theoretically reduce system complexity by minimizing external attachments. However, these robots struggle with either low accuracy caused by extra actuators, or mechanical complexity because of movable structures, limiting their operating range, precision, and speed. These challenges all underscore the necessity of a novel flying manipulation robot, which simultaneously satisfies compact design, simplified mechanism, wide adaptability, superior agility, stability and passibility, as well as high autonomy.

As the call for the appearance of such an ideal flying operational robot, we dive into the fundamentals of nature. Human hands exhibit dexterous interaction movements (Fig. 1a), efficiently adapting to complex environments and performing a wide range of tasks. For instance, humans grasp large objects like cups or doorknobs using the palm (Fig. 1c), and delicately pinch smaller objects such as paper or pills with fingertips (Fig. 1d). Research[52,53] shows that the bones, joints, muscles, and tendons of hands constitute a highly efficient biological structure, precisely adapting to the shape and size of objects through multi-degree-of-freedom (DOF) movements and tendon drive mechanisms. This notable architecture inspires our integrated design, which combines the dexterous grasping abilities of human hands with the swift maneuverability of aerial flight, leading to a **H**and-l**I**ke compact **A**erial **R**obot for **M**anipulation, abbreviated as **HI-ARM** in this article (Fig. 1b). The proposed flying robot achieves delicate, multifunctional, maneuverable, and continuous aerial manipulation with a size of solely an adult hand (see Supplementary Movie 1). HI-ARM's design incorporates hand-like features for functional grasping, including an open C-shaped grasping contour to extend its range, a multi-DOF deformable joint structure to accommodate objects of various shapes, and a concise tendon-driven mechanism to reduce its total size and weight. As illustrated in Fig. 1b, the C-shaped grasping contour provides a hand-like enveloping geometry, significantly enhancing grasping stability and adaptability for objects. The composite 5-DOF finger-like structure, including a 2-DOF torsion and a 3-DOF extension (Fig. 2a) parts, enables efficient manipulation. Additionally, HI-ARM equips four rotor-propellers for locomotion (Fig. 2a), inheriting the nature of flight agility and control simplicity from a conventional quadrotor aircraft. Thanks to the hand-like structure, HI-ARM enjoys enhanced adaptability for versatile tasks. It not only performs hand-like grasping such as palm gripping and fingertip pinching (Fig. 1e and 1f), but also executes sophisticated operations like tree perching, door opening, object transportation, and human interaction (Fig. 1g-1j).

To accomplish complicated tasks with precision and smoothness, autonomy is also crucial for HI-ARM. To this end, we propose a framework consisting of a task planner, trajectory generation, state feedback, parameter estimation, and adaptive control (Fig. 2b). HI-ARM supports two working modes: (1) autonomous operation, where the task planner selects proper operating sequences from an action library according to input task types and the robot executes them under closed-loop perception, planning, and control (e.g., grasping, perching, door opening, and human interaction), and (2) human-robot collaboration, where the task planner generates reference trajectories and grasping commands from human intentions, and the controller subsequently tracks these commands (e.g., human-robot collaborative

teleoperation). Thanks to its integrated hardware configuration, HI-ARM naturally divides its mission planning into flight trajectory planning (millisecond-level) and end-effector deformation planning (microsecond-level). This decoupled scheme significantly reduces the planning complexity, and can run at high frequency to meet the real-time demands for airborne reactive operation. State feedback is necessary for closed-loop autonomy, HI-ARM achieves this by using a 6-DoF state estimator to online update the pose and position of its center of gravity (COG), and a motor observer to monitor the angle and displacement of its end-effectors. In the real world, HI-ARM encounters notable parameter changes and non-negligible external disturbances, caused by close-proximity operation, load variation, and deformation dynamics (Fig. 2c). The above-mentioned model mismatch commonly occurs in HI-ARM, thereby drastically harming the accuracy of its flight and manipulation. To address these issues, the proposed controller integrates a lightweight, high-frequency disturbance estimator along with a torque and thrust compensator. The entire algorithm architecture is shown in Fig. 2b. With all these components integrated, the proposed HI-ARM successfully achieves autonomous planning, accurate control, task decoupling, and human interaction, therefore can be applied to multiple situations, as shown in Fig. 1g–j.

Contributions. Firstly, we propose a biomimetic design that integrates grasping with flight in a compact robot platform. To the best of our knowledge, this represents a pioneering flying robotic hand in the robotics community. Secondly, we propose an efficient planner for empowering the flying robotic hand with precise autonomous operations, along with an adaptive controller for stable flight with varying loads and dynamic interactions, and state feedback for closed-loop autonomy. Finally, we demonstrate the huge potential of applying HI-ARM to versatile tasks, including object grasping, door opening, pole perching, cross-terrain transportation, and continuous human-robot interactions, pushing the boundaries of aerial robots from passive observation to active manipulation. In what follows, HI-ARM successfully completes multiple continuous aerial interactive tasks quickly and smoothly, highlighting its great potential as an intelligent assistant (see Section Interaction with humans). Moreover, the robot demonstrates rapid object grasping and cross-terrain transportation, presenting an innovative solution for aerial delivery, in Section Applications in the wild. These experimental results not only validate HI-ARM's multi-task capabilities but also lay the foundation for its application in unmanned autonomous operations, robotic household service, wilderness rescue, remote assistance, and more.

## Results
### Hand-like mechanism design
The design of HI-ARM draws inspiration from the grasping configuration, biological structure, and tendon drive mechanism of human hands. The employed configuration adopts a hand-like open grasping contour, providing a relatively broad grabbing range. In order to replicate dexterous grabbing capabilities while remaining mechanically efficient, the robot incorporates finger and palm modules as its core operational units (Fig. 3b(i)). With this design, HI-ARM includes a palm region for powerful gripping and fingertip areas for precise pinching (Fig. 3a(ii)). This configuration supports a wide grasping range (0 ∼ 10.0 cm), allowing HI-ARM to securely hold larger objects (e.g., water bottles) with its palm while delicately picking up smaller items (e.g., tissues) using its fingertips, showcasing multi-modal grasping capabilities.

Human grasping relies on joint angle variations to achieve contraction (Fig. 3a(iii)). Following this functional principle, HI-ARM's finger modules incorporate a torsion structure comprising torsion springs and circular bearings (Fig. 3b(i)) to enable flexible finger bending. Given their ability to displace significantly in a short time, telescopic structures are incorporated into the deformable

**Hand-inspired structure**

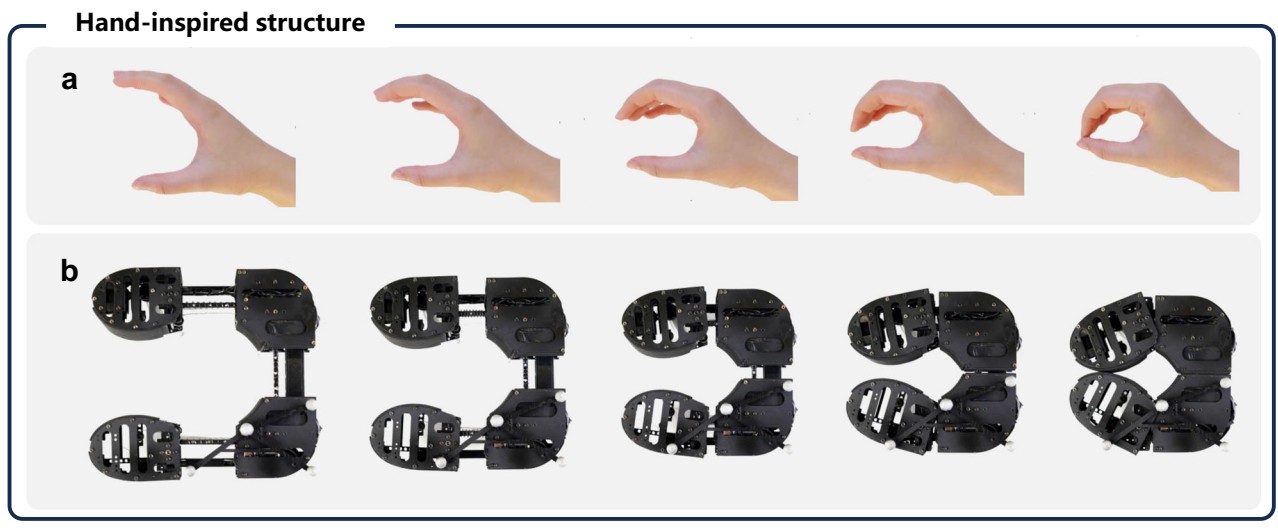

**Hand-inspired grasping**

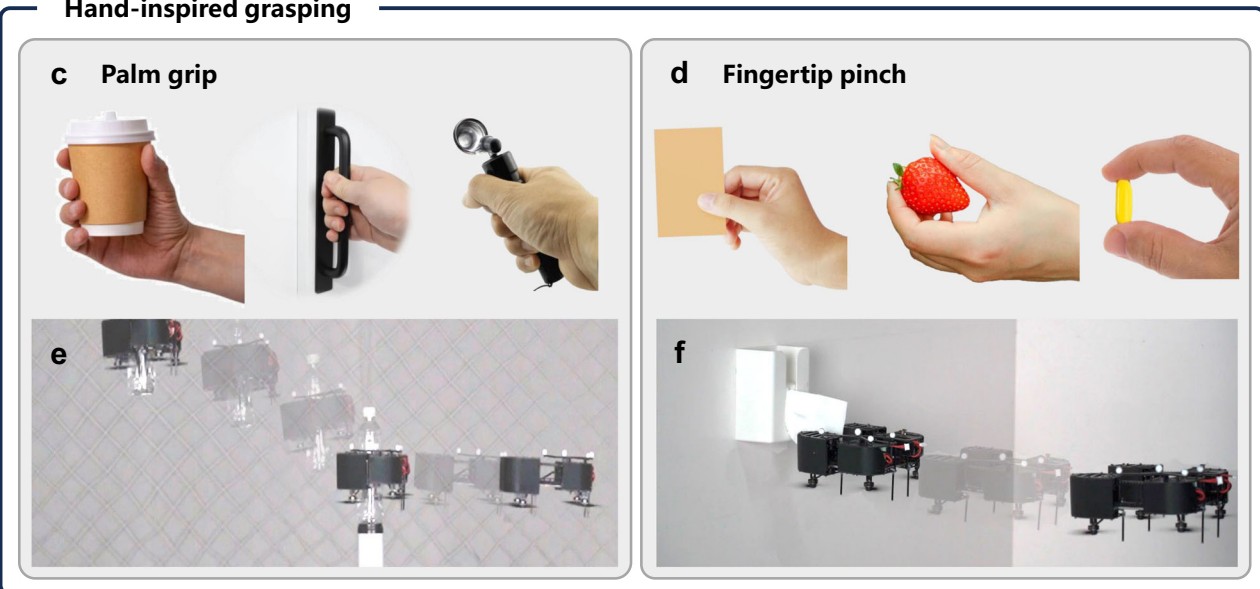

**Hand-inspired application**

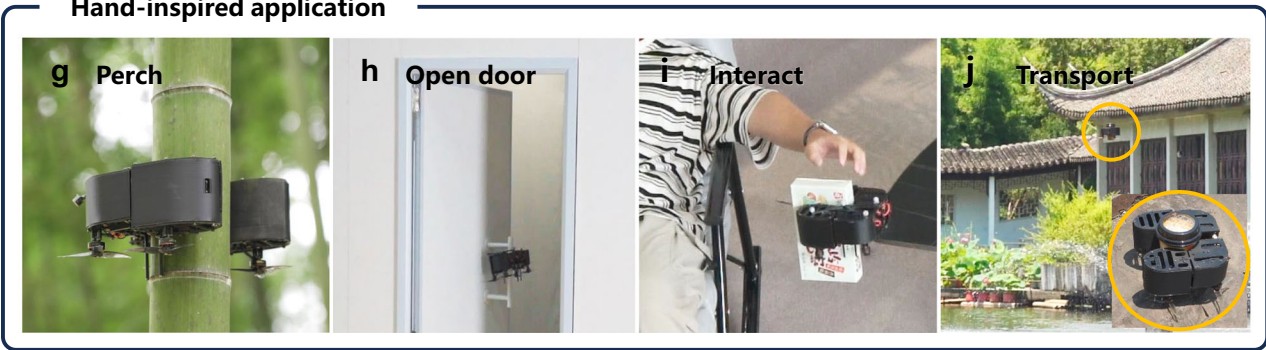

**Fig. 1 | Overview of the proposed HI-ARM. a** The sequence of hand grasping motion. **b** The process of HI-ARM morphing and grasping. **c** Palm gripping of human hands. **d** Fingertip pinching of human hands. **e** HI-ARM palm grips a bottle of water. **f** HI-ARM fingertip pinches a thin napkin. **g** HI-ARM perches on a bamboo. **h** HI-ARM opens a door. **i** HI-ARM interacts with a human. **j** HI-ARM transports an item across the river.

mechanism to increase the speed of grasping movements. To ensure that the telescopic modules are activated first, their spring stiffness is deliberately designed to be lower than that of the rotational modules, thereby enabling rapid contraction and closer contact with the target object. As illustrated in Fig. 3e, the springs in the mechanism absorb energy during compression and torsion, and then release their stored energy to restore the shape of the robot, thereby reducing overall energy consumption. The integration of telescopic and torsional mechanisms forms a hybrid 5-DoF structure, allowing a variety of adaptive and flexible grasping, as shown in Fig. 3f.

Building on the principles of the finger tendon sheath pulley system[53], HI-ARM employs a tendon drive mechanism for shape

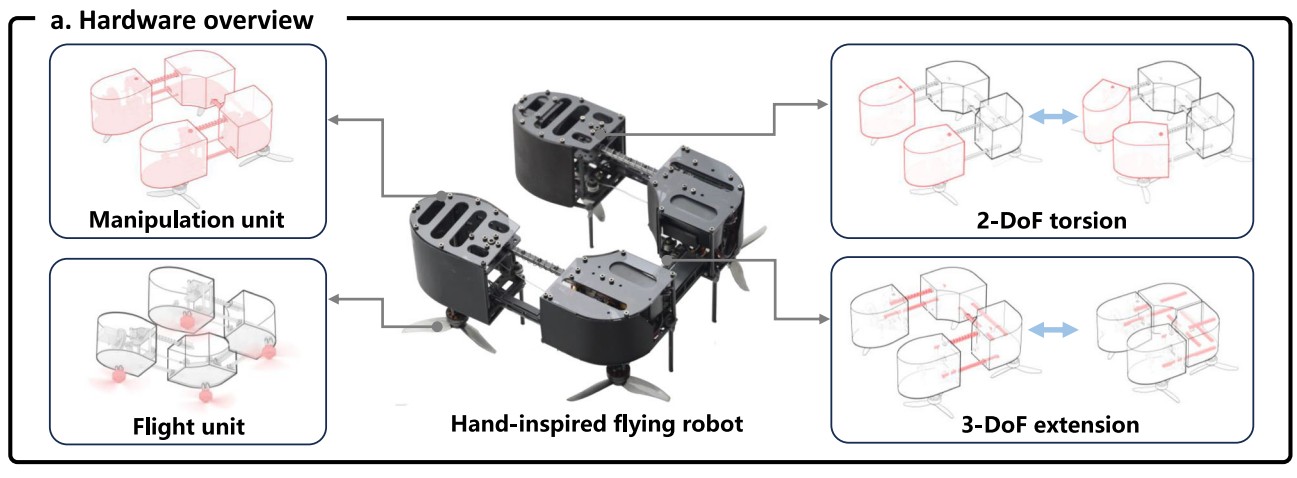

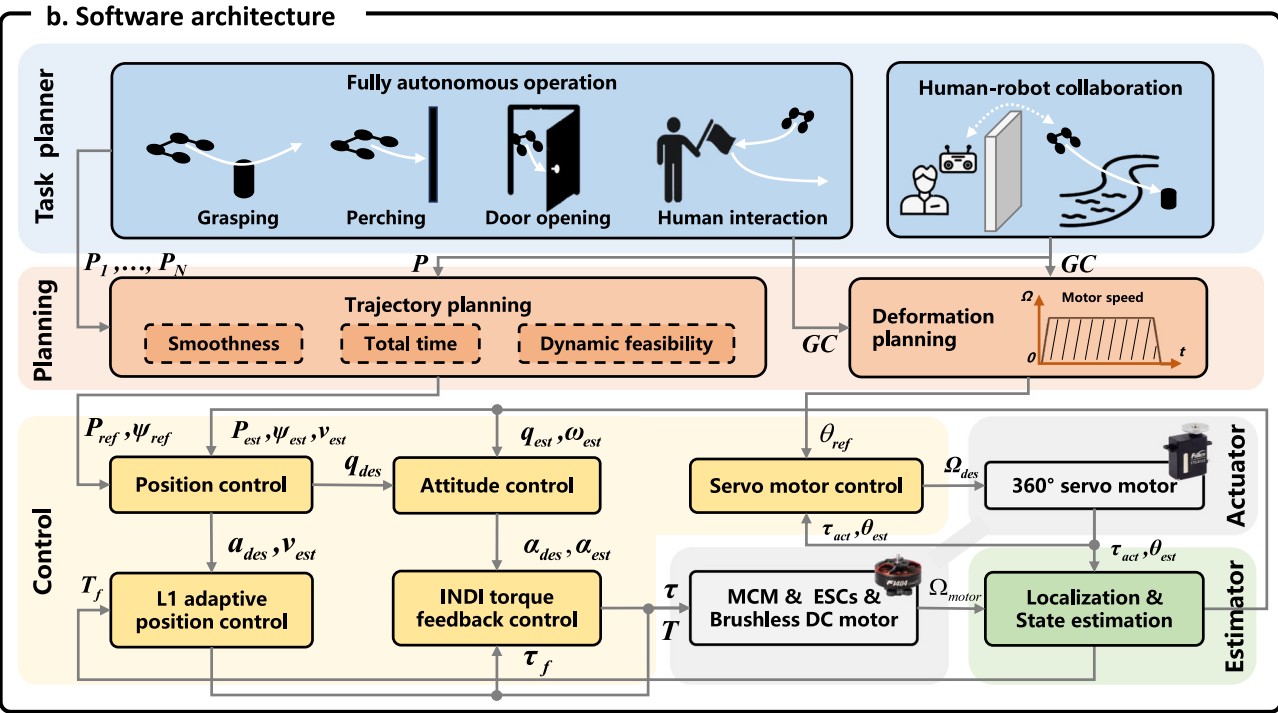

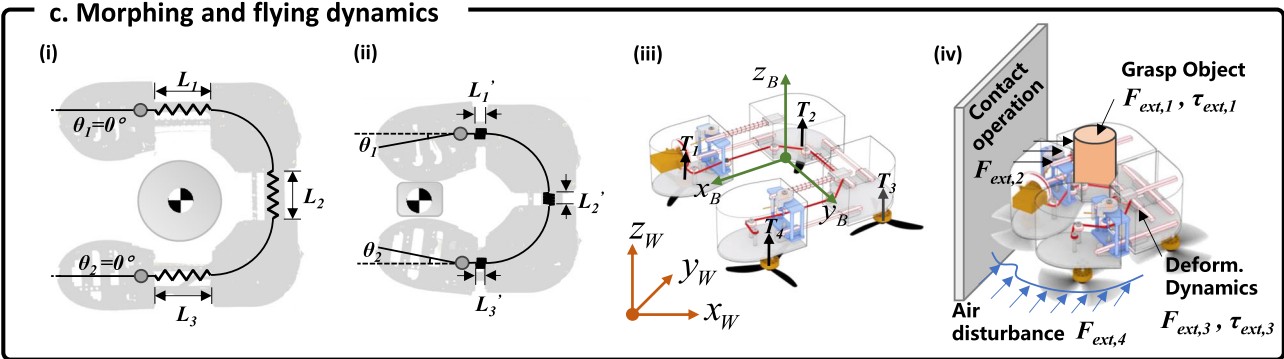

**Fig. 2 | Hardware and software architecture. a** Hardware overview. **b** Software architecture, including a task planner, motion planning, adaptive control, estimation, and actuation. **c** Morphing and flying models: (i) morphing state at Normal Size, (ii) morphing state during object grasping, (iii) coordinate system representation, (iv) External interference during aerial operation.

adaptation (Fig. 3e). Mimicking a finger's flexor digitorum profundus tendon (FDP tendon, shown in Fig. 3a(i)), a lightweight nylon rope is employed to transmit driving forces. Several V-shaped pulleys are integrated into the inner sides of finger and palm modules to redirect the force (Fig. 3b(i)). Unlike conventional morphing drones that rely on multiple actuators, this tendon-driven design utilizes a single actuator to drive the 5-DoF composite structure, minimizing the robot's size, weight, and energy consumption while simplifying its control complexity. Without knowing an object's shape, this underactuated structure can conform to the object's contour, and passively and

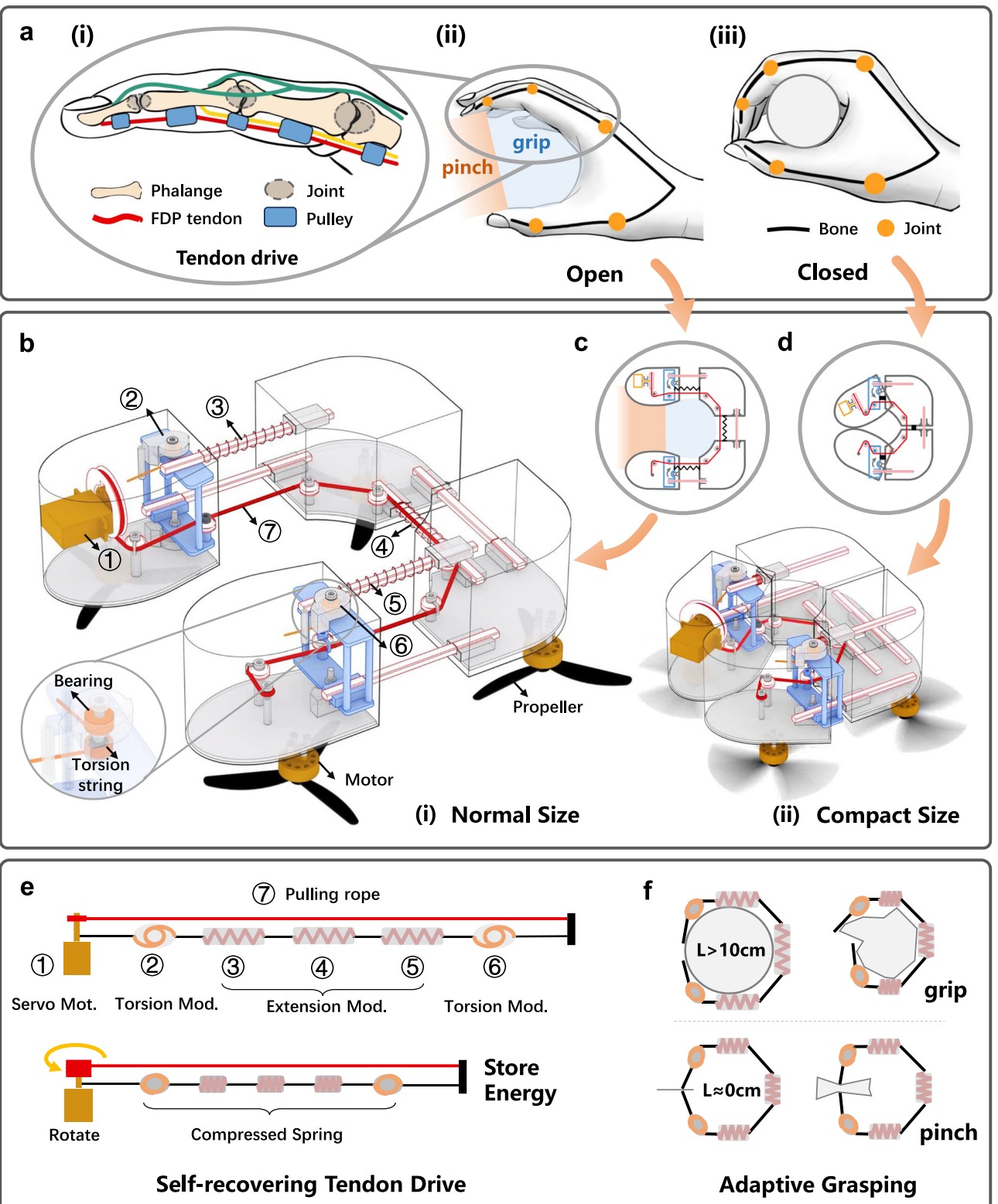

**Fig. 3 | Hand-like mechanical design. a** Biological structure of human hands: (i) tendon drive mechanism, (ii) open grasping configuration, (iii) multi-DoF joint structure. **b** HI-ARM's mechanical design and component details: (i) status of Normal Size, (ii) status of Compact Size. **c** Structure diagram in Normal Size. **d** Structure diagram in Compact Size. **e** Self-recovering tendon drive mechanism. **f** Adaptive deformation for grasping different objects.

collaboratively adjust the deformation of each torsion and extension component under single rope actuation to achieve stable grasping, demonstrating its adaptive grasping ability for various objects, as shown in Fig. 4c.

With the proposed integrated design, HI-ARM features a compact size with a hand-like profile, with a total weight of just 556g. This allows it to possess more space for flight and grasping operations in narrow indoor environments, a task that can be challenging for larger aerial

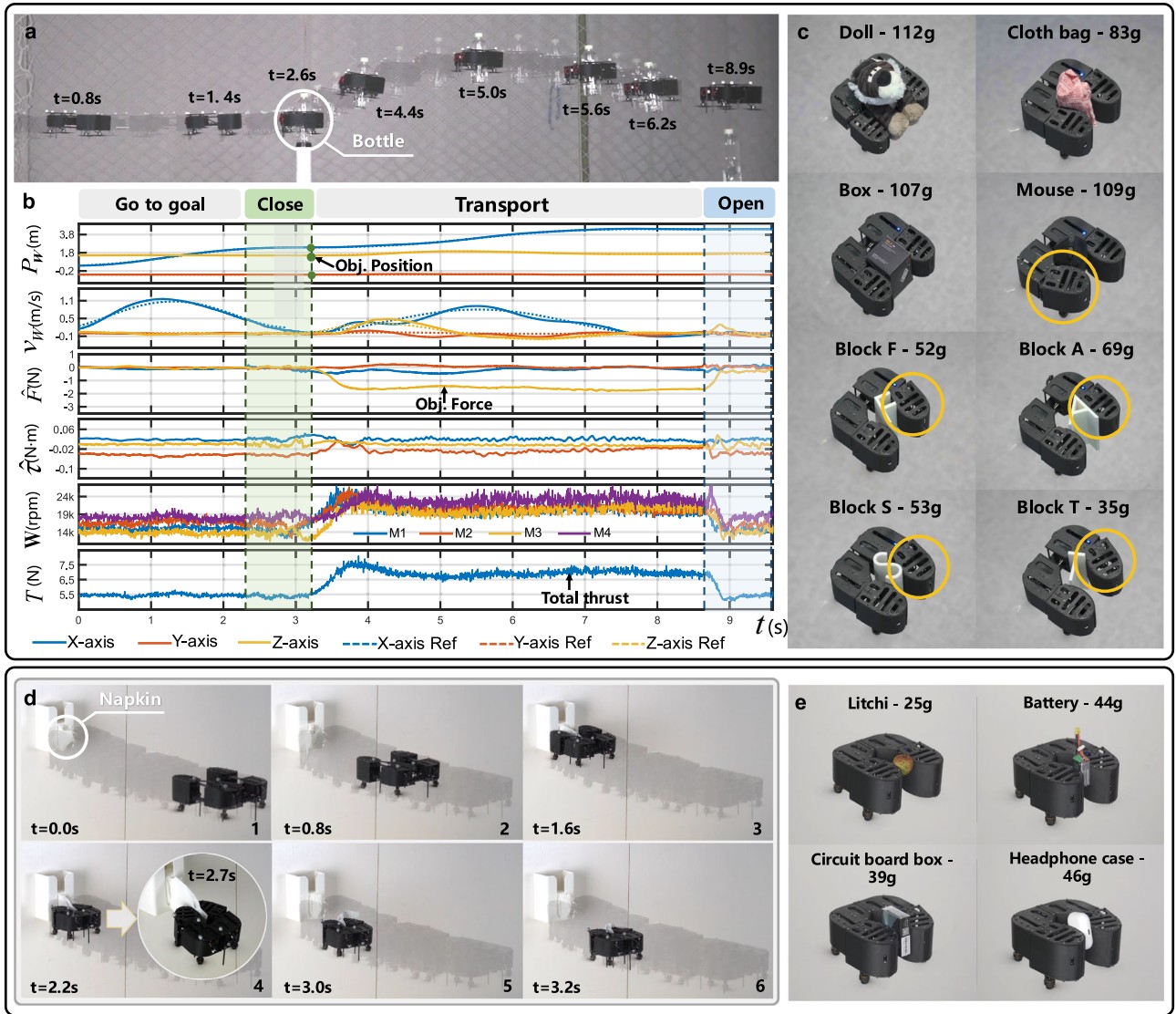

**Fig. 4 | Hand-like grasping. a** HI-ARM palm grasps, transports, and releases a water bottle (image mirrored horizontally for better readability). **b** State curves of HI-ARM during the water bottle grasping process, including position tracking, velocity tracking, estimated external forces, estimated external torques, motor speed, and actual total thrust. **c** Adaptive grasping of objects with various shapes by HI-ARM. **d** A sequence of HI-ARM performing fingertip grasping on a napkin. **e** Fingertip grasping of items with various shapes.

robots. Additionally, the robot can deform to reduce its dimensions (Fig. 3b(ii)), improving passability in narrow spaces (as validated in Section Applications in the wild). HI-ARM defaults to the open configuration, enabling rapid execution of grasping tasks. By contrast, the closed configuration imposes continuous torque demands on the servo motor and partially obstructs propeller airflow, elevating power consumption and compromising flight endurance.

### Flight design and electronic components

This section provides a detailed overview of the HI-ARM flight system, which is powered by four rotor-propellers. As shown in Fig. 2a, HI-ARM is equipped with 3.5-inch propellers, each driven by a T-motor F1404 2900KV motor, capable of generating a combined thrust of up to 1000 g. The motors are mounted beneath the finger and palm modules, and the bottom-mounted layout avoids direct propeller airflow and enhances human-robot interaction safety by minimizing hand contact. The propellers are designed with a 6 mm height offset to prevent interference during module deformation and folding.

With a 70C battery discharge rate, the system delivers sufficient instantaneous power output to handle payloads of over 450 g for

grasping operations. The primary structure of the flight platform is composed of 2 mm thick carbon fiber plates, selected for their high strength and lightweight properties. For flight control, HI-ARM uses a Kakute H7 mini flight control board to run the ArduPilot flight firmware, which collects high-frequency ( > 300Hz) real-time data on motor speed from the electronic speed controller (ESC) and posture from the Inertial Measurement Unit (IMU). High-frequency feedback helps the robot to estimate its motion states quickly, reducing system delay and improving control accuracy. The on-board computer, a Radxa ZERO board, runs a mission planner and adaptive controller (to be introduced in the following sections) for autonomous aerial manipulation. This controller processes system state data from the IMU, the servo motor, and the localization module, then sends control inputs to the brushless motor and the servo motor, which adjusts motor speeds for precise flight operations. More details about the components can be found in Supplementary Information.

For localization and estimation, we aim to balance accuracy, weight, and onboard computational load. In indoor experiments, an external optical motion-capture system provides sub-millimeter state estimation of both the aerial robot and the manipulated objects,

enabling closed-loop integration with mission planning and adaptive control for autonomous operation. In outdoor experiments, we employ the Intel RealSense T261 tracking module, which weighs 26 g and offers a lightweight alternative to LiDAR-based localization systems while maintaining stable short-range state estimation (for example, within 20 m) suitable for experimental validation.

## Mission planning

To perform diverse aerial tasks, HI-ARM introduces an efficient autonomous framework that supports both autonomous operation and human-robot collaboration. As shown in Fig. 2b, we establish an action library for common tasks such as grasping, perching, door opening, and human interaction. For autonomous operation, HI-ARM dynamically assembles basic motions from the library based on the task type to generate desired operation sequences (as validated in Section Interaction with humans). For human-robot collaboration, HI-ARM generates appropriate commands for flight and grasping based on the operator's intent. These commands are subsequently sent to the mission planner.

Unlike traditional aerial manipulation robots with multi-DOF robotic arms, HI-ARM's integrated structure reduces the number of actuators, significantly decreasing the number of planning variables. Thanks to this integrated design, we decompose the mission planner into two independent modules: flight trajectory planning and end-effector deformation planning. In terms of trajectory planning, the quadrotor configuration employed by HI-ARM exhibits differential flatness[54], allowing constraint simplification to accelerate solving speed[55]. To obtain an efficient, trackable trajectory for the robot, the optimization problem incorporates constraints on trajectory smoothness, total time, and dynamic feasibility. This process can be completed at a millisecond level on onboard computing devices, meeting the rapid real-time solving requirements for aerial operations. The trajectory planner eventually outputs the desired state sequence related to time, as detailed information provided in Methods.

For end-effector deformation planning, HI-ARM utilizes a single-motor actuation strategy and a tendon drive mechanism to achieve structural deformation. We establish a univariate quadratic mapping model between motor angles and morphing states, with additional details available in Methods. Given the desired deformation state, HI-ARM rapidly generates a time-dependent deformation sequence in microseconds based on the above mapping model. Then we synchronize this sequence with the time series of the flight trajectory, and send them to the adaptive controller, as shown in Fig. 2b.

## Multi-level adaptive control

An accurate, adaptive, and efficient controller is essential for smooth aerial manipulation. Compared to mature controllers for traditional quadrotors, the flight control system of HI-ARM faces greater challenges. These challenges mainly arise from two aspects (Fig. 2c(iv)): first, the model parameters (such as size, shape, COG, and inertia) dynamically change due to body deformation, which significantly disturbs flight control; second, during airborne operations, external factors such as load variations, close-range interaction forces, and air disturbances severely affect the stability and accuracy of the robot's control. In particular, to increase the speed of aerial operations, HI-ARM needs to deform while flying to complete tasks, imposing higher requirements on both flight control and deformation control.

To address these challenges, we propose a multi-level adaptive controller (Fig. 2b), endowing HI-ARM with precise aerial operation capabilities. To mitigate the negative effects of model variations, the robot incorporates an online model parameter identification approach to estimate key physical parameters, such as COG and inertia, as illustrated in Fig. 2b and detailed further in Supplementary Information. For external disturbances, the system introduces an estimation and

compensation method for external forces and torque disturbances to reduce their impact on control. Specifically, the $\mathcal{L}_1$ adaptive control algorithm[56] is employed in the position loop to mitigate external forces, while the incremental nonlinear dynamic inversion (INDI) control algorithm[57] is used in the attitude loop to handle external torques. These adaptive algorithms also reduce the negative impact of factors that affect trajectory tracking performance, such as propeller airflow interference and model mismatch[58].

In terms of deformation control, we implement a feedback control method based on angular errors to regulate the servo motor's motion. The servo motor provides torque state feedback, which helps determine the success of the gripping action. Through merging the above-mentioned grasping and flight control, HI-ARM can quickly and efficiently perform operations such as grasping, perching, and interaction, which are verified in the following experiments. More details of the control and modeling can be found in Methods.

## Hand-like grasping performance

The proposed biomimetic design endows HI-ARM with versatile gripping mechanisms that facilitate palm, fingertip, and adaptive grasping (see Supplementary Movie 2). These capabilities are demonstrated in experiments, as described below.

**Palm grasping.** Humans typically grasp larger objects (e.g., water bottles or oranges) using their palms, HI-ARM exhibits a similar ability. We conduct an autonomous grasping experiment, where HI-ARM is asked to grip a water bottle (diameter: 62 mm, mass: 153 g) before transporting it to a destination (Fig. 4a). The position tracking curves shown in Fig. 4b indicate that HI-ARM can successfully track a reference trajectory with relative accuracy at a maximum velocity of 1.1 m s$^{-1}$. Our proposed controller can accurately estimate external force and torque disturbances in real time (Fig. 4b) and effectively compensate for them. As shown in Fig. 4b, after correcting for thrust loss, the robot's actual total thrust increases by about 1.5 N upon grasping, closely matching the object's gravitational force. During gripping, once the set torque has been reached, the grasping action is considered to be completed, and the servo motor stops rotating.

**Fingertip grasping.** Other than palm grasping, fingertip grasping is also a normal operation for human hands, especially for taking small objects. In this experiment, HI-ARM is commanded to grasp some napkins. As shown in Fig. 4d, napkins are quite thin (less than 1 mm), soft, and light (less than 1 g), thus pose unique challenges for accurate automated grasping. Fig. 4d demonstrates that HI-ARM grasps a single napkin with fingertip precision. During tissue extraction, HI-ARM encounters disturbances caused by friction, which are effectively compensated by the controller, resulting in a control error of less than 3 cm.

**Adaptive grasping.** HI-ARM's 5-DoF grasping structure can deform flexibly, and its under-actuated design allows it to conform to the shape of objects upon contact. This feature enables the system to adaptively grasp objects without prior knowledge of their precise contours and shapes. We test HI-ARM on common objects with various shapes and sizes, proving that it can conform to the surface of objects ranging from small to big, and grip and carry them effectively (Figs. 4c and e). To increase the difficulty of grasping, we also introduce irregular letter-shaped blocks. As shown in Fig. 4c, HI-ARM is capable of deforming asymmetrically to fit the shape of these objects, holding them accurately and stably.

## Human-like applications

HI-ARM, with the above biomimetic features, can perform more human-like applications (see Supplementary Movie 3).

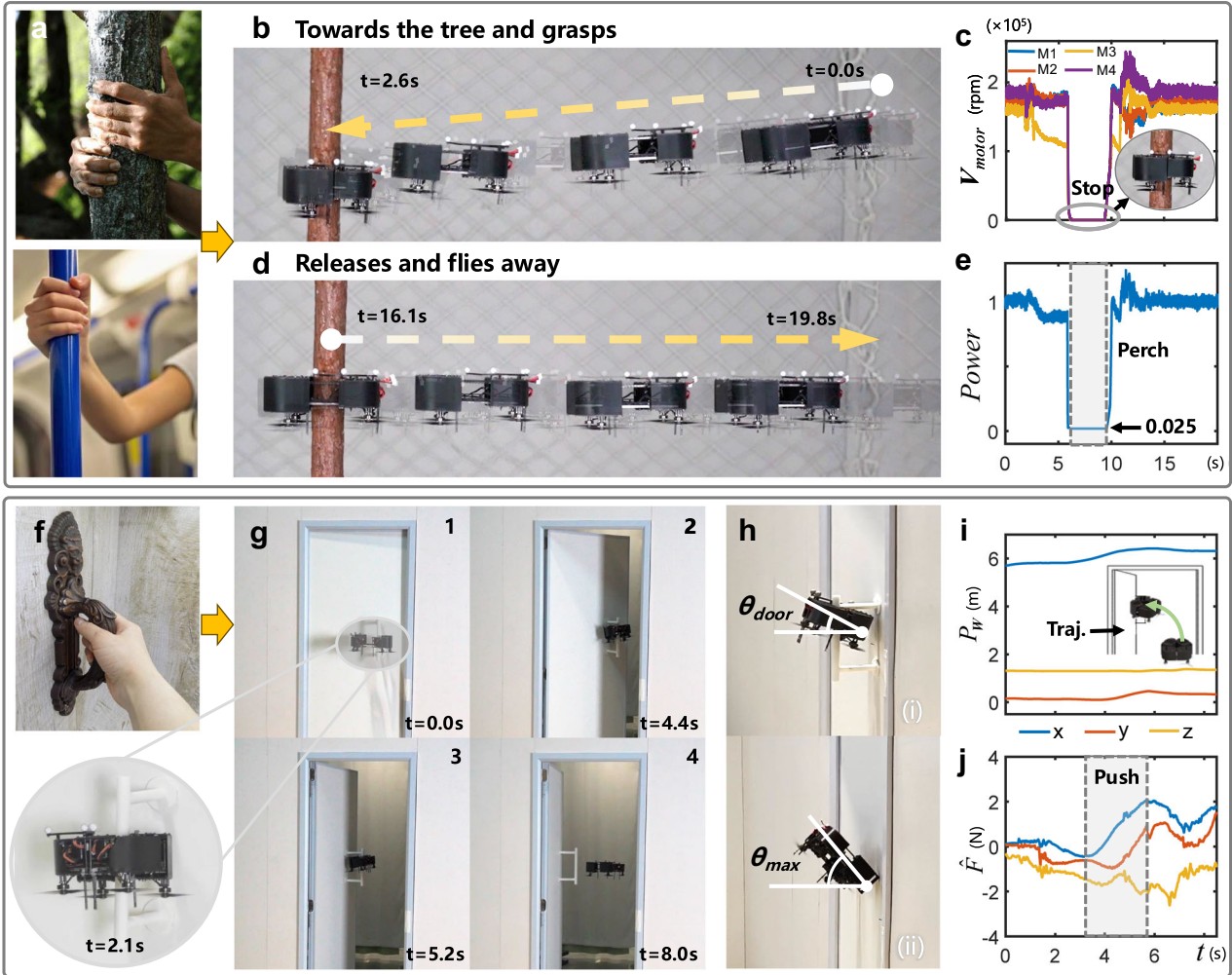

**Fig. 5 | Performance of perching and door opening. a** Humans grasp a handrail on trains and grip a tree trunk. **b** HI-ARM is perching on a tree trunk. **c** Motor speed curve during the perching maneuver. **d** HI-ARM is releasing from the tree trunk and flying away. **e** Normalized power curve. **f** Humans grab door handles to open doors.

**g** Sequence of HI-ARM opening a door. **h** Max force test for opening the door with/without the door handle. **i** Position curve during the door opening process. **j** Estimated external forces during the door opening action.

**Perching.** Similar to the way humans use handrails on trains or grasp tree trunks (Fig. 5a), HI-ARM can utilize its specialized structure to perch on fixed objects. To evaluate the perching capability, we set up a tree trunk as the test object. In this experiment, the flying robot autonomously flies toward the tree and accurately grasps and perches on it (Fig. 5b). As shown in Fig. 5c, all motors cease rotation, the system's gravity is counterbalanced by the friction generated from the HI-ARM's grip on the tree trunk, and the energy consumption of the system is significantly reduced (the servo motor consumes two orders of magnitude less energy than the rotor-propellers, as shown in Fig. 5e). By contrast, conventional hovering consumes over 160 W. After a while, the drone receives a release command and the propellers begin to rotate again. The servo motor drives the vehicle to subsequently expand, gradually disengaging from the tree, as HI-ARM re-enters flight mode (Fig. 5d). This innovative perch-and-relaunch mechanism allows for a break between consecutive missions and supports a long-stay mission with low energy consumption.

**Door opening.** Thanks to its superior operational capability, HI-ARM can even be used to open a door. In this experiment, when the door is being pushed open, it introduces significant disturbances to the control of the drone. To ensure smooth door-opening (Fig. 5h), we employ the mission planner to generate an appropriate reference trajectory

that accounts for dynamic constraints (more details can be found in Methods), enabling the drone to accurately grasp the door handle and open the door, as shown in Fig. 5g. External force estimations suggest that the door mainly exerts disturbing forces on the robot along the X and Y axes (Fig. 5j), which are compensated by the proposed controller. As illustrated in Fig. 5h, when the robot pushes the door while grasping the handle, the door-opening angle $\theta_{door}$ is -30°, and the maximum pushing force reaches about $T_{max} \sin 30° \approx 5$ N. Without the constraint of the handle, the maximum achievable door-opening angle $\theta_{max}$ increases to 55°, with a corresponding maximum pushing force of 8.2N.

**Interaction with humans.** The compact size of HI-ARM enables it to adapt to spatially constrained environments in typical households, offering autonomous robotic services with great potential as an intelligent home assistant. We design a series of experiments that simulate daily scenarios to evaluate the possibility of HI-ARM in domestic environments (Fig. 6a). In this experiment, the positions and orientations of objects are acquired using a motion capture system, while the flying robot is required to sequentially perform multiple tasks involving human-robot interaction. Figure 6b illustrates the continuous and smooth flight reference trajectory for multi-task operations generated by the aforementioned mission planner. Upon

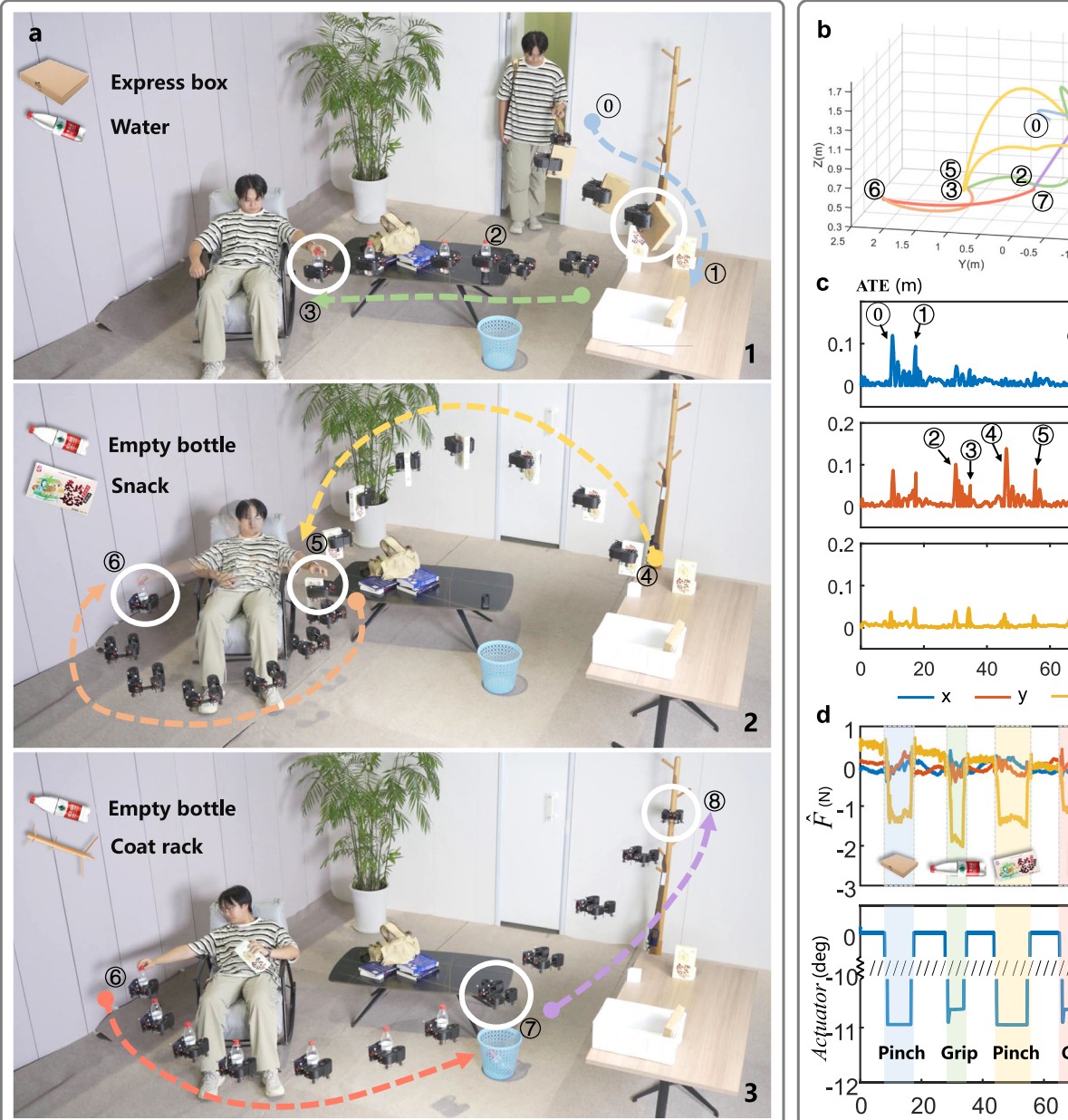

**Fig. 6 | Continuous multi-task interactions with human. a** HI-ARM successfully executes continuous multiple aerial interactive tasks including: (i) grasping and transporting an express box (75g) and a water bottle (134g), (ii) delivering a snack (60g) and grasping an empty bottle (23g), and (iii) transporting the bottle to a trash can and perching on a coat rack. **b** Reference trajectories for continuous tasks. **c** The absolute tracking error of each axis in continuous tasks. **d** External force estimation and servo motor angle for different objects.

the arrival of the person, HI-ARM approaches the door, performs a fingertip grasp on an express box held by him, and delivers it to a storage bin (Fig. 6a ⓪①). The drone then flies to a table, uses a palm grasp to pick up a bottle of water, and hands it to the person (Fig. 6a ②③). While the person is drinking, the flying robot retrieves a boxed snack from the table (Fig. 6a ④⑤). Once the person finishes drinking, the empty bottle is handed to HI-ARM, which then deposits it into a trash can (Fig. 6a ⑥⑦). After completing services, HI-ARM flies to the coat rack and perches on it, entering standby mode (Fig. 6a ⑧). As shown in Fig. 6d, HI-ARM can adaptively grasp different objects by adjusting the actuator's angle accordingly. During this experiment, HI-ARM encounters perturbation errors at each operation point (Fig. 6c), primarily caused by load variations introduced by the objects. Thanks to the multi-level adaptive controller, HI-ARM is able to accurately

estimate external force interference (Fig. 6d) and effectively compensate for the disturbances, gradually reducing errors. As shown in Fig. 6c, the mean absolute tracking error (ATE) of the single-axis trajectory during the experiment is less than 2 cm. This coherent process demonstrates HI-ARM's notable ability to adaptively grasp different objects and move flexibly in domestic scenarios.

## Applications in the wild

We also present several outdoor experiments to confirm that HI-ARM can be applied to natural environments. Firstly, we test the perching capability of HI-ARM on a variety of objects in outdoor environments, such as bamboo, different trees, and electric poles. As shown in Fig. 7a, HI-ARM successfully completes the grasping and staying in various scenes without external structures, showing great potential for wild

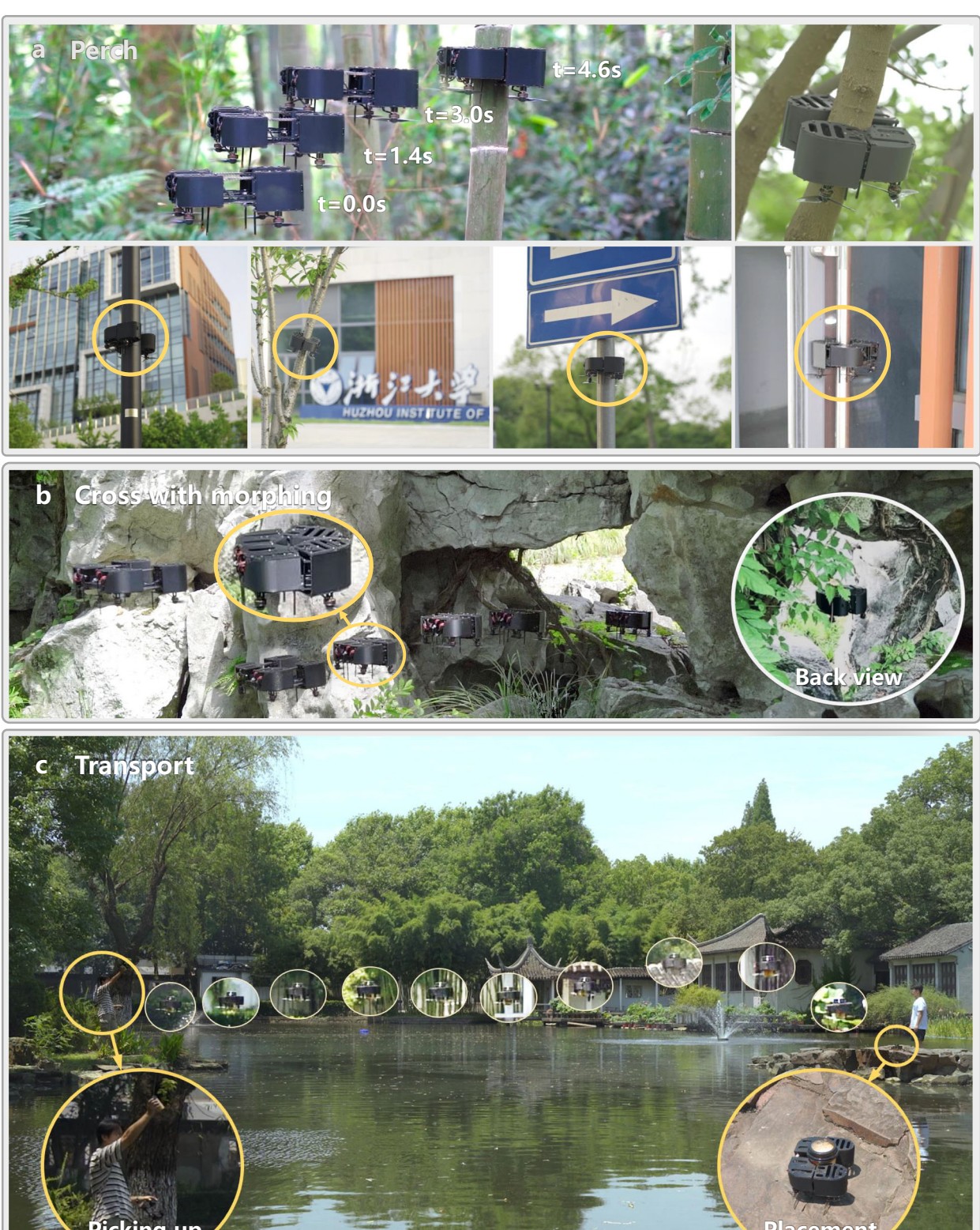

**Fig. 7 | Applications in the wild. a** Perching in various environments. **b** Crossing a narrow space by morphing. **c** Transporting an item across a river (the image of HI-ARM in the white box is a zoomed-in view centered on the original position for clarity).

applications where long-stay is necessary. In the second experiment, we test the passability of HI-ARM utilizing its shrinking ability in a quite narrow cave. As shown in Fig. 7b, HI-ARM deforms to reduce its width and successfully passes through a narrow space. Moreover, we demonstrate that the proposed flying robot may offer a

groundbreaking solution for aerial delivery, addressing the growing demand for flexible and rapid daily logistics. Due to its compact and simple structure and high mobility, HI-ARM can easily grasp various objects and perform swift aerial transportation. In this experiment, we give HI-ARM a bottle of water that needs to be delivered across a river.

As shown in Fig. 7c, HI-ARM successfully grips a drink cup using its deformable mechanism and carries it to the other side of the river.

### Teleoperation for human-robot collaboration

As a bio-inspired aerial interaction device, HI-ARM can function as a third flying hand for humans, responding to human intentions. As shown in Fig. 8a, HI-ARM is equipped with a remote video transmission system that provides first-person-view (FPV) visual feedback. Similar to DJI's Avatar FPV drone (¹ DJI's Avatar FPV drone specifications: https://www.dji.com/cn/avata-2), HI-ARM can be operated with a simplified, single-handed motion-based 3D controller, which maps hand movements to velocity commands in different directions. The controller also integrates a grasp button, allowing both flight and grasping actions to be performed with one hand. As shown in Fig. 8a, the communication between the controller and the robot is established via the ROS framework. As described in Section Mission planning, these commands are then processed by the mission planner to generate flight and deformation inputs for the controller, enabling the robot to execute aerial tasks. The following two experiments demonstrate HI-ARM's potential in human-assisted remote operations (see Supplementary Movie 4).

For individuals with limited mobility, retrieving items from different locations can be inaccessible, particularly in rugged terrains, areas with stairs/steps, or regions with significant height variations. In this experiment, we invite a participant with mobility impairment to operate HI-ARM, wearing video glasses to receive the onboard perspective. After a brief tutorial, the user is able to control HI-ARM to complete the object retrieval task from a distance according to his/her intention. As shown in Fig. 8b, HI-ARM takes off from the second floor, navigates through trees and shrubs, and reaches the target position near the ground. It precisely grasps the target object-a cup of coffee-and successfully returns to the participant's side, demonstrating HI-ARM's potential in assisting individuals with disabilities. In this task, the 46.2 m total retrieval trajectory is completed at an average velocity of 0.33 m/s, with an end-effector ATE of 0.08 m and a control latency of 256ms, highlighting the stability and effectiveness of HI-ARM's teleoperation over distances exceeding 40 m.

Even for individuals with normal mobility, there may be some situations where reaching objects at high places is difficult. This experiment simulates a badminton stuck in a tree, where the operator remotely assesses its position via onboard imagery. Subsequently, the person maneuvers HI-ARM to fly to the tree, grasp the badminton with its fingertips, and smoothly return to the ground (Fig. 8c), demonstrating its potential for remote airborne operations. In this scenario, the trajectory measures 15.4 m, with an average velocity of 0.10 m/s, and an end-effector ATE of 0.04 m. The lower speed compared to the coffee task reflects the higher precision required for grasping lightweight objects.

## Discussion

In this study, we develop an innovative hand-like compact flying robot for manipulation, which integrates biomimetic grasping and aerial flying capabilities. We propose an efficient autonomous framework that enables the robot to perform precise and smooth aerial manipulation in real-world environments. The proposed robot demonstrates notable performance across indoor and outdoor tasks, showing considerable potential as future smart home assistants, moving flying cameras, aerial teleoperation tools and flying delivery robots. However, current state estimation depends on external localization, and the onboard visual localization module exhibits cumulative drift during long-range outdoor tasks. Achieving full autonomy requires not only accurate localization but also the ability to interpret the environment from visual inputs. End-to-end visual reinforcement learning

approaches[59,60], which directly map perception to control while adaptively correcting state estimation errors, offer a promising route to enhance system autonomy in the future.

We also provide a theoretical analysis of the system control stability, with detailed information provided in Supplementary Information. The experimental results align with and validate this analysis. In addition, we conduct a robustness analysis of the proposed controller, followed by experiments in different scenarios to assess its performance under external disturbances. The results demonstrate that HI-ARM effectively estimates external interference, with the adaptive controller's error gradually converging, ultimately restoring the system to a stable hovering state.

In future study, we plan to integrate force feedback into HI-ARM to enhance its ability to grasp fragile objects, such as eggs. Additionally, we aim to develop an innovative type of inner gripping surface to increase friction on smooth objects. We also plan to incorporate a multi-modal foundation model to enhance the cognitive ability of HI-ARM, enabling it to perform more sophisticated tasks, such as autonomous valve operation in industrial scenarios. Notably, human-robot collaboration demonstrates HI-ARM's teleoperation capabilities, enabling the acquisition of high-quality real-world data to train models with minimized sim-to-real gaps. Furthermore, building on our previous research on aerial swarm[61], we intend to develop a team of HI-ARM robots, which can execute much more complicated tasks such as collaborative transportation. We will continue our in-depth research into miniaturization and autonomy of the proposed flying robot under limited onboard resources, and push it to be industrialized.

## Methods

### Dynamics

In this study, we employ bold lowercase letters to denote vectors (e.g., $\mathbf{v}$) and bold uppercase letters for matrices (e.g., $\mathbf{J}$). Scalars are represented otherwise. As shown in Fig. 2c(iii), we utilize a world frame $\mathbf{W}$ with an orthonormal basis $\{\mathbf{x}_W, \mathbf{y}_W, \mathbf{z}_W\}$, where $\mathbf{z}_W$ is oriented upward, counter to the direction of gravity. The body frame $\mathbf{B}$, situated at the geometric center of the proposed vehicle, is defined with an orthonormal basis $\{\mathbf{x}_B, \mathbf{y}_B, \mathbf{z}_B\}$. Let the position of the vehicle in the world frame be represented by $\mathbf{p}_W = (p_x, p_y, p_z)^\top$, its orientation by the quaternion $\mathbf{q}_W = (q_x, q_w, q_y, q_z)^\top$, and its linear velocity by $\mathbf{v}_W = (v_x, v_y, v_z)^\top$. Additionally, the angular velocity of the vehicle, expressed in the body frame, is given by $\omega_B = (\omega_x, \omega_y, \omega_z)^\top$.

HI-ARM flight system is composed of four parallel rotor-propellers, whose position distribution approximates the 'X' configuration of a conventional quadrotor. Consequently, its flight dynamics model is similar to that of traditional quadrotors, and the device possesses the differential flatness characteristics[54] as general quadrotors. We utilize 6 DoFs to describe the ideal rigid-body kinematics and dynamics model of the robot. For the translational dynamics component, the following equations are utilized:

$$
\begin{aligned}
\dot{\mathbf{p}}_W &= \mathbf{v}_W, \\
\dot{\mathbf{v}}_W &= T\mathbf{z}_B/m + \mathbf{g},
\end{aligned}
\tag{1}
$$

where $T$ and $m$ are the collective thrust and total mass, respectively; $\mathbf{z}_B$ is the $Z$ axis of the body frame expressed in the world frame; $\mathbf{g} = [0, 0, -g]^\top$ is the gravitational vector.

The rotational kinematic and dynamic equations are expressed as

$$
\begin{aligned}
\dot{\mathbf{q}}_W &= \frac{1}{2}\left(\begin{bmatrix} 0 \\ \omega_B \end{bmatrix}_\times\right) \cdot \mathbf{q}_W, \\
\dot{\omega}_B &= \mathbf{J}^{-1}(\tau - \omega_B \times \mathbf{J}\omega_B),
\end{aligned}
\tag{2}
$$

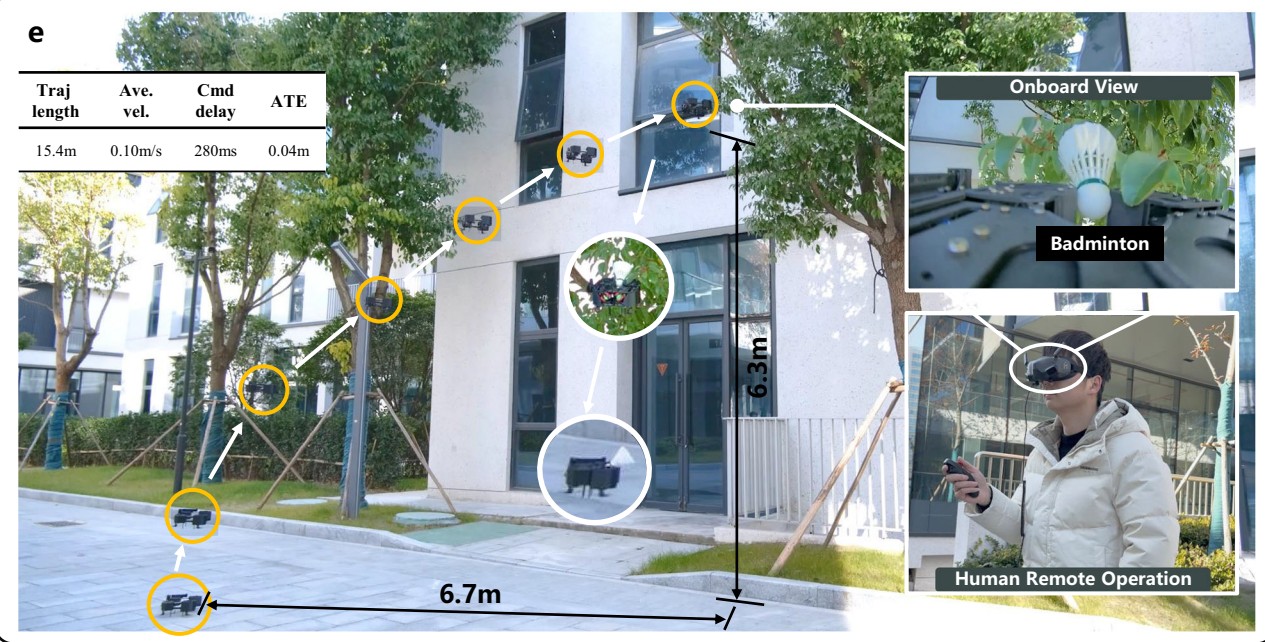

**Fig. 8 | Human-robot collaborative remote aerial operation. a** Illustration of the equipment and principles of human-robot collaborative operation. **b** State curves of HI-ARM during the water bottle grasping process, including position tracking, force estimation, and thrust. **c** Quantitative teleoperation performance, including trajectory length, average velocity, control latency, and average tracking error (ATE). **d** Mobility-impaired people remotely pick up a cup of coffee using HI-ARM. **e** Humans retrieve a badminton from a tall tree using HI-ARM.

where $[\cdot]_\times$ is the skew-symmetric matrix; $\tau$ and $\mathbf{J}$ are the total torque and inertia tensor matrix, respectively, both of which are dynamically updated during morphing and grasping, with more details provided in Supplementary Information.

Let $k_t$ and $k_c$ denote the thrust coefficient and torque coefficient, respectively, associated with the $j$-th motor. The rotational speed of the $j$-th motor is represented by $\Omega_j$, and its position within the body frame is given by $\mathbf{l}_j = [l_{x_j}, l_{y_j}, l_{z_j}]^\top$. The collective thrust $T$ and the torque

$\tau$ generated by the actuators are expressed as follows:

$$\begin{bmatrix} T \\ \tau \end{bmatrix} = \mathbf{H}_k \mathbf{t}, \tag{3}$$

where $\mathbf{t} = [k_t\Omega_1^2, k_t\Omega_2^2, k_t\Omega_3^2, k_t\Omega_4^2]^\top$ represents the thrust generated by each rotor, $\mathbf{H}_k$ is the time-variant mixed control matrix (MCM) while the vehicle morphing. Assuming that the center of gravity is $\mathbf{r}_{COG} = (r_x, r_y, r_z)^\top$ at this time, then $\mathbf{H}_k$ is as follows:

$$\mathbf{H}_k = \begin{bmatrix} 1 & 1 & 1 & 1 \\ r_y + l_{y_1} & r_y + l_{y_2} & r_y + l_{y_3} & r_y + l_{y_4} \\ r_x - l_{x_1} & r_x - l_{x_2} & r_x - l_{x_3} & r_x - l_{x_4} \\ -k_c/k_t & k_c/k_t & -k_c/k_t & k_c/k_t \end{bmatrix}. \tag{4}$$

The module positions $\mathbf{r}_{mod_i}$ and propeller coordinates $(l_{x_i}, l_{y_i})$ and not fixed; instead, they are dynamically updated during morphing and flight. Specifically, these parameters are derived from the tendon-driven morphing kinematics: the servo motor angle $\theta_a$ and rope displacement $L_t$ determine the deformation of both the telescopic and torsional mechanisms, which in turn define the instantaneous propeller positions. As illustrated in Fig. 2c, $L_i'$ and $L_i$ denote the lengths of the telescopic mechanism during motion and under nominal conditions, respectively, with $L_i' - L_i$ representing the compression displacement and reflecting variations in the distances between adjacent modules. As shown in Fig. 2c(ii), $\theta_j$ denotes the rotation angle of the $j$-th torsional mechanism (comprising a torsion spring and a circumferential bearing). Accordingly, the propeller coordinates $(l_{x_i}, l_{y_i})$ are updated to $(l_{x_i}', l_{y_i}')$ as follows:

$$\begin{aligned} l_{x_1}' &= l_{x_1} - (L_i' - L_i)/2 - l_0(1 - \cos\theta_1) \\ l_{x_2}' &= l_{x_2} + (L_i' - L_i)/2 \\ l_{x_3}' &= l_{x_3} + (L_i' - L_i)/2 \\ l_{x_4}' &= l_{x_4} - (L_i' - L_i)/2 - l_0(1 - \cos\theta_2) \\ l_{y_1}' &= l_{y_1} + (L_i' - L_i)/2 + l_0(1 - \sin\theta_1) \\ l_{y_2}' &= l_{y_2} + (L_i' - L_i)/2 \\ l_{y_3}' &= l_{y_3} - (L_i' - L_i)/2 \\ l_{y_4}' &= l_{y_4} - (L_i' - L_i)/2 - l_0(1 - \sin\theta_2) \end{aligned} \tag{5}$$

As a consequence, the control matrix $\mathbf{H}_k$ is updated to $\mathbf{H}_k'$ online according to the current morphing state. This formulation explicitly incorporates geometry changes induced by morphing into the control layer, rather than treating them as static.

HI-ARM utilizes the tendon drive mechanism powered by a single servo motor to actuate deformation. This system enables dynamic and precise structural deformation by transmitting force through a lightweight rope. To facilitate accurate control of morphing movements, we model the deformation structure. As the servo motor's rotary disk rotates, the rope is guided through pulleys, driving coordinated deformations of both passive telescopic and rotational mechanisms, thus enabling body contraction. This study further establishes deformation dynamics equations, providing an effective motion control strategy for the morphing process.

Assume the radius of the servo motor's rotary disk is $R$, its torque is $\tau_a$, the rope tension is $F_t$, the servo motor's angular displacement is $\theta_a$, and the rope displacement is $L_t$. The dynamic equations are expressed as follows:

$$\begin{aligned} F_t &= \frac{\tau_a}{R}, \\ L_t &= \theta_a \cdot R. \end{aligned} \tag{6}$$

As illustrated in Fig. 2c(i), the rope displacement $L_t$ is the sum of the deformations of three telescopic mechanisms and two torsional mechanisms, given by:

$$L_t = \sum_{i=1}^{3} (L_i' - L_i) + \sum_{j=1}^{2} \gamma(\theta_j), \tag{7}$$

where $L_i'$ and $L_i$ represent the lengths of the telescopic mechanisms during motion and under normal conditions. As shown in Fig. 2c(ii), $\theta_j$ denotes the rotation angle of the $j$-th torsional mechanism (comprising a torsion spring and a circumferential bearing), which is $0°$ under normal conditions (Fig. 2c(i)). $\gamma(\theta_j)$ represents the rope displacement corresponding to the rotation angle $\theta_j$.

Assuming the spring stiffness of the telescopic mechanisms is $k_{cs}$ and the torsional stiffness is $k_{ts}$, the forces $F_{c,i}$ applied by the rope on the telescopic mechanisms and the torques $M_{t,j}$ on the torsional mechanisms are:

$$\begin{aligned} F_{c,i} &= k_{cs}(L_i' - L_i), \ i = 1, 2, 3, \\ M_{t,j} &= k_{ts}\theta_j, \ j = 1, 2. \end{aligned} \tag{8}$$

If the moment arm corresponding to the rope tension for the $j$-th torsional mechanism is $l_{t,j}$, the total rope force exerted by the servo motor can be expressed as:

$$F_t = \sum_{i=1}^{3} F_{c,i} + \sum_{j=1}^{2} \frac{M_{t,j}}{l_{t,j}}. \tag{9}$$

Given that the spring constants of all telescopic mechanisms are identical, as are the torsional stiffness values of the torsional mechanisms, and that the body exhibits symmetrical deformation under no-load conditions, the deformation variables of each mechanism satisfy:

$$F_{c,1} = F_{c,2} = F_{c,3}, \quad \frac{M_{t,1}}{l_{t,1}} = \frac{M_{t,2}}{l_{t,2}}. \tag{10}$$

Through the above modeling, we can accurately calculate the motion changes of each module and angular variations of the servo motor during no-load deformation. Combined with the following time-varying curve of the servo motor angle $\theta_a(t)$:

$$\theta_a(t) = \begin{cases} 100t^2, & 0 \leq t \leq 0.1 \\ 12.5t - 0.25, & 0.1 < t < t_0 - 0.1 \\ -100(t - t_0)^2 + 12.5t_0 - 0.5, & t_0 - 0.1 \leq t \leq t_0 \end{cases} \tag{11}$$

where $t_0$ is the total deformation time. Ultimately, these calculations can obtain reference angle commands for the servo motor, thus providing a theoretical foundation for deformation control.

## Flight and deformation control

As previously discussed, the flight dynamics model of HI-ARM closely resembles that of traditional quadrotors. Inspired by research on quadrotor control[54,62], we adopt a simple yet effective geometric tracking control strategy for reference trajectories in Normal Size. To mitigate the effects of perturbations, the $\mathcal{L}_1$ adaptive control algorithm is implemented to compensate for external forces, while the INDI control algorithm is employed to handle external torques. For grasping control, a feedback controller based on angular error is used to facilitate morphing, which is quasi-decoupled from flight control. As illustrated in Fig. 2b, the flight and deformation control form an integrated multi-level adaptive control framework to execute autonomous aerial manipulation tasks. Further details of each component in the control architecture are provided below.

**Geometric tracking control baseline.** The purpose of the geometric tracking controller is to enable the vehicle to track the predetermined trajectory $\mathbf{p}_d(t) \in \mathbb{R}^3$ and yaw angle $\psi(t)$ within the specified time interval $[0, t_f]$. Ignoring disturbances such as the dynamics of the motors and the aerodynamic effects of the propellers, the translational and rotational motions are controlled by the desired thrust and torque as follows:

$$
\begin{aligned}
T &= \parallel -\mathbf{K}_p \mathbf{e}_p - \mathbf{K}_v \mathbf{e}_v - m\mathbf{g} + m\ddot{\mathbf{p}}_W \parallel, \\
\tau &= -\mathbf{K}_R \mathbf{e}_R - \mathbf{K}_\omega \mathbf{e}_\omega + \omega_B \times \mathbf{J}\omega_B - \mathbf{J}(\widehat{\omega}_B \mathbf{R}^\top \mathbf{R}_d \omega_{B,d} - \mathbf{R}^\top \mathbf{R}_d \dot{\omega}_{B,d}),
\end{aligned}
\tag{12}
$$

where $\mathbf{K}_R, \mathbf{K}_\omega \in \mathbb{R}^{3\times3}$ are positive definite gain matrices selected by the user; $\mathbf{R}_d$, $\omega_{B,d}$, and $\dot{\omega}_{B,d}$ are the desired rotation matrix, desired angular velocity, and desired angular velocity derivative, respectively. $\mathbf{e}_R = (\mathbf{R}_d^\top \mathbf{R} - \mathbf{R}^\top \mathbf{R}_d)^\vee / 2$ and $\mathbf{e}_\omega = \omega_B - \mathbf{R}^\top \mathbf{R}_d \omega_{B,d}$ are the rotation error and angular velocity error, respectively. The geometric tracking controller provides a flight control baseline framework for aerial robots, enabling precise flight control in Normal Size.

**$\mathcal{L}_1$ adaptive position control.** The dynamics equation (1) provides a theoretical description of the robot's motion under ideal conditions. However, in real-world scenarios, HI-ARM is also subject to deformation motion disturbances, external forces, and propeller aerodynamic effects. Therefore, in order to improve the control performance of tasks such as aerial grasping, we consider incorporating these uncertainties into the state space representation of the system.

Since these uncertainties appear purely in the robot dynamics, we ignore the kinematic part of the motion dynamics and consider only the dynamics. Disturbances experienced by the proposed robot during translational motion is represented by $\sigma \in \mathbb{R}^3$. Due to the underactuated characteristic of the robot, its rotor thrust is aligned in a single plane and can only provide linear acceleration along the body's $Z$ axis. Therefore, in the actual modeling, we further divide the disturbances into matched modeling $\sigma_m = f_z \in \mathbb{R}^1$ and unmatched modeling $\sigma_{um} = [f_x \, f_y]^\top \in \mathbb{R}^2$.

The $\mathcal{L}_1$ adaptive controller consists of a state predictor, an adaptation law, and a low-pass filter (LPF). Inspired by previous studies[56,63], we select $\mathbf{z} = \mathbf{v}_W \in \mathbb{R}^3$ as the state variables of the dynamics, and the dynamics equation considering external disturbances is:

$$
\dot{\mathbf{z}} = \mathbf{g} + \frac{\mathbf{z}_B}{m}(T + f_{\mathcal{L}_1} + \sigma_m) + \frac{[\mathbf{x}_B \, \mathbf{y}_B]}{m}\sigma_{um}.
\tag{13}
$$

Further, it is written in a more general form as follows:

$$
\dot{\mathbf{z}} = \mathbf{f}(\mathbf{R}_B^W) + \mathbf{g}(\mathbf{R}_B^W)(f_{\mathcal{L}_1} + \sigma_m) + \mathbf{g}^\perp(\mathbf{R}_B^W)\sigma_{um},
\tag{14}
$$

where

$$
\mathbf{f}(\mathbf{R}_B^W) = \mathbf{g} + \frac{\mathbf{z}_B}{m}T, \quad \mathbf{g}(\mathbf{R}_B^W) = \frac{\mathbf{z}_B}{m}, \quad \mathbf{g}^\perp(\mathbf{R}_B^W) = \frac{[\mathbf{x}_B \, \mathbf{y}_B]}{m}.
$$

The state predictor of the $\mathcal{L}_1$ adaptive controller is defined as:

$$
\dot{\mathbf{z}} = \mathbf{f} + \mathbf{g}(f_{\mathcal{L}_1} + \widehat{\sigma}_m) + \mathbf{g}^\perp \widehat{\sigma}_{um} + \mathbf{A}_s \widetilde{\mathbf{z}},
\tag{15}
$$

where $\widetilde{\mathbf{z}} = \widehat{\mathbf{z}} - \mathbf{z}$ is the prediction error, and for simplicity, we assume $\widehat{\mathbf{z}}(0) = \mathbf{z}(0)$. $\mathbf{A}_s$ represents a Hurwitz matrix, which makes the prediction error $\parallel \widetilde{\mathbf{z}} \parallel$ exponentially converge to 0 rapidly. Define $\boldsymbol{\Phi} \triangleq \mathbf{A}_s^{-1}(\exp(\mathbf{A}_s T_s) - \mathbf{I})$, we use a piecewise constant adaptation law formula instead of the projection operator formula because the former is numerically robust. For $t \in [iT_s, (i+1)T_s]$, the piecewise constant adaptation law formula is:

$$
\begin{bmatrix} \widehat{\sigma}_m(iT_s) \\ \widehat{\sigma}_{um}(iT_s) \end{bmatrix} = -\begin{bmatrix} \mathbf{1}_{1\times1} & \mathbf{0}_{1\times2} \\ \mathbf{0}_{2\times1} & \mathbf{1}_{2\times2} \end{bmatrix} \mathbf{G}(iT_s)^{-1} \boldsymbol{\Phi}^{-1}\mu(iT_s),
\tag{16}
$$

where $\mathbf{G}(iT_s) = [\mathbf{g}(\mathbf{R}_B^W) \, \mathbf{g}^\perp(\mathbf{R}_B^W)]$, $\mu(iT_s) = \exp(\mathbf{A}_s T_s)\widetilde{\mathbf{z}}(iT_s)$, and for $i \in \mathbb{N}$.

The $\mathcal{L}_1$ adaptive controller only compensates for the matched components of uncertainties within the strictly proper stable bandwidth of the low-pass filter $\mathbf{C}(s)$:

$$
f_{\mathcal{L}_1} = -\mathbf{C}(s)\widehat{\sigma}_m(s).
\tag{17}
$$

Here we use a first-order low-pass filter as an example, with the transfer function $\mathbf{C}(s) = \omega_c/(s + \omega_c)$. Moreover, the effect of the unmatched modeling part in the formula does not need to be compensated directly; they can be indirectly canceled out by the baseline control law. Proofs regarding stability and bounds on states and controls can be found in ref. 58.

**INDI torque feedback control.** The dynamics equation (2) neglects unmodeled torque terms, which can adversely affect the overall control performance. To address this issue, we adopt INDI, a control strategy based on motor speed and IMU feedback. This approach uses instantaneous sensor measurements to represent system dynamics and is capable of responding quickly to input commands in real time.

INDI is also robust against model uncertainties and external disturbances, and its effectiveness and robustness of INDI have been confirmed in previous studies[57,64]. Drawing on previous work[57], we design an INDI torque compensation control method for HI-ARM.

We re-model the rotational dynamics equation (2), incorporating external torque disturbances $\tau_{ext}$, and its expression is given by:

$$
\dot{\omega}_B = \mathbf{J}^{-1}(\tau + \tau_{ext} - \omega_B \times \mathbf{J}\omega_B).
\tag{18}
$$

Using angular velocity, angular acceleration, and control torque, we can calculate the external torque disturbance, as expressed below:

$$
\tau_{ext} = \mathbf{J}\dot{\omega}_{B,f} - \tau_f + \omega_{B,f} \times \mathbf{J}\omega_{B,f},
\tag{19}
$$

where $\tau_f$ is the control torque in the body coordinate system, which is obtained by measuring the motor speed through low-pass filter. The terms $\omega_{B,f}$ and $\dot{\omega}_{B,f}$ represent the measured body angular velocity and angular acceleration, respectively. The external torque $\tau_{ext}$ is assumed to vary slowly relative to the LPF dynamics. Substituting the above equation (19) into equation (18), we get:

$$
\begin{aligned}
\dot{\omega}_B &= \mathbf{J}^{-1}(\tau + \tau_{ext} - \omega_B \times \mathbf{J}\omega_B) \\
&= \mathbf{J}^{-1}(\tau + (\mathbf{J}\dot{\omega}_{B,f} - \tau_f + \omega_{B,f} \times \mathbf{J}\omega_{B,f}) - \omega_B \times \mathbf{J}\omega_B) \\
&= \dot{\omega}_{B,f} + \mathbf{J}^{-1}(\tau - \tau_f).
\end{aligned}
\tag{20}
$$

In equation (20), we assume that the difference between the gyroscopic torque term and its filtered counterpart is sufficiently small to be negligible, as it changes relatively slowly compared to angular acceleration and control torque.

Using the four motor speed commands, the desired total thrust and angular acceleration can be inferred. By inverting equation (20), we obtain the incremental expression of the desired control torque command $\tau_{\text{INDI}}$ as follows:

$$
\tau_{\text{INDI}} = \tau_f + \mathbf{J}(\widehat{\boldsymbol{\omega}}_{B,d} - \dot{\omega}_{B,f}).
\tag{21}
$$

**Multi-level adaptive flight control.** External forces and torques estimated using the aforementioned $\mathcal{L}_1$ adaptive position control and INDI torque feedback algorithms are incorporated into a geometric tracking baseline through feedback compensation. The final form of the multi-

level adaptive flight control module can then be represented as follows:

$$T_d = \| -\mathbf{K}_p \mathbf{e}_p - \mathbf{K}_v \mathbf{e}_v - m\mathbf{g} + m\ddot{\mathbf{p}}_W \| - f_{\mathcal{L}1},$$
$$\tau_d = -\mathbf{K}_R \mathbf{e}_R - \mathbf{K}_\omega \mathbf{e}_\omega + \omega_B \times \mathbf{J}\omega_B - \mathbf{J}(\hat{\omega}_B \mathbf{R}^\top \mathbf{R}_d \omega_{B,d} - \mathbf{R}^\top \mathbf{R}_d \dot{\omega}_{B,d}) - \tau_{\text{INDI}}.$$

$$(22)$$

**Servo motor control.** The nylon rope towed by the servo motor is lightweight, and the tension experienced during deformation is much greater than the friction, allowing us to assume uniform tension in the rope for simplification. In the actual grasping scenario, HI-ARM's deformation is controlled by the servo motor to accurately grasp a target object. With an initial reference angle of the servo motor set to $\theta_r$ and a reference angular velocity $\Omega_r$, we design a proportional controller based on the angle position error. The servo motor's rotational angular velocity is adjusted as follows:

$$\Omega_d = K_\theta(\theta_r - \hat{\theta}) + \Omega_r, \qquad (23)$$

where $\hat{\theta}$ is the estimated angle of the servo motor, and $K_\theta$ is the control gain. Since the initial reference angle may not be perfectly accurate due to varying object sizes and shapes, torque sensing becomes crucial. The servo motor includes torque feedback and overload protection mechanisms to ensure precise grasping control. If the set angle reaches or the motor torque exceeds the blocking torque, the grasping action is considered complete, indicating that the robot has successfully grasped the object.

## Trajectory planning

Quadrotors possess differential flatness, meaning that their system states and system inputs can be represented by a set of system outputs such as $[x, y, z, \psi]^\top$[54]. This characteristic provides a foundation for the drone such as HI-ARM to achieve fast and efficient trajectory planning. Inspired by our previous work[55], we employ a trajectory representation method called MINCO, which is based on the differential flatness of quadrotors, to plan the trajectory in space-time for flight. The advantage of MINCO is that it allows users to independently adjust the spatial and temporal attributes of the path, creating efficient operations with linear complexity and making spatial-temporal deformation more convenient.

The MINCO piecewise trajectory involves two main parameters: first, the duration of each segment, represented by $\mathbf{T} \in \mathbb{R}^M$; second, the waypoints connecting each segment, represented by $\mathbf{q} \in \mathbb{R}^{3 \times (M-1)}$, where $M$ is the number of segments. Then, the three-dimensional spatial point $p(t)$ on the MINCO trajectory at a time $t$ is determined through the operation $\mathcal{M}$:

$$p(t) = \mathcal{M}_{\mathbf{q},\mathbf{T}}(t). \qquad (24)$$

For the s-integrator chain dynamics (in this work s=3), the MINCO trajectory is a $\mathcal{C}^{s-1}$ polynomial spline of degree 2s-1 by default, with constant boundaries and minimum control effort given $\{\mathbf{q}, \mathbf{T}\}$. Since we are using a jerk-based control system model, smoothness is maximized by minimizing control effort. For the trajectory defined over the time domain $t \in [t_0, t_M]$, the control effort optimization is given by:

$$\min_{p(t)} \int_{t_0}^{t_M} \| p^{(s)}(t) \|^2 dt. \qquad (25)$$

MINCO demonstrates its advantage of linear complexity when converting user-defined parameters $\{\mathbf{q}, \mathbf{T}\}$ into polynomial coefficients $\mathbf{c}$ and time profiles $\mathbf{T}_p$. This conversion can be achieved through a non-singular banded matrix $\mathbf{M} \in \mathbb{R}^{2Ms \times 2Ms}$ and $\mathbf{b} \in \mathbb{R}^{2Ms \times 3}$, which are valid for any $\mathbf{T} > 0$, as follows:

$$\mathbf{M}(\mathbf{T})\mathbf{c} = \mathbf{b}(\mathbf{q}), \mathbf{T}_p = \mathbf{T}, \qquad (26)$$

at the same time, the restoration of the trajectory through banded PLU Factorization also has linear complexity, and the gradients of the polynomial coefficients can also be propagated to MINCO parameters in linear time. More details on the representation of MINCO trajectories can be found in the literature[55].

Based on the above advantages of MINCO trajectories, we design a trajectory planning method for HI-ARM. To achieve smooth motion and efficient flight for HI-ARM, we define two metrics for smoothness and time, and minimize their weighted sum. Decision variables are the MINCO parameters $\mathbf{q}$ and $\mathbf{T}$.

Within the time $t \in [t_0, t_M]$, the MINCO trajectory optimization form is as follows:

$$\min_{\mathbf{q},\mathbf{T}} \int_{t_0}^{t_M} J(\mathbf{q}, \mathbf{T}, t) \, dt,$$
$$\text{s.t. } \mathcal{H}(\mathbf{q}, \mathbf{T}, t) = 0, \mathcal{G}(\mathbf{q}, \mathbf{T}, t) \leq 0, \qquad (27)$$

where $\mathcal{H}$ and $\mathcal{G}$ represent equality and inequality constraints, respectively, including continuity constraints for start and end states, dynamic feasibility constraints, time constraints, etc. According to the literature[65], the above constrained optimization problem can be further transformed into an unconstrained problem for more efficient solution. The transformed trajectory optimization problem is as follows:

$$\min_{\mathbf{q},\mathbf{T}} \sum_x \lambda_x J_x, \qquad (28)$$

where $J_x$ are various penalty terms, and $\lambda_x$ are relative weights. The subscript $x = \{s, t, d\}$ represents smoothness (s), total time (t), and dynamic feasibility (d), etc. Their corresponding detailed penalty functions are as follows:

Maximizing Smoothness: According to equation (25), the smoothness penalty $J_s$ is defined as the integral of the square of the s-order derivative, i.e.,

$$J_s = \int_{t_0}^{t_M} \| p^{(s)}(t) \|^2 \, dt. \qquad (29)$$

This integral can be analytically calculated because the MINCO trajectory can be represented as a piecewise polynomial according to equation (26).

Minimizing Total Time: In most cases, shorter flight time is desirable, so we also minimize the weighted total flight time to obtain the total time penalty $J_t$ as

$$J_t = \text{sum}(\mathbf{T}). \qquad (30)$$

Dynamic Feasibility: For differentially flat multicopters, dynamic feasibility is ensured by limiting the magnitude of trajectory derivatives. In our work, we add penalties to limit the amplitude of velocity, acceleration, and jerk if these derivatives exceed physical thresholds, i.e.,

$$J_{d,v} = \sum_{i=0}^{\kappa} \max \{(\dot{p}(t_i)^2 - v_m^2), 0\}^3, \qquad (31a)$$

$$J_{d,a} = \sum_{i=0}^{\kappa} \max \{(\ddot{p}(t_i)^2 - a_m^2), 0\}^3, \qquad (31b)$$

$$J_{d,j} = \sum_{i=0}^{\kappa} \max \{(p^{(3)}(t_i)^2 - j_m^2), 0\}^3, \qquad (31c)$$

$$J_d = J_{d,v} + J_{d,a} + J_{d,j}, \qquad (31d)$$

where $v_m$, $a_m$, and $j_m$ are the maximum permissible magnitudes of velocity, acceleration, and jerk, respectively; $t_i = t_0 + (t_M - t_0)i/\kappa$ indicates a finite number of sampled timestamps, where $\kappa + 1$ equals the sample number. We sum $J_{d,v}$, $J_{d,a}$, and $J_{d,j}$ directly because they are of similar magnitude.

Finally, we begin to solve the trajectory optimization problem (28) using the open-source L-BFGS solver. The trajectory planner starts with the global target position given by the user and iteratively optimizes to ultimately generate a local trajectory that meets task requirements. Specific details can be found in the literature[55].

## Data availability
The data that support the findings of this study are publicly available on Zenodo at https://doi.org/10.5281/zenodo.17340496.

## Code availability
The code of the autonomous manipulation mode used in this study is publicly available at (https://github.com/Wyz000/HIARM)(https://doi.org/10.5281/zenodo.17310380).

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

## Acknowledgements

We thank M. Wang, T. Wu, Y. Zhong, X. Zhou, T. Zhang, Y. Gao, who offered valuable suggestions to the manuscript, and R. Jin for photography and video recording. We sincerely appreciate the work of J.W. and Y.Z. for help on experiments. Furthermore, we are truly grateful for J. Zhang's help in artwork. This work was supported by the National Key R&D Program of China under grant no. 2023YFB4706600 and the National Natural Science Foundation of China under grant no. 62322314.

## Author contributions

Y.W. contributed to the hardware and software design, experiments, and manuscript writing. F.Y. contributed to the hardware design, controller design, and experiments. R.J. contributed to artwork and experiments. Y.Z. contributed to experiments and gave several suggestions for manuscript writing. J.W. and X.W. contributed to experiments. F.G. directed the research, provided the primary idea and funding with some key suggestions about software and hardware debugging, and revised the manuscript.

## Competing interests

The authors declare no competing interests.
