## [Transparent Peer Review file · Nature Communications]

Hand-like Autonomous Flying Robot for Airborne Grasping and Interaction

Corresponding Author: Professor Fei Gao

Version 0:

Reviewer comments:

Reviewer #1

(Remarks to the Author)

> Key results

The paper presents the design, development and validation of a novel type of aerial manipulation robot serving as a flying hand to assist people. It relies on a quadrotor configuration whose mechanical structure in U-shape, built in carbon fiber, replicates the shape of the human hand to allow object grasping, perching on poles, interacting with the environment (like opening doors), and allows morphing for flying through narrow spaces. This is achieved by incorporating three prismatic joints on the corresponding links of the U-shape structure and two revolute joints at the finger tips that are actuated by compression and torsion springs respectively for keeping the structure expanded. A pulley-cable mechanism driven by a continuous rotation servo implements the contraction of the structure. The paper demonstrates through an outstanding video several applications and ways in which the aerial robot can serve the human users by delivering objects in domestic and outdoor environments, helping also people with mobility problems. It also demonstrates other capabilities, like perching, opening doors, grasping different objects, and flying through narrow obstacles.

The concept is very nice, the video demonstration very illustrative, and the design approach suitable in terms of size of the platform taking into account the integration in human environments. The selected methods and technologies seem also appropriate.

> Validity

Although the concept is really good, there are some aspects requiring clarification regarding the automatic operation, particularly since this feature is pointed out in the title of the paper. The impression is that the aerial robot is manually operated by an skilled human pilot in some cases in which a position estimation cannot be obtained. The main problem for achieving autonomous operation is implementing onboard an accurate positioning system, taking into account weight and size constraints. In the video indoors (in the domestic scenario), I guess a positioning system like Vicon/Optitrack was used since the IR balls can be seen in the frame structure of the aerial robot. This is not necessarily bad or something to hide, but it could be stated for example that this application would require deploying some cameras in the house to localize the robot indoors. Also, localization outdoors is non trivial without LiDAR, relying only on camera. It is stated that the system employs an Intel stereo pair for localization, but it is difficult to determine if navigation outdoors relies on on-board perception.

The design of the aerial robot is not properly described in the manuscript. It would be convenient to represent a 3D model showing the integration of the onboard computer (does it really fits within the frame structure), batteries, camera, ... Also please, include dimensions of the platform.

Regarding the pulley-cable mechanism and the springs, you should clarify why the design of the configuration of the gripper functionality is open by default instead of close

I also have some doubts about the real payload capacity. It is stated that the weight of the platform is 0.55 kg, providing around 0.4 kg payload, whereas the four propellers achieve 1 kg thrust. It should be specified what is the throttle percentage

for 1 kg. For example, if 1 kg thrust is achieved at 80% throttle, this is very close to actuators saturation and probably cannot be maintained more than 2 min or less. The LiPo battery is 2x 550 mAh 3S, that, given the model of propellers, probable last for less than 5 min flight.

> Originality and significance

The proposed concept is novel.

The methods implemented are well chosen, but no novelty is introduced. L1 is a good adaptive controller. The planning scheme is logical, but not completely original.

The application and demonstration is outstanding.

> Data & methodolog

Authors employ the expression millisecond or microsecond to remark the time performance of the methods, but proper numerical results should be provided for the selected computer board.

For perching on poles, it requires active energy insertion. Which is the most suitable configuration by default: open or close? Close by default would be more convenient for saving energy, but this would require changing the spring configuration (from compression to extension).

In Figure 4-b: External estimation is not very good if force is estimated around 2.2 N while the bottle is 1.5 N. Provide thrust or PWM values from flight controller to validate the test.

Fully autonomy should be properly demonstrated, particularly regarding the on-board position estimation, both indoors/outdoors. This should be clarified in the manuscript.

Fig. 5-e: why power is not plotted in Watts?

Fig. 5-i: The force figure seems a pattern typical of manual flight rather than an automatic flight control.

Fig. 6-i: These payloads are very low compared to the 0.45 kg payload claimed before.

> Conclusions

The experimental validation regarding the fully autonomous operation is not completely convincing. It seems in some cases external positioning systems are employed, or manual flight. This is important, since the title and scope of the paper is also autonomy, and the platform size/weight is determined by the weight of the on-board components. Having a small size of platform is essential for safe interaction with humans, involving weight reduction. In this sense, on-board positioning is particularly challenging.

> Suggested improvements

- To my understanding, the nylon cable, which is stiff, is used to close the hand. Therefore, there is no compliant adaptation in the closing procedure. The action of the springs, which provide the compliance, is for opening the gripper. The servo motor will be stalled, and this may damage the gearbox or increase the current consumption excessively.
- Clarify which is the thickness of the carbon fiber boards, also provide a more in detail picture of the mechanical construction, showing the assembly of the mechanical elements (linear guides) and the on-board components (battery, computer board, camera). How is the camera connected to the computer board?
- Aerial "manipulative" sounds strange. Why not "manipulation"?
- The sentence "The servo motor provides torque state feedback, which helps determine the success of the gripping action". The servo employed seems the typical RC PWM servo. Does it really provides feedback? What is more, if it is continuous rotation, how do you address the problem of multi-turn counting?
- "non-negligible damping" I think friction is more suitable.
- In Human-robot collaborative operation (pg. 18), which control reference is provided by the user, velocity? How can a 3D reference be provided with a single input device? Drone pilots need two sticks.
- Regarding the dynamics, which are the transformations applied to account for the displacement of the IMU with respect to the center of the platform?
- Since there is a single actuator for controlling the deflection of three linear segments and two torsion sprints (subactuation), it should be established an order in the passive actuation, which is determined by the relative value of stiffness. For example, if it is desired that the finger joints rotate first, their stiffness should be lower compared to the ones of the linear guides.

> References

The state of the art section includes relevant references, but it is not up to date. See for example the survey "Past, present

and future of aerial robotic manipulators" published in T-RO. Other related references: "Physical human-aerial robot interaction and collaboration: Exploratory results and lessons learned", "Through-Window Home Aerial Delivery System with In-Flight Parcel Load and Handover: Design and Validation in Indoor Scenario", "Autonomous drone delivery to your door and yard".

> Clarity and context

There are some statements at the Introduction that require revision. For example, birds also use their beak to perform manipulation tasks, becoming quite skilled (like the parrots). The sentence "Flying robots [...] are highly anticipated to deeply participate in our social activities, especially safety-critical ones" is not clear to me. Also "replacing humans" is an statement that should be used carefully, since human workers do not like to be replaced. In general, the introduction should be revised and improved, it is not properly written. Sentences as the first one on page 3: "making them unsuitable for sophisticated operations" are controversial.

> Inflammatory material

Nothing detected.

Reviewer #2

(Remarks to the Author)
General Assessment

The authors present the design, modeling, control, and experimental validation of a compact aerial robot that integrates flight and manipulation, enabled by a novel morphing mechanism. The system is supported by an autonomous control framework and showcases various demonstrations ranging from object grasping to human-robot interaction. The breadth of experimental results and the robot's operational versatility are highly impressive, and they convincingly demonstrate the potential utility of the proposed platform.

However, the repeated emphasis on the "hand-inspired" or "bio-inspired" nature of the robot feels somewhat overstated. While the robot features an interesting grasping mechanism, its similarity to the biological hand remains limited, both structurally and functionally. Furthermore, while the teleoperation demonstration is visually compelling, it is only loosely related to the essential of the proposed robot and may not be scientifically appropriate to include as a core contribution in this paper.

Major Comments

Overstatement of "hand-inspired" design: The proposed mechanism is certainly innovative, but the claim that it is "hand-inspired" may be an overreach. From a mechanical perspective, the device offers only basic grasping functionality and, at most, resembles a gripper rather than a biologically inspired hand. Thus, referring to it as "hand-inspired" seems exaggerated. Additionally, the linear actuated motion, in particular, is not biologically inspired and should instead be recognized as a novel mechanical design in its own right. Moreover, the drone is restricted to $xyz + \text{yaw}$ (i.e., SE(2)+R) motions and can only grasp vertically elongated objects. Given this limitation, the claim of generalized aerial manipulation is arguably overstated.

Unclear sensing of morphing-related parameters: A central concern is how the system handles changes in geometry due to morphing, especially regarding the shifting propeller positions. For example, the propeller coordinates l_{x_i}, l_{y_i} in Equation (4), and r_{mod_i} in Supplementary Eq. (S2), must vary according to the deformation. Yet it is unclear how these are measured or estimated during flight. The software architecture diagram (Fig. 2b) does not provide any indication of how this information is integrated into the control layer. Clarification on this point is critical for assessing the feasibility of the proposed control framework.

Lack of comparative validation for control schemes: The proposed adaptive controller, including L1 and INDI schemes, appears theoretically sound, but its practical necessity remains unclear. Comparative experiments with baseline controllers (e.g., PID or standard geometric control) are essential to justify the added complexity. In particular, the perching detachment sequence shown at ~2:48 in the main video appears unstable, raising concerns about the real-world robustness of the control system.

Demonstrations lack scientific depth: While the demonstrations are entertaining and diverse, many are superficial in their analysis. The "Interaction with humans" scenario appears overly choreographed and lacks scientific rigor. Similarly, the "Human-robot collaborative operation" section is interesting from an application standpoint but largely unrelated to the core mechanical and control contributions of the paper. It would be more appropriate to present this part as a separate study focused on teleoperation, with quantitative analysis of flight behavior during human operation.

Minor Comments

INDI control: Why is the external torque estimated and compensated, but not the external force? Given that external forces can also affect flight performance, this choice needs justification.

Door opening (Fig. 5): A door that can be opened with only 2–3 N is not realistic in most practical settings. Moreover, it seems like the robot can also push the door directly without actually grasping the handle, which undermines the credibility of this demonstration. Besides, what is the actual maximum pushing force the robot can exert?

Applications in the wild (Fig. 7): The narrow-space traversal demo appears unconvincing, as the passage seems wide enough even without morphing. What are the robot's maximum and minimum widths before and after deformation?

Propeller placement: Placing propellers below the robot may aid manipulation, but it also raises safety concerns, especially regarding potential collisions with the environment during tilted flight. The authors should discuss the implications of this design decision. Also, no flight endurance is clearly stated for the 3.5-inch propellers used.

Conclusion

While the system integration is commendable and the demonstrations are engaging, the manuscript contains numerous exaggerated claims and ad-hoc experimental setups. The core mechanical design is interesting and merits publication, but the current framing and presentation are not well-aligned with the standards of Nature Communications. A more suitable venue would be a specialized robotics or mechatronics journal where system integration and design novelty are more central.

Reviewer #3

(Remarks to the Author)

The authors present an aerial robot with morphing capabilities, enabling functionalities such as grasping and perching. They also provide a detailed account of trajectory generation and control strategies. However, I have the following questions and suggestions for clarification:

1. The authors state that their work offers improvements over other aerial robots (e.g., references 40–47). It would strengthen the paper to include a clear, quantitative comparison of the proposed platform with existing systems. Relevant metrics might include platform size, weight, maximum payload, flight time, and other performance indicators.
2. It appears that the front and rear motor cases differ in height. Is there a specific design rationale for this asymmetry? Would having equal heights simplify control or improve performance in any way?
3. For the autonomous tasks demonstrated, it seems the robot is aware of the positions and orientations of the target objects (e.g., the water bottle and door handle). If this is the case, the authors should state this assumption explicitly. Otherwise, could the authors clarify how the robot estimates the positions and orientations of these objects?

Version 1:

Reviewer comments:

Reviewer #1

(Remarks to the Author)

I would like to thank Authors for addressing my comments and suggestions. I am aware of the time and effort devoted to understand my points, conduct the new experiments, and provide a suitable response. The response letter is clear and well presented. The provided responses are convincing and concise. Authors provide the requested information, and they go even further for example in the control ablation study. Just three minor clarification are required: 1) in the endurance flight tests (Comment 5), it is stated that the aerial robot can fly for 237 s in the open configuration, and 212 s in the close configuration due to the thrust loss. In both cases the endurance test is conducted for the 450 grams weight? Please, provide the maximum flight time for the no load and full load cases; 2) the time metrics relative to the planning method (Comment 7) are quite low, in the range of ms and micro-second. For a computer board with a processor running at 1.8 GHz, I guess that the complexity of the trajectories is relatively low to achieve such low computational time, or rather the methods is very efficiently implemented; 3) In the external load estimation (Comment 9) I think the plot of the external force is the same on pages 13 and 14 on the response letter. I expected to see here the corrected estimation compensating the thrust loss to achieve the 1.5 N ground truth. Please, check also the manuscript. Finally, I recommend authors to perform a full proof-read of the paper to make sure there is no typo.

(Remarks on code availability)

Reviewer #2

(Remarks to the Author)

Thank authors for addressing the concerns raised in my previous review round. The revisions significantly improve the clarity and rigor of the manuscript, and I am overall satisfied with the authors' responses and the updated content.

There is one remaining minor comment that should be clarified before final acceptance:

Immediately following Equation (4), the expression referring to "the module positions $r_{\{\text{mod}_i\}}$ " lacks sufficient explanation. It is not clear to the reader which figure or diagram provides the spatial definition or visualization of this symbol. A pointer to the appropriate figure (e.g., Fig. X) would greatly help readers.

(Remarks on code availability)

Reviewer #3

(Remarks to the Author)

The authors have addressed my questions and suggestions.

(Remarks on code availability)

Revision Report for NCOMMS-25-21863

Hand-Inspired Autonomous Flying Robot for Airborne Manipulation

Yuze Wu, Fan Yang, Rui Jin, Yuhang Zhong, Junjie Wang, Xuankang Wu, and Fei Gao

The authors of the paper would like to thank the Editors and Reviewers for their timely handling of our manuscript and useful feedback. They have helped us improve our paper in several aspects, for which we are really grateful. We provide this document as the reply to the previous review comments and the revision of the submitted manuscript (“Hand-Inspired Autonomous Flying Robot for Airborne Manipulation”, submission ID NCOMMS-25-21863).

As instructed, the paper has been modified and extended according to the comments from editors and reviewers. Details on the revisions and specific answers to all queries are provided herein and can also be found in the table below for further clarification. Please note that, the comments of editor and reviewer are shown in red boxes. Author responses appear in normal font. Text of the revised manuscripts is shown in blue boxes with deletions marked and revised text in red. Text quoted from the manuscripts is underlined. Figure, table, and equation numbers in the Supplementary Information are preceded by an “S”, e.g., Fig. S1, Table S2, and Eq. S3. Elements appearing with an “R” prefix, e.g., Fig. R1 correspond to this response letter, while unmarked elements, e.g., Fig. 1, correspond to the main text. Herein, main text, Supplementary Information, and response letter are abbreviated as M.T., S.M., and R.L., respectively.

Table R1. Explanation of abbreviations, notations and annotations.

Abbreviation	Explanation
M.T.	Main Text
S.M.	Supplementary Information
R.L.	Response Letter
Notations and Annotations	Explanation
Normal Font	Author responses
red boxes	Reviewer comments
blue boxes	Revised text
deletions	Deletions
red	Revised text in red
underlined	Quoted text from the manuscripts
S	Supplementary Information figure, table, and equation numbers
R	Figure and table numbers in the Response Letter

Responses to Comments by Reviewer #1

Comment 1

Key Results

The paper presents the design, development and validation of a novel type of aerial manipulation robot serving as a flying hand to assist people. It relies on a quadrotor configuration whose mechanical structure in U-shape, built in carbon fiber, replicates the shape of the human hand to allow object grasping, perching on poles, interacting with the environment (like opening doors), and allows morphing for flying through narrow spaces. This is achieved by incorporating three prismatic joints on the corresponding links of the U-shape structure and two revolute joints at the finger tips that are actuated by compression and torsion springs respectively for keeping the structure expanded. A pulley-cable mechanism driven by a continuous rotation servo implements the contraction of the structure. The paper demonstrates through an outstanding video several applications and ways in which the aerial robot can serve the human users by delivering objects in domestic and outdoor environments, helping also people with mobility problems. It also demonstrates other capabilities, like perching, opening doors, grasping different objects, and flying through narrow obstacles.

The concept is very nice, the video demonstration very illustrative, and the design approach suitable in terms of size of the platform taking into account the integration in human environments. The selected methods and technologies seem also appropriate.

Response: We sincerely appreciate your recognition of our contributions, which reinforces our motivation to continue addressing scientific and engineering challenges of aerial manipulation within the robotics community.

Comment 2

Validity

Although the concept is really good, there are some aspects requiring clarification regarding the automatic operation, particularly since this feature is pointed out in the title of the paper. The impression is that the aerial robot is manually operated by a skilled human pilot in some cases in which a position estimation cannot be obtained. The main problem for achieving autonomous operation is implementing onboard an accurate positioning system, taking into account weight and size constraints. In the video indoors (in the domestic scenario), I guess a positioning system like Vicon/Optitrack was used since the IR balls can be seen in the frame structure of the aerial robot. This is not necessarily bad or something to hide, but it could be stated for example that this application would require deploying some cameras in the house to localize the robot indoors. Also, localization outdoors is non-trivial without LiDAR, relying only on camera. It is stated that the system employs an Intel stereo pair for localization, but it is difficult to determine if navigation outdoors relies on on-board perception.

Response: We sincerely thank your insightful suggestions, which help us present our work in a more comprehensive manner and allow readers to better understand our contributions.

For precise tasks in indoor scenarios, high-accuracy state estimation systems, such as LiDAR, are typically required, and existing technologies can indeed achieve efficient real-time localization and mapping [1, 2]. However, LiDAR sensors are generally heavy; to our knowledge, even one of the lightest models, the Livox Mid-360, weighs 265 g¹. Deploying such a sensor on our compact aerial robot platform, which weighs only 556 g, poses significant challenges. Therefore, we make a trade-off by employing an

¹Livox Mid-360 LiDAR specifications: <https://www.livoxtech.com/mid-360>

external camera system (e.g., a motion capture system) to obtain the states of both the robot and the objects. We regret that this rationale is not sufficiently explained in the original manuscript, and now add a detailed description of the localization and estimation strategy in the revised version.

In outdoor scenarios, we use the Intel RealSense T261 tracking module². Although its accuracy is lower than LiDAR-based approaches, it is considerably lighter and provides relatively stable short-range localization performance (for example, within 20 m), which is sufficient for validating outdoor applications. Accordingly, we adopt an onboard vision module for state estimation, thereby completing the final step of the autonomous manipulation in outdoor experiments.

Furthermore, as rightly noted, achieving precise, robust, and lightweight localization remains a formidable challenge. Although state estimation is not the primary focus of this work, we recognize it as a crucial step toward achieving higher levels of autonomy. We explicitly acknowledge this challenge and highlight lightweight and robust onboard localization and estimation as key directions for future research. A detailed discussion of this aspect has been incorporated into the revised *Discussion* section of the manuscript.

Revise

Introduction

...

To accomplish complicated tasks with precision and smoothness, autonomy is also crucial for HI-ARM. To this end, we propose a framework consisting of a task planner, trajectory generation, state feedback, **parameter estimation**, and adaptive control (Fig. 2b). HI-ARM supports two working modes: (1) ~~fully~~ autonomous operation, where the task planner selects proper operating sequences from an action library according to input task types **and the robot executes them under closed-loop perception, planning, and control** (e.g., grasping, perching, door opening, and human interaction), and (2) human-robot collaboration, where the task planner generates reference trajectories and grasping commands from human intentions, **and the controller subsequently tracks these commands**.

...

Results

Flight design and electronic components

This section provides a detailed overview of the HI-ARM flight system, which is powered by four rotor-propellers. As shown in Figure 2a, HI-ARM is equipped with 3.5-inch propellers, each driven by a T-motor F1404 2900KV motor, capable of generating a combined thrust of up to 1000g. The motors are mounted beneath the finger and palm modules, and the bottom-mounted layout avoids direct propeller airflow and enhances human-robot interaction safety by minimizing hand contact. The propellers are designed with a 6 mm height offset to prevent interference during module deformation and folding. With a 70C battery discharge rate, the system delivers sufficient instantaneous power output to handle payloads of over 450g for grasping operations. The primary structure of the flight platform is composed of thick carbon fiber plates, selected for their high strength and lightweight properties. For flight control, HI-ARM uses a Kakute H7 mini flight control board to run the ArduPilot flight firmware, which collects high-frequency (>300Hz) real-time data on motor speed from the electronic speed controller (ESC) and posture from the Inertial Measurement Unit

²Intel RealSense T261 tracking camera specifications: https://www.mouser.com/pdfDocs/Intel_RealSense_Tracking_Camera_Datasheet_Rev004_r-1665928.pdf

(IMU). High-frequency feedback helps the robot to estimate its motion states quickly, reducing system delay and improving control accuracy. The on-board computer, a Radxa ZERO board, runs a mission planner and adaptive controller (to be introduced in the following sections) for autonomous aerial manipulation. This controller processes system state data from the IMU, the servo motor, and the localization module, then sends control inputs to the brushless motor and the servo motor, which adjusts motor speeds for precise flight operations. More details about the components can be found in *Supplementary Information*.

For localization and estimation, we aim to balance accuracy, weight, and onboard computational load. In indoor experiments, an external optical motion-capture system provides sub-millimeter state estimation of both the aerial robot and the manipulated objects, enabling closed-loop integration with mission planning and adaptive control for autonomous operation. In outdoor experiments, we employ the Intel RealSense T261 tracking module, which weighs 26 g and offers a lightweight alternative to LiDAR-based localization systems while maintaining stable short-range state estimation (for example, within 20 m) suitable for experimental validation.

...

Discussion

In this study, we develop an innovative hand-inspired compact flying robot for manipulation, which integrates biomimetic grasping and aerial flying capabilities. We propose an efficient autonomous framework that enables the robot to perform precise and smooth aerial manipulation in real-world environments. The proposed robot demonstrates excellent performance across indoor and outdoor tasks, showing considerable potential as future smart home assistants, moving flying cameras, aerial teleoperation tools and flying delivery robots. However, current state estimation depends on external localization, and the onboard visual localization module exhibits cumulative drift during long-range outdoor tasks. Achieving full autonomy requires not only accurate localization but also the ability to interpret the environment from visual inputs. End-to-end visual reinforcement learning approaches [3, 4], which directly map perception to control while adaptively correcting state estimation errors, offer a promising route to enhance system autonomy in the future.

...

Comment 3

The design of the aerial robot is not properly described in the manuscript. It would be convenient to represent a 3D model showing the integration of the onboard computer, batteries, camera, etc. Also please include dimensions of the platform.

Response: We sincerely appreciate your suggestions. We have added the 3D model figure to clearly illustrate the structure, battery, camera, and other critical components, and have supplemented the relevant dimensions and measurements. To provide a clearer view of the internal configuration, the external shell is rendered transparent. As shown in Figure R1c, the layout of key onboard components—including the battery, onboard computer, and flight controller—is explicitly displayed. The Intel RealSense T261 module is mounted on the upper side of the body for visual sensing and localization, as shown in Figure R1a.

Experimental platform and parameters

This section presents the experimental platform of the robot, as shown in Figure R1. To provide a clearer view of the internal configuration, the external shell is rendered transparent. As shown in Figure R1c, the layout of key onboard components—including the battery, onboard computer, and flight controller—is explicitly displayed. The Intel RealSense T261 module is mounted on the upper side of the body for visual sensing and localization, as shown in Figure R1a. ... HI-ARM's physical parameters such as weight, dimensions, and inertia tensor are shown in Tab. R2. ...

Fig. R1. Hardware components of HI-ARM. (a) Components description. (b) Electronics architecture. (c) 3D structural illustration.

Table R2. Initial configuration of HI-ARM.

Parameters	Values
$m[g]$	556
$k_t[N \cdot s^2]$	$4.602e^{-9}$
$k_c[Nm \cdot s^2]$	$7.536e^{-11}$
$r_{COG}[m]$	(-0.007, -0.002, 0.000)
$(J_{xx}, J_{yy}, J_{zz}) [kg \cdot m^2]$	(0.0023, 0.0024, 0.0047)
L size[m] (w/o blade)	0.195×0.208
S size[m] (w/o blade)	0.153×0.152
max size[m] (with blade)	0.235×0.235

Comment 4

Regarding the pulley-cable mechanism and the springs, you should clarify why the configuration of the gripper functionality is open by default instead of closed.

Response: We sincerely appreciate your comments. There are several reasons for selecting the open configuration as the default. First, from a functional perspective, tasks such as grasping and perching are executed through deformation and contraction, and the open state allows the aerial robot to perform these tasks more rapidly. Second, maintaining the closed configuration requires the servo motor to continuously output shrinking torque, which increases energy consumption and reduces sustained operational time. Finally, from a structural standpoint, while the closed state minimizes the robot's size and maximizes passable space, the propellers are located beneath the robot. In this configuration, the airflow is partially obstructed, leading to approximately 10% higher hovering energy consumption compared to the open state, thereby negatively impacting flight endurance.

Considering these trade-offs, we have selected the open configuration as the default.

Revise**Bio-inspired mechanism design**

...

With the proposed integrated design, HI-ARM features a compact size with a hand-like profile, with a total weight of just 556g. This allows it to possess more space for flight and grasping operations in narrow indoor environments, a task that can be challenging for larger aerial robots. Additionally, the robot can deform to reduce its dimensions (Fig. 3b(ii)), improving passability in extremely narrow spaces, as validated in Section *Applications in the wild*. HI-ARM defaults to the open configuration, enabling rapid execution of grasping tasks. By contrast, the closed configuration imposes continuous torque demands on the servo motor and partially obstructs propeller airflow, elevating power consumption and compromising flight endurance.

Comment 5

I also have some doubts about the real payload capacity. It is stated that the weight of the platform is 0.55 kg, providing around 0.4 kg payload, whereas the four propellers achieve 1 kg thrust. It should

be specified what is the throttle percentage for 1 kg. For example, if 1 kg thrust is achieved at 80% throttle, this is very close to actuators saturation and probably cannot be maintained more than 2 min or less. **The LiPo battery is 2x 550 mAh 3S, that, given the model of propellers, probably lasts for less than 5 min flight.**

Response: We sincerely appreciate your questions.

The proposed platform has a total thrust of approximately 1 kg. As you noted, the 1 kg thrust is achieved at a throttle close to 0.8, representing the maximum payload capacity that cannot be sustained for extended periods, and we have supplemented payload experiments to validate this specification. As shown in Fig. R2, the proposed aerial robot is capable of carrying a payload of approximately 450 g, corresponding to a thrust of about 0.81. Additionally, we have included flight endurance measurements in *Supplementary Information*: the hover time without payload is approximately 237 s, providing a clearer and more complete presentation of the platform's capabilities.

Revise

Flight performance

This section evaluates the flight performance of the proposed aerial robot, with a focus on payload capacity and endurance.

Payload capacity: The aerial robot is equipped with 3.5-inch propellers, each driven by a T-motor F1404 2900KV motor, providing a maximum total thrust of approximately 1000 g. With a structural mass of 556 g, the theoretical payload capacity is around 450 g. We conduct experimental validation by testing with a water bottle weighing approximately 450 g, as shown in Fig. R2. The robot successfully grasps the object and maintains stable hovering. During normal hovering, the normalized thrust is about 0.38, while after grasping the 450 g payload, the normalized thrust increases to approximately 0.81. These results confirm that the system remains capable of reliable flight even near its maximum load, consistent with the thrust capacity exceeding 1000 g.

Endurance: In the open configuration, the aerial robot achieves a stable flight endurance of about 237 s, sufficient for performing representative tasks such as grasping, perching, door opening, and human-robot interaction. In the contracted configuration, airflow blockage causes an estimated power loss of about 10%, reducing the endurance to approximately 212 s.

Fig. R2. Flight performance of payload.

Comment 6

Originality and Significance

The proposed concept is novel. The methods implemented are well chosen, but no novelty is introduced. L1 is a good adaptive controller. The planning scheme is logical, but not completely original. The application and demonstration is outstanding.

Response: We appreciate your recognition of the novelty and quality of our aerial manipulation concept, which encourages us to further explore new possibilities and practical applications. Our objective is to design a compact and multifunctional aerial robot capable of performing a broad spectrum of tasks. To this end, we integrate L1 and INDI control methods for disturbance compensation, thereby ensuring robust performance across different configurations and payload conditions. Although these control strategies are not entirely novel, their integration within our deformable aerial manipulation platform enables reliable execution of complex tasks in constrained environments. As demonstrated by the controller ablation studies presented in *Supplementary Information*, the combined Baseline + L1 + INDI control configuration consistently achieves the best performance across all tested scenarios, which is essential for accurate trajectory tracking and aerial manipulation. The details of the controller ablation experiments are presented as follows.

Revise

Controller ablation

This section evaluates the ablation experiment of the proposed adaptive control. We adopt a geometric controller as the baseline, which serves as the conventional control scheme for quadrotors. To compensate for disturbances arising from structural deformation, manipulation, and other uncertainties, we integrate an L1 adaptive controller for real-time estimation and rejection of external forces, together with an Incremental Nonlinear Dynamic Inversion (INDI) controller for compensating external torque disturbances. To rigorously assess the effectiveness of the L1 and INDI controllers, we

conduct a series of ablation experiments.

Specifically, four control modes are implemented: baseline (geometric control only), baseline+L1, baseline+INDI, and baseline+L1+INDI. Under each mode, the robot is tested across three representative scenarios: (1) tracking a figure-eight trajectory in the open configuration, (2) tracking while undergoing structural deformation, and (3) tracking with an additional payload of about 200 g. The corresponding experimental results are presented in Fig. R3.

The results demonstrate that the integration of L1 and INDI controllers significantly enhances tracking accuracy and system stability across all scenarios, with particularly notable improvements under deformation and external disturbance conditions. Across all scenarios, the baseline controller consistently yields the poorest performance. The baseline+L1 configuration effectively compensates for errors in COG estimation and uncertain external forces, incorporating these corrections into the position control loop. As a result, the position-tracking error is substantially reduced, as shown in Fig. R3. This improvement is particularly pronounced in the payload-tracking experiment, where the dominant external disturbance arose from the gravitational force of the attached object.

In contrast, the baseline+INDI configuration primarily addresses external torque disturbances. During structural deformation, inaccuracies in inertia tensor estimation lead to attitude control errors, which in turn propagate to position tracking due to the inherent coupling between attitude and position dynamics in quadrotors. By compensating for these torque-induced errors, INDI significantly improves attitude stability and overall tracking precision.

Finally, the combined baseline+L1+INDI configuration achieved the best performance across all tested scenarios. By simultaneously providing position and attitude compensation, this multi-layer adaptive strategy delivers robust rejection of external disturbances and enables accurate and stable trajectory tracking in open, deforming, and payload-bearing configurations.

Fig. R3. Controller ablation. Four control modes: baseline, baseline+L1, baseline+INDI, and baseline+L1+INDI. Under each mode, the robot is tested across three representative scenarios: (1) tracking a figure-eight trajectory in the open configuration, (2) tracking while undergoing structural deformation, and (3) tracking with an additional payload of about 200 g.

Comment 7

Data & methodolog

Authors employ the expression millisecond or microsecond to remark the time performance of the methods, **but proper numerical results should be provided for the selected computer board.**

Response: Thank you for your suggestions. We have added the results of motion planning and deformation planning times in *Supplementary Information*. As shown in Figure R4, we test the planning time for tasks such as grasping, perching, and opening doors on the airborne platform. Each task is repeated several times, and the results are summarized as follows. For motion trajectory optimization, both the maximum and average computation times across different tasks are approximately 1 ms, indicating millisecond-level planning efficiency. For deformation planning, the corresponding maximum and average computation times are approximately 0.009 ms, demonstrating microsecond-level optimization performance.

Revise

Planning performance

This section evaluates the performance of the proposed task planner. As described earlier, the aerial robot decomposes task planning into two components: flight trajectory planning and deformation planning. Trajectory optimization is achieved within the millisecond scale, while deformation planning is computed at the microsecond scale. To validate this capability, we conduct real-world experiments involving four representative tasks: palm grasping, fingertip pinching, door opening, and perching. The onboard computation is performed using a Radxa ZERO board^a, and the results are summarized in Fig. R4.

Across nearly all tasks, the trajectory optimization time remained around 1 ms. Longer trajectories naturally require slightly more computation time—for example, the palm grasping of a water bottle takes marginally longer than the fingertip pinching of a tissue. The door-opening task results in the longest computation time, approximately 1.652 ms, due to the need for multiple interpolations along the curved trajectory. In contrast, deformation planning involves solving a closed-form mapping between the desired deformation and the servo motor's angle, yielding highly consistent computation times across tasks, at about 0.009 ms. These results demonstrate that the aerial robot achieves millisecond-level trajectory optimization and microsecond-level deformation planning, thereby enabling real-time execution of complex tasks with high efficiency.

Fig. R4. The planning time for flight and morphing on the airborne platform during tasks such as palm grasp, fingertip pinch, perching, and door opening.

⁴Radxa ZERO board specifications: <https://radxa.com/products/zeros/zero/>

Comment 8

For perching on poles, it requires active energy insertion. **Which is the most suitable configuration by default: open or close?** Close by default would be more convenient for saving energy, but this would require changing the spring configuration (from compression to extension).

Response: We sincerely appreciate your comments. In the perching scenario, a default closed design is indeed more efficient, as it eliminates the need for the servo motor to expend extra energy during perching. This seems to be a reasonable design choice. However, if the default state is closed, the actuator needs to provide an expanding force to open the robot, which can be infeasible for tendon-driven systems. One potential solution is to add extra actuators to each deformable joint to provide the expansion force, but this significantly increases the robot's mass and size.

Additionally, when considering other scenarios such as grasping, pinching, and interaction, we find that the robot's operations rely on the closing-from-opening actions. As mentioned earlier, its default open state allows for more convenient task completion. Additionally, maintaining a closed state requires the servo motor to continuously output shrinking torque, consuming more energy. Therefore, a default open state is the better choice.

Revise

Bio-inspired mechanism design

...

With the proposed integrated design, HI-ARM features a compact size with a hand-like profile, with a total weight of just 556g. This allows it to possess more space for flight and grasping operations in narrow indoor environments, a task that can be challenging for larger aerial robots. Additionally, the robot can deform to reduce its dimensions (Fig. 3b(ii)), improving passability in extremely narrow spaces, as validated in Section *Applications in the wild*. **HI-ARM defaults to the open configuration, enabling rapid execution of grasping tasks. By contrast, the closed configuration**

imposes continuous torque demands on the servo motor and partially obstructs propeller airflow, elevating power consumption and compromising flight endurance.

Comment 9

In Figure 4-b: External estimation is not very good if force is estimated around 2.2 N while the bottle is 1.5 N. Provide thrust or PWM values from flight controller to validate the test.

Response: We sincerely appreciate your suggestions. The weight of the bottle is approximately 154 g, corresponding to a gravitational force of around 1.51 N. However, Figure 4b shows the estimated force to be about 2.20 N. This is because the estimated external force is **the total external disturbance force**. As mentioned in the *Multi-level adaptive control* section, the total external disturbance force mainly comes from the gravitational force of the load, but it is also affected by deformation disturbances, intake flow obstruction disturbances, propeller power loss, and other factors. Accordingly, we have incorporated the actual thrust variation curve corrected for power loss, as illustrated in Fig. R5, where the additional thrust generated after grasping closely matches the bottle's gravity. In addition, we have included the motor speed curves to further verify the thrust variation following object grasping.

Revise

Palm grasping

Humans typically grasp larger objects (e.g., water bottles or oranges) using their palms, HI-ARM exhibits a similar ability. We conduct an autonomous grasping experiment, where HI-ARM is asked to grip a water bottle (diameter: 62 mm, mass: 153 g) before transporting it to a destination (Fig. R5a). The position tracking curves shown in Figure R5b indicate that HI-ARM can successfully track a reference trajectory with relative accuracy at a maximum velocity of 1.1 m s^{-1} . Our proposed controller can accurately estimate external force and torque disturbances in real time (Fig. R5b) and effectively compensate for them. **As shown in Fig. R5b, after correcting for thrust loss, the robot's actual total thrust increases by about 1.5 N upon grasping, closely matching the object's gravitational force.** During gripping, once the set torque has been reached, the grasping action is considered to be completed, and the servo motor stops rotating.

Fig. R5. Hand-inspired grasping. (a) HI-ARM palm grasps, transports, and releases a water bottle (image mirrored horizontally for better readability). (b) State curves of HI-ARM during the water bottle grasping process, including position tracking, velocity tracking, estimated external forces, and estimated external torques. State curves of HI-ARM during the water bottle grasping process, including position tracking, velocity tracking, estimated external forces, estimated external torques, **motor speed, and actual thrust**. (c) Adaptive grasping of objects with various shapes by HI-ARM. (d) A sequence of HI-ARM performing fingertip grasping on a napkin. (e) Fingertip grasping of items with various shapes.

Comment 10

Fully autonomy should be properly demonstrated, particularly regarding the on-board position estimation, both indoors/outdoors. This should be clarified in the manuscript.

Response: Thank you very much for your suggestion. As mentioned earlier, we have provided a more specific and rigorous description of the indoor and outdoor localization schemes in the manuscript.

Indoors: for tasks such as grasping, perching, opening doors, and interacting with people, we use an external optical tracking system (such as motion capture systems) for both the robot's state estimation and the object's position. *Outdoors:* we use the Intel RealSense T261 module as the localization solution, which can output results suitable for tasks like perching, river-crossing transport, and traversal, as demonstrated in experimental validations. These descriptions have been incorporated into the revised manuscript.

Furthermore, to avoid potential misunderstandings for localization, we have also revised the description of "full autonomy" or "fully autonomous operation" in the main text, as shown below.

Introduction

...

To accomplish complicated tasks with precision and smoothness, autonomy is also crucial for HI-ARM. To this end, we propose a framework consisting of a task planner, trajectory generation, state feedback, **parameter estimation**, and adaptive control (Fig. 2b). HI-ARM supports two working modes: (1) **fully** autonomous operation, where the task planner selects proper operating sequences from an action library according to input task types **and the robot executes them under closed-loop perception, planning, and control** (e.g., grasping, perching, door opening, and human interaction), and (2) human-robot collaboration, where the task planner generates reference trajectories and grasping commands from human intentions, **and the controller subsequently tracks these commands**.

...

Results

Flight design and electronic components

...

For localization and estimation, we aim to balance accuracy, weight, and onboard computational load. **In indoor experiments, an external optical motion-capture system provides sub-millimeter state estimation of both the aerial robot and the manipulated objects, enabling closed-loop integration with mission planning and adaptive control for autonomous operation. In outdoor experiments, we employ the Intel RealSense T261 tracking module, which weighs 26 g and offers a lightweight alternative to LiDAR-based localization systems while maintaining stable short-range state estimation (for example, within 20 m) suitable for experimental validation.**

...

Mission planning

To perform diverse aerial tasks, HI-ARM introduces an efficient autonomous framework that supports both **fully** autonomous operation and human-robot collaboration. As shown in Figure 2b, we establish an action library for common tasks such as grasping, perching, door opening, and human interaction. For autonomous operation, HI-ARM dynamically assembles basic motions from the library based on the task type to generate desired operation sequences, as validated in Section *Interaction with humans*. For human-robot collaboration, HI-ARM generates appropriate commands for flight and grasping based on the operator's intent. These commands are subsequently sent to the mission planner.

...

Comment 11

Fig. 5-e: why power is **not plotted in Watts**?

Response: We greatly appreciate your insightful comments. The normalized power curves are used primarily to highlight the ratio between the power required for perching at the pole and hovering in the air, as normalization allows for a more direct comparison. To enhance clarity, we have included the hovering

power in the main text, as shown below.

Revise

Perching

Similar to the way humans use handrails on trains or grasp tree trunks (Fig. R16a), HI-ARM can utilize its specialized structure to perch on fixed objects. To evaluate the perching capability, we set up a tree trunk as the test object. In this experiment, the flying robot autonomously flies toward the tree and accurately grasps and perches on it (Fig. R16b). As shown in Figure R16c, all motors cease rotation, the system's gravity is counterbalanced by the friction generated from the HI-ARM's grip on the tree trunk, and the energy consumption of the system is significantly reduced (the servo motor consumes two orders of magnitude less energy than the rotor-propellers, as shown in Figure R16e). **By contrast, conventional hovering consumes over 160 W.** After a while, the drone receives a release command and the propellers begin to rotate again. The servo motor drives the vehicle to subsequently expand, gradually disengaging from the tree, as HI-ARM re-enters flight mode (Fig. R16d). This unique perch-and-relaunch mechanism allows for a break between consecutive missions and supports a long-stay mission with low energy consumption.

Comment 12

Fig. 5-i: The force figure **seems a pattern typical of manual flight** rather than an automatic flight control.

Response: We greatly appreciate your comments. The door-opening experiments employ an autonomous algorithm framework. Once the robot acquires the position and radius of the door handle, it can autonomously generate the door-opening trajectory, which is subsequently executed by an adaptive controller.

However, manual control of the robot for door opening is impractical. To verify this, we conduct human-operated experiments, as shown in Fig. R6a. First, it is extremely challenging for a human operator to precisely align and grasp the door handle, as the task requires very fine manipulation. Second, interactions between the flying robot and the environment generate substantial external disturbances, producing unpredictable effects on the robot's flight state. For a pilot without direct force feedback, rapidly adjusting the robot's posture to maintain real-time interaction with the environment is extremely challenging. Even small deviations in interaction forces can easily lead to a crash.

Fig. R6. Door opening test. (a) Manual flight for door opening. (b) Using no door handle for door opening. (c) Max force test for opening the door.

Comment 13

Fig. 6-i: These payloads are very low compared to the 0.45 kg payload claimed before.

Response: We sincerely appreciate your insightful feedback. In our previous experiments, we primarily demonstrated the robot's capability to perform various manipulation tasks, but overlooked the evaluation of its payload capacity. To address this, we have added grasping tests with heavier loads, such as objects weighing 450 g, as shown in Fig. R7. These experiments verify that the proposed robot can grasp a maximum payload of up to 450 g. We have also included the payload test in *Supplementary Information*.

Revise

Flight performance

This section evaluates the flight performance of the proposed aerial robot, with a focus on payload capacity and endurance.

Payload capacity: The aerial robot is equipped with 3.5-inch propellers, each driven by a T-motor F1404 2900KV motor, providing a maximum total thrust of approximately 1000 g. With a structural mass of 556 g, the theoretical payload capacity is around 450 g. We conduct experimental validation by testing with a water bottle weighing approximately 450 g, as shown in Fig. R7. The robot successfully grasps the object and maintains stable hovering. During normal hovering, the normalized thrust is about 0.38, while after grasping the 450 g payload, the normalized thrust increases to approximately 0.81. These results confirm that the system remains capable of reliable flight even near its maximum load, consistent with the thrust capacity exceeding 1000 g.

...

Fig. R7. Flight performance of payload.

Comment 14

Conclusions

The experimental validation regarding the fully autonomous operation is not completely convincing. It seems in some cases external positioning systems are employed, or manual flight. This is important, since the title and scope of the paper is also autonomy, and the platform size/weight is determined by the weight of the on-board components. Having a small size of platform is essential for safe interaction with humans, involving weight reduction. In this sense, on-board positioning is particularly challenging.

Response: We sincerely appreciate your comments and suggestions. As you correctly noted, achieving precise localization on such a compact platform is challenging, as it requires high-precision sensing and substantial computational resources, both of which increase the robot's weight. To enable small-scale yet accurate aerial operations, we make trade-offs in our localization strategy: for indoor experiments, we use an external optical tracking system (such as motion capture systems) for both the robot's state estimation and the object's position; while for outdoor experiments, we employ the Intel RealSense T261 module as the localization solution, which provides acceptable accuracy. These localization configurations, together with adaptive control and motion planning, have enabled the successful execution of both indoor and outdoor

experiments.

To avoid potential misunderstandings, we have revised the description of “full autonomy” or “fully autonomous operation” in the main text, as shown below. As you pointed out, fully onboard perception and localization is our ultimate goal. In future work, we plan to achieve this by leveraging the latest technologies, such as end-to-end vision-to-action approaches, which has been added to the *Discussion* section.

Revise

Introduction

...

To accomplish complicated tasks with precision and smoothness, autonomy is also crucial for HI-ARM. To this end, we propose a framework consisting of a task planner, trajectory generation, state feedback, **parameter estimation**, and adaptive control (Fig. 2b). HI-ARM supports two working modes: (1) **fully** autonomous operation, where the task planner selects proper operating sequences from an action library according to input task types **and the robot executes them under closed-loop perception, planning, and control** (e.g., grasping, perching, door opening, and human interaction), and (2) human-robot collaboration, where the task planner generates reference trajectories and grasping commands from human intentions, **and the controller subsequently tracks these commands**.

...

Results

Flight design and electronic components

...

For localization and estimation, we aim to balance accuracy, weight, and onboard computational load. In indoor experiments, an external optical motion-capture system provides sub-millimeter state estimation of both the aerial robot and the manipulated objects, enabling closed-loop integration with mission planning and adaptive control for autonomous operation. In outdoor experiments, we employ the Intel RealSense T261 tracking module, which weighs 26 g and offers a lightweight alternative to LiDAR-based localization systems while maintaining stable short-range state estimation (for example, within 20 m) suitable for experimental validation.

...

Mission planning

To perform diverse aerial tasks, HI-ARM introduces an efficient autonomous framework that supports both **fully** autonomous operation and human-robot collaboration. As shown in Figure 2b, we establish an action library for common tasks such as grasping, perching, door opening, and human interaction. For autonomous operation, HI-ARM dynamically assembles basic motions from the library based on the task type to generate desired operation sequences, as validated in Section *Interaction with humans*. For human-robot collaboration, HI-ARM generates appropriate commands for flight and grasping based on the operator’s intent. These commands are subsequently sent to the mission planner.

...

Discussion

In this study, we develop an innovative hand-inspired compact flying robot for manipulation, which integrates biomimetic grasping and aerial flying capabilities. We propose an efficient autonomous framework that enables the robot to perform precise and smooth aerial manipulation in real-world environments. The proposed robot demonstrates excellent performance across indoor and outdoor tasks, showing considerable potential as future smart home assistants, moving flying cameras, aerial teleoperation tools and flying delivery robots. However, current state estimation depends on external localization, and the onboard visual localization module exhibits cumulative drift during long-range outdoor tasks. Achieving full autonomy requires not only accurate localization but also the ability to interpret the environment from visual inputs. End-to-end visual reinforcement learning approaches [3, 4], which directly map perception to control while adaptively correcting state estimation errors, offer a promising route to enhance system autonomy in the future.

...

Comment 15

Suggested improvements

To my understanding, the nylon cable, which is stiff, is used to close the hand. Therefore, there is no compliant adaptation in the closing procedure. The action of the springs, which provide the compliance, is for opening the gripper. The servo motor will be stalled, and this may damage the gearbox or increase the current consumption excessively.

Response: We sincerely appreciate your suggestions. In practice, we employ specially designed flexible nylon cords, which help mitigate the robot's structural stiffness during deformation. In addition, the servo motor used for morphing is equipped with torque overload protection. When the deformation force reaches the cutoff torque, the motor activates the overload protection, causing the current to decrease and thereby reducing power consumption.

Comment 16

- Clarify which is the thickness of the carbon fiber boards, also provide a more in detail picture of the mechanical construction, showing the assembly of the mechanical elements (linear guides) and the on-board components (battery, computer board, camera). How is the camera connected to the computer board?

Response: We sincerely appreciate your comments. The carbon plate has a thickness of approximately 2 mm, as updated in the main text.

Revise

Flight design and electronic components

... The primary structure of the flight platform is composed of 2 mm thick carbon fiber plates, selected for their high strength and lightweight properties. ...

Following your suggestion, we have added a mechanical schematic that provides a detailed illustration of the mechanical components, battery, onboard computer, camera, and flight controller, as shown in Figure R8a and R8c. The interconnections of signal and power lines among the individual components are also

schematically presented in Figure R8b.

Revise

Experimental platform and parameters

This section presents the experimental platform of the robot, as shown in Figure R8. To provide a clearer view of the internal configuration, the external shell is rendered transparent. As shown in Figure R8c, the layout of key onboard components—including the battery, onboard computer, and flight controller—is explicitly displayed. The Intel RealSense T261 module is mounted on the upper side of the body for visual sensing and localization, as shown in Figure R8a. The interconnections of signal and power lines among the individual components are also schematically presented in Figure R8b.

...

Fig. R8. Hardware components of HI-ARM. (a) Components description. (b) Electronics architecture. (c) 3D structural illustration.

Comment 17

- Aerial "manipulative" sounds strange. Why not "manipulation"?

Response: We appreciate your suggestion; "manipulation" indeed seems more appropriate, and we have revised many parts of the manuscript accordingly.

Introduction

Through evolutionary processes, birds have developed a remarkable ability to use their forelimbs (wings) for aerial locomotion while employing their hindlimbs (talons) for interaction, allowing complex activities such as aerial hunting, grasping, perching, and nest building [5,6]. This biological characteristic has ignited numerous research interests to employ flying machines with **manipulative manipulation** capabilities [7–15], in order to interact with objects and humans in midair. ...

... In response, researchers opt to leverage robots' own structures for aerial manipulation, works [16–23] try to theoretically reduce system complexity by minimizing external attachments. However, these robots struggle with either low accuracy caused by extra actuators, or mechanical complexity because of movable structures, limiting their operating range, precision, and speed. These challenges all underscore the necessity of a novel flying **manipulative manipulation** robot, which simultaneously satisfies compact design, simplified mechanism, wide adaptability, superior agility, stability and passibility, as well as high autonomy.

...

Results

Bio-inspired mechanism design

...

With the proposed integrated design, HI-ARM features a compact structure resembling an open human hand, with a total weight of just 556g. This allows it to possess more space for flight and manipulation in narrow indoor environments, which is challenging for large-sized aerial **manipulative manipulation** robots. Additionally, the robot can deform to reduce its dimensions (Fig. 3b(ii)), improving passibility in extremely narrow spaces, as validated in Section *Applications in the wild*.

Mission planning

... Unlike traditional aerial **manipulative manipulation** robots with multi-DOF robotic arms, HI-ARM's integrated structure reduces the number of actuators, significantly decreasing the number of planning variables. Thanks to this integrated design, we decompose the mission planner into two independent modules: flight trajectory planning and end-effector deformation planning. In terms of trajectory planning, the quadrotor configuration employed by HI-ARM exhibits differential flatness [24], allowing constraint simplification to accelerate solving speed [25]. ...

Comment 18

- The sentence "The servo motor provides torque state feedback, which helps determine the success of the gripping action". The servo employed seems the typical RC PWM servo. **Does it really provides feedback? What is more, if it is continuous rotation, how do you address the problem of multi-turn counting?**

Response: We sincerely appreciate your comments. The Feetech STS3032 motor supports continuous rotation with angle control and provides angle feedback. For example, in the palm grasping experiment with a water bottle, the motor rotates approximately 12 rad (about two full turns), as illustrated by the angle curve

in Fig. R9.

Moreover, the servo motor also provides current feedback, which can be used to obtain a coarse estimation of torque, as shown by the torque curve in Fig. R9. Although the torque cannot be precisely measured, the motor is equipped with a torque protection feedback mechanism that outputs an overload signal, as shown by the overload curve in Fig. R9. By combining these two curves, it is possible to infer whether sufficient contact has been established to successfully grasp the target object.

Fig. R9. Hand-inspired grasping. State curves of HI-ARM during the water bottle grasping process, including position tracking, velocity tracking, estimated external forces, servo motor's angle, servo motor's speed, servo motor's torque, and overload feedback of the servo motor.

Comment 19

- "non-negligible damping", I think friction is more suitable.

Response: Thank you for your suggestion. "Friction" appears more appropriate, and we have updated the corresponding results in the main text.

Revise

Fingertip grasping

Other than palm grasping, fingertip grasping is also a normal operation for human hands, especially for taking small objects. In this experiment, HI-ARM is commanded to grasp some napkins. As shown in Figure 4d, napkins are extremely thin (less than 1 mm), soft, and light (less than 1 g), thus pose unique challenges for accurate automated grasping. Figure 4d demonstrates that HI-ARM grasps a single napkin with fingertip precision. During tissue extraction, HI-ARM encounters disturbances caused by ~~non-negligible damping~~ friction, which are effectively compensated by the controller, resulting in a control error of less than 3 cm.

Comment 20

- In Human-robot collaborative operation (pg. 18), which control reference is provided by the user, velocity? How can a 3D reference be provided with a single input device? Drone pilots need two sticks.

Response: We sincerely appreciate your comments. The control input of teleoperation is the velocity command. Similar to DJI's Avatar FPV drone³, a single-hand controller can be used to control three-dimensional flight. In our system, the robot employs a motion-based 3D controller, which maps hand movements to velocity commands in different directions. The controller also includes a grasp button, allowing both flight and grasping actions to be performed with one hand.

Revise

Human-robot collaborative operation

As a bio-inspired aerial interaction device, HI-ARM can function as a third flying hand for humans, responding to human intentions. As shown in Figure 8a, HI-ARM is equipped with a remote video transmission system that provides first-person-view (FPV) visual feedback. ~~Using a simplified, single-handed remote controller, the user easily sends flight and grasping commands to HI-ARM.~~ Similar to DJI's Avatar FPV drone, HI-ARM can be operated with a simplified, single-handed motion-based 3D controller, which maps hand movements to velocity commands in different directions. The controller also integrates a grasp button, allowing both flight and grasping actions to be performed with one hand. As shown in Fig. 8a, the communication between the controller and the robot is established via the ROS framework. As described in Section *Mission planning*, these commands are then processed by the mission planner to generate flight and deformation inputs for the controller, enabling the robot to execute aerial tasks. The following two experiments demonstrate HI-ARM's potential in human-assisted remote operations (see Supplementary Movie 4).

...

³DJI's Avatar FPV drone specifications: <https://www.dji.com/cn/avata-2>

Comment 21

- Regarding the dynamics, **which are the transformations applied to account for the displacement of the IMU with respect to the center of the platform?**

Response: We thank you for your comments. The IMU and external raw localization module are placed within the same module, and the robot's motion state is estimated using an Extended Kalman Filter (EKF). Based on the servo motor's deformation angle, we can get the deformation of different modules (as described in *Supplementary Information*), and then estimate the relationship between each module's motion and the COG, allowing for real-time COG estimation.

Revise

Methods

Dynamics

In this study, we employ bold lowercase letters to denote vectors (e.g., \mathbf{v}) and bold uppercase letters for matrices (e.g., \mathbf{J}). Scalars are represented otherwise. As shown in Figure 2c(iii), we utilize a world frame \mathbf{W} with an orthonormal basis $\{\mathbf{x}_W, \mathbf{y}_W, \mathbf{z}_W\}$, where \mathbf{z}_W is oriented upward, counter to the direction of gravity. The body frame \mathbf{B} , situated at the geometric center of the proposed vehicle, is defined with an orthonormal basis $\{\mathbf{x}_B, \mathbf{y}_B, \mathbf{z}_B\}$. Let the position of the vehicle in the world frame be represented by $\mathbf{p}_W = (p_x, p_y, p_z)^\top$, its orientation by the quaternion $\mathbf{q}_W = (q_x, q_w, q_y, q_z)^\top$, and its linear velocity by $\mathbf{v}_W = (v_x, v_y, v_z)^\top$. Additionally, the angular velocity of the vehicle, expressed in the body frame, is given by $\boldsymbol{\omega}_B = (\omega_x, \omega_y, \omega_z)^\top$.

HI-ARM flight system is composed of four parallel rotor-propellers, whose position distribution approximates the 'X' configuration of a conventional quadrotor. Consequently, its flight dynamics model is similar to that of traditional quadrotors, and the device possesses the differential flatness characteristics [24] as general quadrotors. We utilize 6 DoFs to describe the ideal rigid-body kinematics and dynamics model of the robot. For the translational dynamics component, the following equations are utilized:

$$\begin{aligned}\dot{\mathbf{p}}_W &= \mathbf{v}_W, \\ \dot{\mathbf{v}}_W &= T\mathbf{z}_B/m + \mathbf{g},\end{aligned}\tag{R1}$$

where T and m are the collective thrust and total mass respectively; \mathbf{z}_B is the Z axis of the body frame expressed in the world frame; $\mathbf{g} = [0, 0, -g]^\top$ is the gravitational vector.

The rotational kinematic and dynamic equations are expressed as

$$\begin{aligned}\dot{\mathbf{q}}_W &= \frac{1}{2} \left(\begin{bmatrix} 0 \\ \boldsymbol{\omega}_B \end{bmatrix} \right)_{\times} \cdot \mathbf{q}_W, \\ \dot{\boldsymbol{\omega}}_B &= \mathbf{J}^{-1}(\boldsymbol{\tau} - \boldsymbol{\omega}_B \times \mathbf{J}\boldsymbol{\omega}_B),\end{aligned}\tag{R2}$$

where $[\cdot]_{\times}$ is the skew-symmetric matrix; $\boldsymbol{\tau}$ and \mathbf{J} are the total torque and inertia tensor matrix, respectively, **both of which are dynamically updated during morphing and grasping, with more details provided in *Supplementary Information*.**

Let k_t and k_c denote the thrust coefficient and torque coefficient, respectively, associated with the j -th motor. The rotational speed of the j -th motor is represented by Ω_j , and its position within the body frame is given by $\mathbf{l}_j = [l_{x_j}, l_{y_j}, l_{z_j}]^\top$. The collective thrust T and the torque $\boldsymbol{\tau}$ generated

by the actuators are expressed as follows:

$$\begin{bmatrix} T \\ \boldsymbol{\tau} \end{bmatrix} = \mathbf{H}_k \mathbf{t}, \quad (\text{R3})$$

where $\mathbf{t} = [k_t \Omega_1^2, k_t \Omega_2^2, k_t \Omega_3^2, k_t \Omega_4^2]^\top$ represents the thrust generated by each rotor, \mathbf{H}_k is the time-variant mixed control matrix (MCM) while the vehicle morphing. Assuming that the center of gravity is $\mathbf{r}_{COG} = (r_x, r_y, r_z)^\top$ at this time, then \mathbf{H}_k is as follows:

$$\mathbf{H}_k = \begin{bmatrix} 1 & 1 & 1 & 1 \\ r_y + l_{y1} & r_y + l_{y2} & r_y + l_{y3} & r_y + l_{y4} \\ r_x - l_{x1} & r_x - l_{x2} & r_x - l_{x3} & r_x - l_{x4} \\ -k_c/k_t & k_c/k_t & -k_c/k_t & k_c/k_t \end{bmatrix}. \quad (\text{R4})$$

The module positions \mathbf{r}_{mod_i} and propeller coordinates (l_{x_i}, l_{y_i}) are not fixed; instead, they are dynamically updated during morphing and flight. Specifically, these parameters are derived from the tendon-driven morphing kinematics: the servo motor angle θ_a and rope displacement L_t determine the deformation of both the telescopic and torsional mechanisms, which in turn define the instantaneous propeller positions. As illustrated in Figure 2c, L'_i and L_i denote the lengths of the telescopic mechanism during motion and under nominal conditions, respectively, with $L'_i - L_i$ representing the compression displacement and reflecting variations in the distances between adjacent modules. As shown in Figure 2c(ii), θ_j denotes the rotation angle of the j -th torsional mechanism (comprising a torsion spring and a circumferential bearing). Accordingly, the propeller coordinates (l_{x_i}, l_{y_i}) are updated to (l'_{x_i}, l'_{y_i}) as follows:

$$\begin{aligned} l'_{x1} &= l_{x1} - (L'_i - L_i)/2 - l_0(1 - \cos\theta_1) \\ l'_{x2} &= l_{x2} + (L'_i - L_i)/2 \\ l'_{x3} &= l_{x3} + (L'_i - L_i)/2 \\ l'_{x4} &= l_{x4} - (L'_i - L_i)/2 - l_0(1 - \cos\theta_2) \\ l'_{y1} &= l_{y1} + (L'_i - L_i)/2 + l_0(1 - \sin\theta_1) \\ l'_{y2} &= l_{y2} + (L'_i - L_i)/2 \\ l'_{y3} &= l_{y3} - (L'_i - L_i)/2 \\ l'_{y4} &= l_{y4} - (L'_i - L_i)/2 - l_0(1 - \sin\theta_2) \end{aligned} \quad (\text{R5})$$

As a consequence, the control matrix \mathbf{H}_k is updated to \mathbf{H}'_k online according to the current morphing state. This formulation explicitly incorporates geometry changes induced by morphing into the control layer, rather than treating them as static.

...

Comment 22

- Since there is a single actuator for controlling the deflection of three linear segments and two torsion springs (subactuation), it should be established an order in the passive actuation, **which is determined by the relative value of stiffness**. For example, if it is desired that the finger joints rotate first, their stiffness should be lower compared to the ones of the linear guides.

Response: We greatly appreciate your suggestions. The desired deformation sequence is first extension,

followed by rotational grasping. Therefore, the stiffness of the compression spring is smaller than that of the torsion spring. We have added this clarification to the manuscript.

Revise

Bio-inspired mechanism design

... Human grasping relies on joint angle variations to achieve contraction (Fig. 3a(iii)). Inspired by this principle, HI-ARM's finger modules incorporate a torsion structure comprising torsion springs and circular bearings (Fig. 3b(i)) to enable flexible finger bending. Given their ability to displace significantly in a short time, telescopic structures are incorporated into the deformable mechanism to increase the speed of grasping movements. **To ensure that the telescopic modules are activated first, their spring stiffness is deliberately designed to be lower than that of the rotational modules, thereby enabling rapid contraction and closer contact with the target object.** As illustrated in Figure 3e, the springs in the mechanism absorb energy during compression and torsion, and then release their stored energy to restore the shape of the robot, thereby reducing overall energy consumption. The integration of telescopic and torsional mechanisms forms a hybrid 5-DoF structure, allowing a variety of adaptive and flexible grasping, as shown in Figure 3f. ...

Comment 23

References

The state of the art section includes relevant references, but it is not up to date. See for example the survey "Past, present and future of aerial robotic manipulators" published in T-RO. Other related references: "Physical human-aerial robot interaction and collaboration: Exploratory results and lessons learned", "Through-Window Home Aerial Delivery System with In-Flight Parcel Load and Handover: Design and Validation in Indoor Scenario", "Autonomous drone delivery to your door and yard".

Response: We greatly appreciate your valuable literature recommendations, which have made the paper more comprehensive and persuasive. The aforementioned references have been incorporated into the revised manuscript.

Revise

Introduction

Through evolutionary processes, birds have developed a remarkable ability to use their forelimbs (wings) for aerial locomotion while employing their hindlimbs (talons) for interaction, allowing complex activities such as aerial hunting, grasping, perching, and nest building [5,6]. This biological characteristic has ignited numerous research interests to employ flying machines with **manipulative manipulation** capabilities ~~[7–15]~~ [7–15,26–29], to interact with objects and humans in midair. Flying robots, as the most maneuverable robots, are highly anticipated to deeply participate in our social activities, especially safety-critical ones.

...

Comment 24

Clarity and context

There are some statements at the Introduction that require revision. For example, birds also use their beak to perform manipulation tasks, becoming quite skilled (like the parrots). The sentence "Flying robots [...] are highly anticipated to deeply participate in our social activities, especially safety-critical ones" is not clear to me. Also "replacing humans" is an statement that should be used carefully, since human workers do not like to be replaced. In general, the introduction should be revised and improved, it is not properly written. Sentences as the first one on page 3: "making them unsuitable for sophisticated operations" are controversial.

Response: We sincerely appreciate your feedback. The above statements have been revised, and the corresponding modifications have been made in the main text.

Revise

Introduction

Through evolutionary processes, birds have developed a remarkable ability to use their forelimbs (wings) for aerial locomotion while employing their hindlimbs (talons) for interaction, allowing complex activities such as aerial hunting, grasping, perching, and nest building [5,6]. This biological characteristic has ignited numerous research interests to employ flying machines with ~~manipulative~~ **manipulation** capabilities ~~[7-15]~~ [7-15,26-29], in order to interact with objects and humans in midair. Flying robots, as the most maneuverable robots, are highly anticipated to deeply participate in our social activities, especially safety-critical ones **like earthquake rescue, high-altitude maintenance, and material transportation**. In the past decade, flying robots are widely used in applications related to information acquisition, such as geographic surveying, aerial photography/videography, inspection and monitoring. However, the emerging demand for close-proximity operation necessitates new-generation flying robots that can ~~replace~~ **help** humans not only for observation, but also for manipulation. For instance, in hazardous environments such as nuclear power plants or chemical facilities, close-range interactions including valve turning and button pushing are quite common. In search-and-rescue missions, quick catch and release are vital for supply delivery or collaborative transportation. In daily life, item distribution across the air, goods retrieval from human-unreachable areas, or even touch-range extending for the disabled, often occur in our imagined future house or factory. These cross-domain applications highlight the vast potential of aerial manipulation, motivating aerial robots from flying eyes to flying hands.

Existing research on aerial manipulation has made significant progress, while some fundamental limits restrict their further applicability and extensibility in real-world scenarios. Early research in this area primarily focuses on directly mounting robotic arms to drones [30-36], but their large size, heavy weight, and high energy consumption severely hurt their maneuverability and endurance, making them unsuitable for ~~sophisticated~~ **delicate or long-duration** operations, especially in confined situations such as human-involved activities. Subsequent efforts seek to address these issues by optimizing end-effector designs, including simplifying actuators [37-50], developing novel drive mechanisms [51-54], and introducing soft grasping components [55-57]. Although these works shine in structural innovations, they introduce unavoidable control coupling problems, resulting in compromises in agility and stability of the robot. In response, researchers opt to leverage robots' own structures for aerial manipulation, works [16-23] try to theoretically reduce system complexity by minimizing external attachments. However, these robots struggle with either low accuracy

caused by extra actuators, or mechanical complexity because of movable structures, limiting their operating range, precision, and speed. These challenges all underscore the necessity of a novel flying ~~manipulative~~ **manipulation** robot, which simultaneously satisfies compact design, simplified mechanism, wide adaptability, superior agility, stability and passibility, as well as high autonomy.

...

Responses to Comments by Reviewer #2

Comment 25

General Assessment

The authors present the design, modeling, control, and experimental validation of a compact aerial robot that integrates flight and manipulation, enabled by a novel morphing mechanism. The system is supported by an autonomous control framework and showcases various demonstrations ranging from object grasping to human-robot interaction. The breadth of experimental results and the robot's operational versatility are highly impressive, and they convincingly demonstrate the potential utility of the proposed platform. However, **the repeated emphasis on the "hand-inspired" or "bio-inspired" nature of the robot feels somewhat overstated.** While the robot features an interesting grasping mechanism, its similarity to the biological hand remains limited, both structurally and functionally.

Major Comments

Overstatement of "hand-inspired" design: The proposed mechanism is certainly innovative, but the claim that it is "hand-inspired" may be an overreach. From a mechanical perspective, the device offers only basic grasping functionality and, at most, resembles a gripper rather than a biologically inspired hand. Thus, referring to it as "hand-inspired" seems exaggerated. Additionally, the linear actuated motion, in particular, is not biologically inspired and should instead be recognized as a novel mechanical design in its own right. **Moreover, the drone is restricted to xyz + yaw (i.e., SE(2)+R) motions and can only grasp vertically elongated objects. Given this limitation, the claim of generalized aerial manipulation is arguably overstated.**

Response: We sincerely appreciate your thoughtful comments and constructive suggestions. We fully acknowledge that certain descriptions in the original manuscript may have potentially overstated the "hand-inspired" nature of our design. While our system performs hand-like operations such as palm grasping and fingertip pinching, it does not yet achieve fully dexterous hand manipulation comparable to that of a biological hand. Accordingly, we have revised the relevant statements in the *Introduction* and *Hand-like mechanism design* sections to clearly delineate the structural and functional limitations of the current system.

Mechanically, the proposed design indeed draws inspiration from the structural configuration and actuation principles of the human hand—particularly the coordination between the thumb and index finger—while simplifying the complexity into a compact, tendon-driven morphing mechanism. This bio-inspired simplification enables efficient palm and fingertip grasping within a lightweight aerial platform. Our use of the term "hand-inspired" is intended to convey imitation of certain biological and functional characteristics rather than a direct replication of the human hand's full dexterity, structure, or appearance. However, we agree that the resemblance remains limited; to avoid potential ambiguity, we have adjusted the terminology throughout the manuscript from "hand-inspired" to "hand-like" to better reflect the actual level of biomimicry.

In addition, we recognize that the robot's motion is constrained by its quadrotor configuration, which provides translational and yaw rotational capabilities in $\mathbb{R}^3 \times S^1$. These inherent kinematic limitations restrict the manipulation workspace and the range of graspable objects. To avoid potential misunderstanding, the original manuscript title, "**Hand-Inspired Autonomous Flying Robot for Airborne Manipulation,**" has been revised to "**Hand-like Autonomous Flying Robot for Airborne Grasping and Interaction,**" and all related descriptions have been updated accordingly.

Finally, we emphasize that developing a more dexterous, biologically inspired aerial manipulation system remains one of our long-term goals. In future work, we plan to incorporate additional actuation mechanisms and learning-based control strategies to further enhance the dexterity and adaptability of the proposed platform.

Title:~~**Hand-Inspired Autonomous Flying Robot for Airborne Manipulation**~~**Hand-like Autonomous Flying Robot for Airborne Grasping and Interaction****Introduction**

...

As the call for the appearance of such an ideal flying operational robot, we dive into the fundamentals of nature. Human hands exhibit dexterous interaction movements (Fig. 1a), efficiently adapting to complex environments and performing a wide range of tasks. For instance, humans grasp large objects like cups or doorknobs using the palm (Fig. 1c), and delicately pinch smaller objects such as paper or pills with fingertips (Fig. 1d). Research [58, 59] shows that the bones, joints, muscles, and tendons of hands constitute a highly efficient biological structure, precisely adapting to the shape and size of objects through multi-degree-of-freedom (DOF) movements and tendon drive mechanisms. This remarkable architecture inspires our integrated design, which combines the dexterous grasping abilities of human hands with the swift maneuverability of aerial flight, leading to a **Hand-Inspired Hand-Like** compact Aerial Robot for Manipulation, abbreviated as **HI-ARM** in this article (Fig. 1b). The proposed flying robot achieves delicate, multifunctional, maneuverable, and continuous aerial manipulation with a size of solely an adult hand (see Supplementary Movie 1). HI-ARM's design ~~inherits the biological characteristics of human hands~~ **incorporates hand-like features for functional grasping**, including an open C-shaped grasping contour to extend its range, a multi-DOF deformable joint structure to accommodate objects of various shapes, and a concise tendon-driven mechanism to reduce its total size and weight. As illustrated in Figure 1b, the C-shaped grasping contour ~~mimics the enveloping structure of a human hand~~ **provides a hand-like enveloping geometry**, significantly enhancing grasping stability and adaptability for objects. The composite 5-DOF finger-like structure, including a 2-DOF torsion and a 3-DOF extension (Fig. 2a) parts, enables ~~human-like~~ **efficient** manipulation. Additionally, HI-ARM equips four rotor-propellers for locomotion (Fig. 2a), inheriting the nature of flight agility and control simplicity from a conventional quadrotor aircraft. Thanks to the ~~hand-inspired hand-like~~ structure, HI-ARM enjoys superior adaptability for versatile tasks. It not only performs ~~human-like hand-like~~ grasping such as palm gripping and fingertip pinching (Fig. 1e and 1f), but also executes sophisticated operations like tree perching, door opening, object transportation, and human interaction (Fig. 1g–1j).

...

Bio-inspired Hand-like mechanism design

The design of HI-ARM draws inspiration from the grasping configuration, biological structure, and tendon drive mechanism of human hands. The employed configuration adopts a ~~hand-inspired hand-like~~ open grasping contour, providing a relatively broad grabbing range. In order to ~~emulate the biological structure and replicate the dexterous grabbing capability of human hands~~ **replicate dexterous grabbing capabilities while remaining mechanically novel**, the robot incorporates finger and palm modules as its core operational units (Fig. 3b(i)). With this design, HI-ARM includes a palm region for powerful gripping and fingertip areas for precise pinching, ~~like a hand~~ (Fig. 3a(ii)). This configuration supports a wide grasping range (0 ~ 10.0 cm), allowing HI-ARM to securely hold larger

objects (e.g., water bottles) with its palm while delicately picking up smaller items (e.g., tissues) using its fingertips, showcasing multi-modal grasping capabilities.

Human grasping relies on joint angle variations to achieve contraction (Fig. 3a(iii)). ~~Inspired by this principle~~ Following this functional principle, HI-ARM's finger modules incorporate a torsion structure comprising torsion springs and circular bearings (Fig. 3b(i)) to enable flexible finger bending. Given their ability to displace significantly in a short time, telescopic structures are incorporated into the deformable mechanism to increase the speed of grasping movements. To ensure that the telescopic modules are activated first, their spring stiffness is deliberately designed to be lower than that of the rotational modules, thereby enabling rapid contraction and closer contact with the target object. As illustrated in Figure 3e, the springs in the mechanism absorb energy during compression and torsion, and then release their stored energy to restore the shape of the robot, thereby reducing overall energy consumption. The integration of telescopic and torsional mechanisms forms a hybrid 5-DoF structure, allowing a variety of adaptive and flexible grasping, as shown in Figure 3f.

Building on the principles of the finger tendon sheath pulley system [59], HI-ARM employs a tendon drive mechanism for shape adaptation (Fig. 3e). Mimicking a finger's flexor digitorum profundus tendon (FDP tendon, shown in Fig. 3a(i)), a lightweight nylon rope is employed to transmit driving forces. Several V-shaped pulleys are integrated into the inner sides of finger and palm modules to redirect the force (Fig. 3b(i)). Unlike conventional morphing drones that rely on multiple actuators, ~~this bio-inspired design~~ this tendon-driven design utilizes a single actuator to drive the 5-DoF composite structure, minimizing the robot's size, weight, and energy consumption while simplifying its control complexity. Without knowing an object's shape, this underactuated structure can conform to the object's contour, and passively and collaboratively adjust the deformation of each torsion and extension component under single rope actuation to achieve stable grasping, demonstrating its adaptive grasping ability for various objects, as shown in Figure 4c.

With the proposed integrated design, HI-ARM features a compact ~~structure resembling an open human hand~~ size with a hand-like profile, with a total weight of just 556g. This allows it to possess more space for flight and ~~manipulation~~ grasping operations in narrow indoor environments, which is a task that can be challenging for ~~large-sized aerial manipulation~~ larger aerial robots. Additionally, the robot can deform to reduce its dimensions (Fig. 3b(ii)), improving passibility in extremely narrow spaces, as validated in Section *Applications in the wild*. HI-ARM defaults to the open configuration, enabling rapid execution of grasping tasks. By contrast, the closed configuration imposes continuous torque demands on the servo motor and partially obstructs propeller airflow, elevating power consumption and compromising flight endurance.

Comment 26

Furthermore, while the teleoperation demonstration is visually compelling, it is only loosely related to the essential of the proposed robot and may not be scientifically appropriate to include as a core contribution in this paper.

Response: We sincerely appreciate your comments. Regarding teleoperation, we agree that it may not be a core contribution of this work. Nevertheless, we would like to highlight its potential for future remote and high-altitude operations, serving as an extension of human capabilities. Alternatively, we can downplay this aspect in the *Introduction* and instead discuss it as a planned future direction in the *Discussion* section, as per your suggestion.

Introduction

...

Contributions. Firstly, we propose a biomimetic design that integrates grasping with flight in a compact robot platform. To the best knowledge of us, this is the first flying robotic hand in the robotics community. Secondly, we propose an efficient planner for empowering the flying robotic hand with precise autonomous operations, along with an adaptive controller for stable flight with varying loads and dynamic interactions, and state feedback for closed-loop autonomy. Finally, we demonstrate the huge potential of applying HI-ARM to versatile tasks, including object grasping, door opening, pole perching, cross-terrain transportation, and continuous human-robot interactions, pushing the boundaries of aerial robots from passive observation to active manipulation. In what follows, HI-ARM successfully completes multiple continuous aerial interactive tasks quickly and smoothly for the first time, highlighting its great potential as an intelligent assistant (see Section *Interaction with humans*). ~~In Section *Human-robot collaborative operation*, we engage individuals with mobility impairment in cross-height object transfer experiments, expanding the spatial reach of human operations.~~ Moreover, the robot demonstrates rapid object grasping and cross-terrain transportation, presenting an innovative solution for aerial delivery in Section *Applications in the wild*. These experimental results not only validate HI-ARM's multi-task capabilities but also pave the way for it in unmanned autonomous operations, robotic household service, wilderness rescue, remote assistance, and more.

...

Discussion

...

In future study, we plan to integrate force feedback into HI-ARM to enhance its ability to grasp fragile objects, such as eggs. Additionally, we aim to develop a new type of inner gripping surface to increase friction on smooth objects. ~~We also plan to incorporate a multi-modal foundation model to enhance the cognitive ability of HI-ARM, enabling it to perform more sophisticated tasks, such as autonomous valve operation in industrial scenarios. Notably, human-robot collaboration demonstrates HI-ARM's teleoperation capabilities, enabling the acquisition of high-quality real-world data to train models with minimized sim-to-real gaps.~~ Furthermore, building on our previous research on aerial swarm [60], we intend to develop a team of HI-ARM robots, which can execute much more complicated tasks such as collaborative transportation. We will continue our in-depth research into miniaturization and autonomy of the proposed flying robot under limited onboard resources, and push it to be industrialized.

Comment 27

Unclear sensing of morphing-related parameters: A central concern is how the system handles changes in geometry due to morphing, especially regarding the shifting propeller positions. For example, the propeller coordinates l_{x_i}, l_{y_i} in Equation (4), and r_{mod_i} in Supplementary Eq. (S2), must vary according to the deformation. Yet it is unclear how these are measured or estimated during flight. ~~The software architecture diagram (Fig. 2b) does not provide any indication of how this information is integrated into the control layer.~~ Clarification on this point is critical for assessing the

feasibility of the proposed control framework.

Response: We sincerely appreciate your suggestions on the sensing and estimation of morphing-related parameters. As noted, the variation of propeller coordinates (l_{x_i}, l_{y_i}) and structural positions \mathbf{r}_{mod_i} during morphing is not assumed to be constant but is explicitly modeled and updated through the deformation kinematics. These updates have been incorporated into the revised manuscript.

As detailed in the revised *Dynamics* section, the tendon-driven mechanism provides a one-to-one mapping between servo angular displacement and the deformation of telescopic and torsional modules. This mapping allows us to continuously compute the geometric changes of each structural module and thus update the positions of the propellers in real time. Specifically, rope displacement L_t (Eq. (S7)) is decomposed into telescopic and torsional deformation variables, which directly yield the updated module coordinates $\mathbf{r}_{mod_i}(t)$. From these, the propeller positions (l_{x_i}, l_{y_i}) in Eq. (4) and Supplementary Eq. (S2) are recalculated at each control cycle. Additional details are provided below.

Revise

Methods

Dynamics

In this study, we employ bold lowercase letters to denote vectors (e.g., \mathbf{v}) and bold uppercase letters for matrices (e.g., \mathbf{J}). Scalars are represented otherwise. As shown in Figure 2c(iii), we utilize a world frame \mathbf{W} with an orthonormal basis $\{\mathbf{x}_W, \mathbf{y}_W, \mathbf{z}_W\}$, where \mathbf{z}_W is oriented upward, counter to the direction of gravity. The body frame \mathbf{B} , situated at the geometric center of the proposed vehicle, is defined with an orthonormal basis $\{\mathbf{x}_B, \mathbf{y}_B, \mathbf{z}_B\}$. Let the position of the vehicle in the world frame be represented by $\mathbf{p}_W = (p_x, p_y, p_z)^\top$, its orientation by the quaternion $\mathbf{q}_W = (q_x, q_w, q_y, q_z)^\top$, and its linear velocity by $\mathbf{v}_W = (v_x, v_y, v_z)^\top$. Additionally, the angular velocity of the vehicle, expressed in the body frame, is given by $\boldsymbol{\omega}_B = (\omega_x, \omega_y, \omega_z)^\top$.

HI-ARM flight system is composed of four parallel rotor-propellers, whose position distribution approximates the 'X' configuration of a conventional quadrotor. Consequently, its flight dynamics model is similar to that of traditional quadrotors, and the device possesses the differential flatness characteristics [24] as general quadrotors. We utilize 6 DoFs to describe the ideal rigid-body kinematics and dynamics model of the robot. For the translational dynamics component, the following equations are utilized:

$$\begin{aligned}\dot{\mathbf{p}}_W &= \mathbf{v}_W, \\ \dot{\mathbf{v}}_W &= T\mathbf{z}_B/m + \mathbf{g},\end{aligned}\tag{R6}$$

where T and m are the collective thrust and total mass respectively; \mathbf{z}_B is the Z axis of the body frame expressed in the world frame; $\mathbf{g} = [0, 0, -g]^\top$ is the gravitational vector.

The rotational kinematic and dynamic equations are expressed as

$$\begin{aligned}\dot{\mathbf{q}}_W &= \frac{1}{2} \left(\begin{bmatrix} 0 \\ \boldsymbol{\omega}_B \end{bmatrix}_{\times} \right) \cdot \mathbf{q}_W, \\ \dot{\boldsymbol{\omega}}_B &= \mathbf{J}^{-1}(\boldsymbol{\tau} - \boldsymbol{\omega}_B \times \mathbf{J}\boldsymbol{\omega}_B),\end{aligned}\tag{R7}$$

where $[\cdot]_{\times}$ is the skew-symmetric matrix; $\boldsymbol{\tau}$ and \mathbf{J} are the total torque and inertia tensor matrix, respectively, **both of which are dynamically updated during morphing and grasping, with more details provided in the *Supplementary Materials*.**

Let k_t and k_c denote the thrust coefficient and torque coefficient, respectively, associated with the j -th motor. The rotational speed of the j -th motor is represented by Ω_j , and its position within the body frame is given by $\mathbf{l}_j = [l_{x_j}, l_{y_j}, l_{z_j}]^\top$. The collective thrust T and the torque $\boldsymbol{\tau}$ generated by the actuators are expressed as follows:

$$\begin{bmatrix} T \\ \boldsymbol{\tau} \end{bmatrix} = \mathbf{H}_k \mathbf{t}, \quad (\text{R8})$$

where $\mathbf{t} = [k_t \Omega_1^2, k_t \Omega_2^2, k_t \Omega_3^2, k_t \Omega_4^2]^\top$ represents the thrust generated by each rotor, \mathbf{H}_k is the time-variant mixed control matrix (MCM) while the vehicle morphing. Assuming that the center of gravity is $\mathbf{r}_{COG} = (r_x, r_y, r_z)^\top$ at this time, then \mathbf{H}_k is as follows:

$$\mathbf{H}_k = \begin{bmatrix} 1 & 1 & 1 & 1 \\ r_y + l_{y_1} & r_y + l_{y_2} & r_y + l_{y_3} & r_y + l_{y_4} \\ r_x - l_{x_1} & r_x - l_{x_2} & r_x - l_{x_3} & r_x - l_{x_4} \\ -k_c/k_t & k_c/k_t & -k_c/k_t & k_c/k_t \end{bmatrix}. \quad (\text{R9})$$

The module positions \mathbf{r}_{mod_i} and propeller coordinates (l_{x_i}, l_{y_i}) are not fixed; instead, they are dynamically updated during morphing and flight. Specifically, these parameters are derived from the tendon-driven morphing kinematics: the servo motor angle θ_a and rope displacement L_t determine the deformation of both the telescopic and torsional mechanisms, which in turn define the instantaneous propeller positions. As illustrated in Figure 2c, L'_i and L_i denote the lengths of the telescopic mechanism during motion and under nominal conditions, respectively, with $L'_i - L_i$ representing the compression displacement and reflecting variations in the distances between adjacent modules. As shown in Figure 2c(ii), θ_j denotes the rotation angle of the j -th torsional mechanism (comprising a torsion spring and a circumferential bearing). Accordingly, the propeller coordinates (l_{x_i}, l_{y_i}) are updated to (l'_{x_i}, l'_{y_i}) as follows:

$$\begin{aligned} l'_{x_1} &= l_{x_1} - (L'_i - L_i)/2 - l_0(1 - \cos\theta_1) \\ l'_{x_2} &= l_{x_2} + (L'_i - L_i)/2 \\ l'_{x_3} &= l_{x_3} + (L'_i - L_i)/2 \\ l'_{x_4} &= l_{x_4} - (L'_i - L_i)/2 - l_0(1 - \cos\theta_2) \\ l'_{y_1} &= l_{y_1} + (L'_i - L_i)/2 + l_0(1 - \sin\theta_1) \\ l'_{y_2} &= l_{y_2} + (L'_i - L_i)/2 \\ l'_{y_3} &= l_{y_3} - (L'_i - L_i)/2 \\ l'_{y_4} &= l_{y_4} - (L'_i - L_i)/2 - l_0(1 - \sin\theta_2) \end{aligned} \quad (\text{R10})$$

As a consequence, the control matrix \mathbf{H}_k is updated to \mathbf{H}'_k online according to the current morphing state. This formulation explicitly incorporates geometry changes induced by morphing into the control layer, rather than treating them as static.

...

Furthermore, this computation is updated into the control architecture shown in Fig. 2b. The servo motor encoder continuously provides angular displacement $\theta_a(t)$, which is mapped to morphing state variables and then passed to both the inertia tensor estimator and the control allocation module. This ensures that the mixed control matrix \mathbf{H}_k is updated online with the time-varying propeller coordinates. Therefore, the

proposed control framework does not rely on static assumptions of geometry; instead, it incorporates a real-time morphing state estimation pipeline that couples tendon actuation, structural kinematics, and control allocation. This clarification has been added to the revised manuscript to avoid ambiguity.

Revise

Multi-level adaptive control

An accurate, adaptive, and efficient controller is essential for smooth aerial manipulation. Compared to mature controllers for traditional quadrotors, the flight control system of HI-ARM faces greater challenges. These challenges mainly arise from two aspects (Fig. 2c(iv)): first, the model parameters (such as size, shape, COG, and inertia) dynamically change due to body deformation, which significantly disturbs flight control; second, during airborne operations, external factors such as load variations, close-range interaction forces, and air disturbances severely affect the stability and accuracy of the robot's control. In particular, to increase the speed of aerial operations, HI-ARM needs to deform while flying to complete tasks, imposing higher requirements on both flight control and deformation control.

To address these challenges, we propose a multi-level adaptive controller (Fig. 2b), endowing HI-ARM with precise aerial operation capabilities. To mitigate the negative effects of model variations, the robot incorporates an online model parameter identification approach to estimate key physical parameters, such as COG and inertia, ~~(detailed in Supplementary Information)~~ as illustrated in Fig. 2b and detailed further in *Supplementary Information*. For external disturbances, the system introduces an estimation and compensation method for external forces and torque disturbances to reduce their impact on control. Specifically, the \mathcal{L}_1 adaptive control algorithm [61] is employed in the position loop to mitigate external forces, while the incremental nonlinear dynamic inversion (INDI) control algorithm [62] is used in the attitude loop to handle external torques. These adaptive algorithms also reduce the negative impact of factors that affect trajectory tracking performance, such as propeller airflow interference and model mismatch [63].

...

Fig. R10

Fig. R11. Hardware and software architecture. (a) Hardware overview. (b) Software architecture, including a task planner, motion planning, adaptive control, estimation, and actuation. (c) Morphing and flying models: (i) morphing state at Normal Size, (ii) morphing state during object grasping, (iii) coordinate system representation, (iv) External interference during aerial operation.

Comment 28

Lack of comparative validation for control schemes: The proposed adaptive controller, including L1 and INDI schemes, appears theoretically sound, but its practical necessity remains unclear. Comparative experiments with baseline controllers (e.g., PID or standard geometric control) are essential to justify the added complexity. In particular, the perching detachment sequence shown at 2:48 in the main video appears unstable, raising concerns about the real-world robustness of the control system.

Response: We sincerely appreciate your suggestions. To address this, we have added ablation experiments for the controller, including comparisons with baseline (geometric control only), baseline+L1, base-

line+INDI, and baseline+L1+INDI. In this experiment, the combined baseline+L1+INDI control configuration achieves the best performance across all tested scenarios. By simultaneously providing position and attitude compensation, this multi-level adaptive strategy delivers robust rejection of external disturbances and enables accurate and stable trajectory tracking in open, deforming, and payload-bearing configurations.

Revise

Controller ablation

This section evaluates the ablation experiment of the proposed adaptive control. We adopt a geometric controller as the baseline, which serves as the conventional control scheme for quadrotors. To compensate for disturbances arising from structural deformation, manipulation, and other uncertainties, we integrate an L1 adaptive controller for real-time estimation and rejection of external forces, together with an Incremental Nonlinear Dynamic Inversion (INDI) controller for compensating external torque disturbances. To rigorously assess the effectiveness of the L1 and INDI controllers, we conduct a series of ablation experiments.

Specifically, four control modes are implemented: baseline (geometric controller only), baseline+L1, baseline+INDI, and baseline+L1+INDI. Under each mode, the robot is tested across three representative scenarios: (1) tracking a figure-eight trajectory in the open configuration, (2) tracking while undergoing structural deformation, and (3) tracking with an additional payload of about 200 g. The corresponding experimental results are presented in Fig. R12.

The results demonstrate that the integration of L1 and INDI controllers significantly enhances tracking accuracy and system stability across all scenarios, with particularly notable improvements under deformation and external disturbance conditions. Across all scenarios, the baseline controller consistently yields the poorest performance. The baseline+L1 configuration effectively compensates for errors in COG estimation and uncertain external forces, incorporating these corrections into the position control loop. As a result, the position-tracking error is substantially reduced, as shown in Fig. R12. This improvement is particularly pronounced in the payload-tracking experiment, where the dominant external disturbance arose from the gravitational force of the attached object.

In contrast, the baseline+INDI configuration primarily addresses external torque disturbances. During structural deformation, inaccuracies in inertia tensor estimation lead to attitude control errors, which in turn propagate to position tracking due to the inherent coupling between attitude and position dynamics in quadrotors. By compensating for these torque-induced errors, INDI significantly improves attitude stability and overall tracking precision.

Finally, the combined baseline+L1+INDI configuration achieved the best performance across all tested scenarios. By simultaneously providing position and attitude compensation, this multi-layer adaptive strategy delivers robust rejection of external disturbances and enables accurate and stable trajectory tracking in open, deforming, and payload-bearing configurations.

Fig. R12. Controller ablation. Four control modes: baseline, baseline+L1, baseline+INDI, and baseline+L1+INDI. Under each mode, the robot is tested across three representative scenarios: (1) tracking a figure-eight trajectory in the open configuration, (2) tracking while undergoing structural deformation, and (3) tracking with an additional payload of about 200 g.

In addition, in the Supplementary Movie, the perching maneuver may appear to involve slight collisions, which does not indicate insufficient controller performance. Rather, these minor impacts result from incidental contacts between the morphing structure and the tree trunk. We have repeated the experiments, and the results confirm that the proposed controller enables successful and smooth perching. The updated perching experiments are now updated in Supplementary Movie 1 and 3.

Comment 29

Demonstrations lack scientific depth: While the demonstrations are entertaining and diverse, many are superficial in their analysis. The “Interaction with humans” scenario appears overly choreographed and lacks scientific rigor. Similarly, the “Human-robot collaborative operation” section is interesting from an application standpoint but largely unrelated to the core mechanical and control contributions of the paper. It would be more appropriate to present this part as a separate study focused on teleoperation, with quantitative analysis of flight behavior during human operation.

Response: We sincerely appreciate your valuable feedback. The primary objective of this work is to develop a compact and agile aerial robot with manipulation capabilities that can operate in real indoor environments, such as smart homes and industrial logistics settings. In line with this goal, the “human-robot interaction” experiments are designed to demonstrate the feasibility of applying the system in such environments, particularly to validate the platform’s robustness and multifunctionality across continuous tasks. Through these demonstrations, we aim to broaden the robotics community’s perspective and explore the potential of aerial robots for future indoor and outdoor applications.

Furthermore, we fully agree that Nature Communications emphasizes scientific depth and rigorous anal-

ysis. In this regard, the manuscript presents a series of functional validation experiments—including palm grasping, fingertip pinching, perching, and door opening—that verify the effectiveness of the proposed algorithmic framework. The controller ablation and robustness tests quantitatively evaluate the performance of the adaptive control system, while experiments on grasping payload, planning time, and flight endurance further assess system-level metrics. Together, these experiments provide comprehensive quantitative evaluations of both the software and hardware components of the system. In contrast, the human–robot interaction experiments serve as system-level integration validations, which are crucial in robotics research. As the current system relies on external camera systems (e.g., a motion capture system) for target pose acquisition, the experimental configuration is subject to certain constraints, which may make the setup appear somewhat choreographed. These clarifications are explicitly added to the revised manuscript.

Revise

Interaction with humans

The compact size of HI-ARM enables it to adapt to spatially constrained environments in typical households, offering autonomous robotic services with great potential as an intelligent home assistant. We design a series of experiments that simulate daily scenarios to evaluate the possibility of HI-ARM in domestic environments (Fig. 6a). In this experiment, ~~the flying robot needs to execute multiple tasks in sequence, while interacting with a human.~~ **the positions and orientations of objects are acquired using a motion capture system, while the flying robot is required to sequentially perform multiple tasks involving human-robot interaction.** Figure 6b illustrates the continuous and smooth flight reference trajectory for multi-task operations generated by the aforementioned mission planner. Upon the arrival of the person, HI-ARM approaches the door, performs a fingertip grasp on an express box held by him, and delivers it to a storage bin (Fig. 6a ①①). The drone then flies to a table, uses a palm grasp to pick up a bottle of water, and hands it to the person (Fig. 6a ②③). While the person is drinking, the flying robot retrieves a boxed snack from the table (Fig. 6a ④⑤). Once the person finishes drinking, the empty bottle is handed to HI-ARM, which then deposits it into a trash can (Fig. 6a ⑥⑦). After completing services, HI-ARM flies to the coat rack and perches on it, entering standby mode (Fig. 6a ⑧). As shown in Figure 6d, HI-ARM can adaptively grasp different objects by adjusting the actuator’s angle accordingly. During this experiment, HI-ARM encounters perturbation errors at each operation point (Fig. 6c), primarily caused by load variations introduced by the objects. Thanks to the multi-level adaptive controller, HI-ARM is able to accurately estimate external force interference (Fig. 6d) and effectively compensate for the disturbances, gradually reducing errors. As shown in Figure 6c, the mean absolute tracking error (ATE) of the single-axis trajectory during the experiment is less than 2 cm. This coherent process demonstrates HI-ARM’s excellent ability to adaptively grasp different objects and move flexibly in domestic scenarios.

Furthermore, in our previous response to “Comment 26,” we have removed the “Human–robot collaborative operation” from the core contributions of this manuscript and repositioned it in the *Discussion* section as an exploration for future work; it will therefore not be reiterated here. Additionally, following your suggestion, we explicitly present the “Human-robot collaborative operation” section as a teleoperation scenario and supplement it with quantitative metrics, including trajectory length, average speed, control latency, and tracking control error. The newly added content is as follows.

Teleoperation for human-robot collaboration ~~collaborative operation~~

As a bio-inspired aerial interaction device, HI-ARM can function as a third flying hand for humans, responding to human intentions. As shown in Figure R13a, HI-ARM is equipped with a remote video transmission system that provides first-person-view (FPV) visual feedback. Similar to DJI's Avatar FPV drone, HI-ARM can be operated with a simplified, single-handed motion-based 3D controller, which maps hand movements to velocity commands in different directions. The controller also integrates a grasp button, allowing both flight and grasping actions to be performed with one hand. As shown in Fig. R13a, the communication between the controller and the robot is established via the ROS framework. As described in Section *Mission planning*, these commands are then processed by the mission planner to generate flight and deformation inputs for the controller, enabling the robot to execute aerial tasks. The following two experiments demonstrate HI-ARM's potential in human-assisted remote operations (see Supplementary Movie 4).

For individuals with limited mobility, retrieving items from different locations can be inaccessible, particularly in rugged terrains, areas with stairs/steps, or regions with significant height variations. In this experiment, we invite a participant with mobility impairment to operate HI-ARM, wearing video glasses to receive the onboard perspective. After a brief tutorial, the user is able to control HI-ARM to complete the object retrieval task from a distance according to his/her intention. As shown in Figure R13b, HI-ARM takes off from the second floor, navigates through trees and shrubs, and reaches the target position near the ground. It precisely grasps the target object—a cup of coffee—and successfully returns to the participant's side, demonstrating HI-ARM's potential in assisting individuals with disabilities. **In this task, the 46.2 m total retrieval trajectory is completed at an average velocity of 0.33 m/s, with an end-effector ATE of 0.08 m and a control latency of 256 ms, highlighting the stability and effectiveness of HI-ARM's teleoperation over distances exceeding 40 m.**

Even for individuals with normal mobility, there may be some situations where reaching objects at high places is difficult. This experiment simulates a badminton stuck in a tree, where the operator remotely assesses its position via onboard imagery. Subsequently, the person maneuvers HI-ARM to fly to the tree, grasp the badminton with its fingertips, and smoothly return to the ground (Fig. R13c), demonstrating its potential for remote airborne operations. **In this scenario, the trajectory measures 15.4 m, with an average velocity of 0.10 m/s, and an end-effector ATE of 0.04 m. The lower speed compared to the coffee task reflects the higher precision required for grasping lightweight objects.**

Fig. R13. Human-robot collaborative remote aerial operation. (a) Illustration of the equipment and principles of human-robot collaborative operation. (b) State curves of HI-ARM during the water bottle grasping process, including position tracking, force estimation, and thrust. (c) Quantitative teleoperation performance, including trajectory length, average velocity, control latency, and average tracking error (ATE). (d) Mobility-impaired people remotely pick up a cup of coffee using HI-ARM. (e) Humans retrieve a badminton from a tall tree using HI-ARM.

Comment 30

Minor Comments

INDI control: Why is the external torque estimated and compensated, but not the external force? Given that external forces can also affect flight performance, this choice needs justification.

Response: We sincerely appreciate your comments. Our multi-level adaptive controller integrates INDI for compensating external torque disturbances and \mathcal{L}_1 adaptive control for estimating external forces.

This design choice arises from the characteristics of each method: INDI relies on noisy measurements, such as motor speeds, which can introduce high-frequency noise, making it less suitable for interactive force operations [62]. In contrast, \mathcal{L}_1 adaptive control uses velocity observations to produce smoother, more stable force estimates, better suited to our control framework.

As demonstrated in the ablation studies and the experiments reported in the manuscript, this multi-level adaptive control scheme—combining force estimation and compensation—effectively enhances the performance of aerial manipulation during flight operations.

Revise

Multi-level adaptive control

...

To address these challenges, we propose a multi-level adaptive controller (Fig. 2b), endowing HI-ARM with precise aerial operation capabilities. To mitigate the negative effects of model variations, the robot incorporates an online model parameter identification approach to estimate key physical parameters, such as COG and inertia (detailed in *Supplementary Information*). For external disturbances, the system introduces an estimation and compensation method for external forces and torque disturbances to reduce their impact on control. **Specifically, the \mathcal{L}_1 adaptive control algorithm [61] is employed in the position loop to mitigate external forces, while the incremental nonlinear dynamic inversion (INDI) control algorithm [62] is used in the attitude loop to handle external torques.** These adaptive algorithms also reduce the negative impact of factors that affect trajectory tracking performance, such as propeller airflow interference and model mismatch [63].

...

Comment 31

Door opening (Fig. 5): **A door that can be opened with only 2–3 N is not realistic in most practical settings. Moreover, it seems like the robot can also push the door directly without actually grasping the handle, which undermines the credibility of this demonstration. Besides, what is the actual maximum pushing force the robot can exert?**

Response: We sincerely appreciate your comments. The experiment primarily demonstrates the capability of the proposed aerial robot to grasp a door handle and then use thrust to open the door, highlighting its combined manipulation and flight abilities. In our tests, the door is made of lightweight composite materials commonly found in temporary factory facilities on construction sites, with smooth hinges that allow humans to push or pull it easily. However, attempting to open a door with a conventional quadrotor is not feasible, as shown in Fig. R14. In a series of experiments, when the aerial robot attempts to push the door directly, the generated forward thrust causes the platform to tilt, resulting in only point contact with the door. As illustrated in Fig. R14b, this unstable contact can lead to slippage, preventing successful door opening. In contrast, our proposed aerial robot encloses the door handle within the central region of the body, enabling a stable and close interaction that allows the door to be successfully opened.

Moreover, we measure the maximum door-pushing force the robot can generate, which depends on the total thrust and the platform’s attitude. As illustrated in Figure R14c, when the robot pushes the door while grasping the handle, the door-opening angle θ_{door} is approximately 30° , and the maximum pushing force reaches about $T_{\text{max}} \sin 30^\circ \approx 5\text{N}$. Without the constraint of the handle, the maximum achievable door-opening angle θ_{max} increases to 55° , with a corresponding maximum pushing force of 8.2N.

Fig. R14. Door opening test. (a) Manual flight for door opening. (b) Using no door handle for door opening. (c) Max force test for opening the door.

Revise

Door opening

Thanks to its superior operational capability, HI-ARM can even be used to open a door. In this experiment, when the door is being pushed open, it introduces significant disturbances to the control of the drone. To ensure smooth door-opening (Fig. R16h), we employ the mission planner to generate an appropriate reference trajectory that accounts for dynamic constraints (more details can be found in *Methods*), enabling the drone to accurately grasp the door handle and open the door, as shown in Figure R16g. External force estimations suggest that the door mainly exerts disturbing forces on the robot along the X and Y axes (Fig. R16j), which are compensated by the proposed controller. As illustrated in Figure R16h, when the robot pushes the door while grasping the handle, the door-opening angle θ_{door} is approximately 30° , and the maximum pushing force reaches about $T_{\text{max}} \sin 30^\circ \approx 5\text{N}$. Without the constraint of the handle, the maximum achievable door-opening angle θ_{max} increases to 55° , with a corresponding maximum pushing force of 8.2N .

Fig. R15. Performance of perching and door opening.

Fig. R16. Performance of perching and door opening. (a) Humans grasp a handrail on trains and grip a tree trunk. (b) HI-ARM is perching on a tree trunk. (c) Motor speed curve during the perching maneuver. (d) HI-ARM is releasing from the tree trunk and flying away. (e) Normalized power curve. (f) Humans grab door handles to open doors. (g) Sequence of HI-ARM opening a door. (h) **Max force test for opening the door with/without the door handle.** (i) Position curve during the door opening process. (j) Estimated external forces during the door opening action.

Comment 32

Applications in the wild (Fig. 7): **The narrow-space traversal demo appears unconvincing**, as the passage seems wide enough even without morphing. **What are the robot's maximum and minimum widths before and after deformation?**

Response: We sincerely appreciate your comments. We have added explicit measurements: maximum width before deformation is 23.5 cm, and minimum width after deformation is about 18.5 cm. Although the test passage in Figure 7 may not appear particularly narrow, its minimum width is nearly equal to that of the drone. To demonstrate that the drone can traverse gaps narrower than its own width, we have supplemented the experiment with a deformation-based passage experiment. In this experiment, the narrow gap measures only 20.0 cm, which is smaller than the robot's maximum width, yet the robot can successfully deform and pass through it, demonstrating its capability to traverse narrow gaps. This experiment has been included in *Supplementary Information*.

Crossing with morphing

This section demonstrates the proposed flying robot's ability to traverse narrow gaps. Unlike conventional quadrotors, one of the key advantages of our design lies in its capacity to actively morph its structure, thereby adapting its size to different environments. To validate this environmental adaptability, we conduct controlled experiments on gap-crossing.

As shown in Fig. R17, the initial width of the robot is 23.5cm, whereas the horizontal gap measures only 20.0cm. Since the gap is narrower than the vehicle itself, direct passage is extremely challenging. Wang et al. [25] propose a planning strategy based on SE(3) to address this issue; however, their approach requires substantial space for acceleration and deceleration, which is particularly challenging in confined environments. In contrast, our flying robot can autonomously contract to a width of 18.8cm and pass smoothly through the gap without the need for obvious acceleration or deceleration. Figure R17 illustrates the moment when the robot successfully traverses the gap.

Fig. R17. Crossing with morphing. (a) A sequence of HI-ARM crossing a narrow gap with morphing. (b) State curves of HI-ARM during crossing with morphing process, including position tracking, Force estimation, servo motor's angle, and actual thrust.

Comment 33

Propeller placement: **Placing propellers below the robot may aid manipulation, but it also raises safety concerns, especially regarding potential collisions with the environment during tilted flight. The authors should discuss the implications of this design decision. Also, no flight endurance is clearly stated for the 3.5-inch propellers used.**

Response: We sincerely appreciate your suggestions and fully acknowledge the safety concerns raised. Compared with conventional quadrotors, our design places greater emphasis on safety during human–robot interactions. For example, in the “interaction with humans” experiment, participants pick up and place objects, typically approaching from above. In this context, a bottom-mounted propeller configuration significantly reduces the risk of accidental contact with human hands and prevents propeller-induced airflow from directly impinging on the vehicle body, thereby mitigating power loss. In contrast, a top-mounted propeller design exposes the vehicle to direct airflow, causing thrust loss; compensating for this effect would require outward extension of the propellers, increasing the robot’s size and compromising compactness. Furthermore, irrespective of propeller placement, inclined flight introduces potential collision risks with the environment.

Revise

Flight design and electronic components

This section provides a detailed overview of the HI-ARM flight system, which is powered by four rotor-propellers. As shown in Figure 2a, HI-ARM is equipped with 3.5-inch propellers, each driven by a T-motor F1404 2900KV motor, capable of generating a combined thrust of up to 1000g. **The motors are mounted beneath the finger and palm modules, and the bottom-mounted layout avoids direct propeller airflow and enhances human–robot interaction safety by minimizing hand contact.** The propellers are designed with a 6 mm height offset to prevent interference during module deformation and folding.

With a 70C discharge rate, the system delivers sufficient instantaneous power output to handle payloads of over 450g for grasping operations. The primary structure of the flight platform is composed of thick carbon fiber plates, selected for their high strength and lightweight properties. For flight control, HI-ARM uses a Kakute H7 mini flight control board to run the ArduPilot flight firmware, which collects high-frequency (>300Hz) real-time data on motor speed from the electronic speed controller (ESC) and posture from the Inertial Measurement Unit (IMU). High-frequency feedback helps the robot to estimate its motion states quickly, reducing system delay and improving control accuracy. The on-board computer, a Radxa ZERO board, runs a mission planner and adaptive controller (to be introduced in the following sections) for autonomous aerial manipulation. This controller processes system state data from the IMU, the servo motor, and the localization module, then sends control inputs to the brushless motor and the servo motor, which adjusts motor speeds for precise flight operations. More details about the components can be found in *Supplementary Information*.

...

We have also added flight endurance measurements, indicating a duration of approximately 237s. We also add the description in the *Supplementary Information*.

Revise

Flight performance

This section evaluates the flight performance of the proposed aerial robot, with a focus on payload capacity and endurance.

...

Endurance: In the open configuration, the aerial robot achieves a stable flight endurance of about 237 s, sufficient for performing representative tasks such as grasping, perching, door opening, and human–robot interaction. In the contracted configuration, airflow blockage causes an estimated power loss of about 10%, reducing the endurance to approximately 212 s.

Comment 34

Conclusion

While the system integration is commendable and the demonstrations are engaging, the manuscript contains numerous exaggerated claims and ad-hoc experimental setups. The core mechanical design is interesting and merits publication, but the current framing and presentation are not well-aligned with the standards of Nature Communications. A more suitable venue would be a specialized robotics or mechatronics journal where system integration and design novelty are more central.

Response: We sincerely appreciate your recognition of our mechanical design and constructive comments on improving the manuscript. While the core mechanical design indeed forms the foundation of our work, we emphasize that the strength of this study lies in the tight integration between hardware design and multi-level control and planning algorithms. These software components are not ad hoc demonstrations but systematically developed and experimentally validated to achieve closed-loop autonomy under complex dynamic conditions. The quantitative ablation studies presented in *Supplementary Information* further verify the necessity and effectiveness of these algorithms, enabling precise and consistent aerial manipulation.

Beyond system integration, our work aims to advance the frontier of aerial manipulation by bridging morphology, perception, and autonomy within a single compact platform. We believe that this integration represents a meaningful step toward expanding the operational and cognitive capabilities of aerial robots in three-dimensional spaces, offering both practical insights for real-world applications and conceptual inspiration for future research in embodied aerial intelligence. We believe that such integration between hardware and autonomy is central to advancing embodied aerial intelligence and aligns well with the interdisciplinary scope of Nature Communications.

Responses to Comments by Reviewer #3

Comment 35

The authors present an aerial robot with morphing capabilities, enabling functionalities such as grasping and perching. They also provide a detailed account of trajectory generation and control strategies. However, I have the following questions and suggestions for clarification:

The authors state that their work offers improvements over other aerial robots (e.g., references 40–47). It would strengthen the paper to include a clear, quantitative comparison of the proposed platform with existing systems. Relevant metrics might include platform size, weight, maximum payload, flight time, and other performance indicators.

Response: We sincerely appreciate your valuable feedback. Following your suggestion, we have conducted a detailed comparison with the works cited in References (ref 40–47), performing a horizontal analysis of platform size, weight, maximum payload, flight time, and other performance metrics. In addition, we have also supplemented the comparison with relative payload ratio, functional demonstrations, and other relevant indicators. Due to the reorganization and addition of references during manuscript revision, the studies originally cited as References (ref 40–47) are now listed as References (ref 44–51); we adopt the updated numbering in the quantitative analysis below.

The comparison shows that the proposed robot exhibits clear advantages in terms of size, weight, relative payload ratio, and functional capabilities. Its compact dimensions enable agile operation in confined indoor spaces, which is particularly important for environments with many people or scenarios involving human–robot interaction, as smaller platforms inherently pose lower safety risks compared with larger aerial robots. However, we also acknowledge that our platform has limitations in flight endurance and absolute payload capacity. Addressing these aspects is an important goal for future optimization to gradually elevate the system to application-level performance.

These updates have been incorporated into *Supplementary Information* in the revised manuscript.

Revise

Quantitative comparison of aerial manipulation robots

In the Introduction, we describe how recent studies (44–51) attempt to exploit the intrinsic structures of robots for aerial manipulation by reducing external attachments in order to lower system complexity. However, these platforms often encounter reduced accuracy due to additional actuators or increased mechanical complexity introduced by movable structures, which in turn constrain their operational range, precision, and speed.

To quantitatively highlight the advantages of our integrated aerial manipulation robot, we conduct a systematic comparison across multiple dimensions, including maximum actual payload, payload ratio, flight time, system size, overall weight, and demonstrated task capabilities. Existing platforms exhibit diverse trade-offs among payload capacity, endurance, and task versatility. The early large-scale morphing system by Zhao et al. (44) achieves relatively long flight durations of 180 seconds but relies on a heavy structure of 7.38 kilograms, resulting in limited payload efficiency. Later designs, such as those by Shi et al. (45) and Zhao et al. (46), increase payload capacity to 1.2 kilograms with a ratio of 35.3 percent, but at the expense of increased system mass of 3.4 kilograms and larger volume. Lightweight systems, such as those developed by Bucki et al. (47) and Zhao et al. (48), reduce the total mass to the sub-kilogram scale, yet exhibit constrained payload ratios of 13.3 percent and 29.4 percent, along with limited endurance. More recent works, including Falanga et al. (49),

Wu et al. (50), and Xu et al. (51), introduce agile morphing and perching capabilities, while their payload ratios remain moderate, ranging from 13.6 to 25.7 percent, and endurance spans from 126 to 253 seconds.

In contrast, our proposed HI-ARM platform achieves a maximum payload ratio of 80.8% (0.45kg payload with only 0.556 kg system weight), substantially surpassing prior designs. Despite a flight endurance of approximately 237 seconds, the platform remains relatively competitive due to its compact size of about 23.5 by 23.5 centimeters and 0.556 kg lightweight structure, while enabling a wide range of versatile operations, including palm and fingertip grasps, perching, door opening, structural morphing, teleoperation, and continuous multi-task operation. This balance between payload efficiency, flight endurance, and manipulation versatility highlights the superiority of HI-ARM and underscores its potential for aerial manipulation tasks in complex indoor and outdoor environments.

[Figure Redacted]

Platform	M. Zhao, et al. (44)	F. Shi, et al. (45)	M. Zhao, et al. (46)	N. Bucki, et al. (47)	N. Zhao, et al. (48)	D. Falanga, et al. (49)	Y. Wu, et al. (50)	M. Xu, et al. (51)	Ours
Max actual payload (presented)	-	1.2 kg	1.0 kg	0.083 kg	~ 0.3 kg	-	0.428 kg	0.160 kg	0.45 kg
Max payload ratio (presented)	-	-	35.30%	13.30%	29.40%	-	25.70%	13.60%	80.80%
Flight time	~ 180s	-	900s	-	-	253 s	126s	-	237s
Normal size	~169.6cm	-	-	44cm*44cm	-	45.7cm*45.7cm	41.4cm*41.4cm	~36cm*36cm	23.5*23.5cm
weight	7.382 kg	-	3.4 kg	0.624g	1.02Kg	0.580 kg	1.665 kg	1.176kg	0.556 kg
Demonstrations	Morphing & crossing	Pivoting	Grasping & releasing, sheet expanding	Morphing & crossing, grasping	Grasping	Grasping	Morphing & crossing, Grasping	Morphing & crossing, grasping, perching	Palm grasp, fingertip pinch, perching, door opening, Morphing & crossing, teleoperation, Continuous multi-task operation

Fig. R18. Quantitative comparison of previous aerial manipulation robots.

Comment 36

It appears that the front and rear motor cases differ in height. Is there a specific design rationale for this asymmetry? Would having equal heights simplify control or improve performance in any way?

Response: We sincerely appreciate your comments. The modules' heights are not completely uniform. This is primarily due to the complex torsion mechanism in the finger module, which occupies additional space and results in height differences between the finger and palm modules. This asymmetric design prevents propeller interference during module deformation and folding. In principle, this height discrepancy could be adjusted by inserting appropriately sized aluminum spacers to modify the mounting height. However, during our testing, we do not observe significant effects on flight performance, and therefore, we retain the current configuration.

Flight design and electronic components

This section provides a detailed overview of the HI-ARM flight system, which is powered by four rotor-propellers. As shown in Figure 2a, HI-ARM is equipped with 3.5-inch propellers, each driven by a T-motor F1404 2900KV motor, capable of generating a combined thrust of up to 1000g. **The motors are mounted beneath the finger and palm modules, and the bottom-mounted layout avoids direct propeller airflow and enhances human–robot interaction safety by minimizing hand contact. The propellers are designed with a 6 mm height offset to prevent interference during module deformation and folding.**

With a 70C discharge rate, the system delivers sufficient instantaneous power output to handle payloads of over 450g for grasping operations. The primary structure of the flight platform is composed of thick carbon fiber plates, selected for their high strength and lightweight properties. For flight control, HI-ARM uses a Kakute H7 mini flight control board to run the ArduPilot flight firmware, which collects high-frequency (>300Hz) real-time data on motor speed from the electronic speed controller (ESC) and posture from the Inertial Measurement Unit (IMU). High-frequency feedback helps the robot to estimate its motion states quickly, reducing system delay and improving control accuracy. The on-board computer, a Radxa ZERO board, runs a mission planner and adaptive controller (to be introduced in the following sections) for autonomous aerial manipulation. This controller processes system state data from the IMU, the servo motor, and the localization module, then sends control inputs to the brushless motor and the servo motor, which adjusts motor speeds for precise flight operations. More details about the components can be found in *Supplementary Information*.

Comment 37

For the autonomous tasks demonstrated, **it seems the robot is aware of the positions and orientations of the target objects (e.g., the water bottle and door handle). If this is the case, the authors should state this assumption explicitly. Otherwise, could the authors clarify how the robot estimates the positions and orientations of these objects?**

Response: We sincerely appreciate your insightful comments and suggestions. In designing the localization and estimation framework, we aim to balance accuracy, weight, and onboard computational load. For indoor tasks, precise object pose estimation is critical for accurate grasping and manipulation. To ensure reliability, we employ an external optical motion-capture system that provides sub-millimeter state estimation for both the aerial robot and the manipulated objects. For outdoor tasks, we utilize the Intel RealSense T261 tracking module, which offers a lightweight and stable solution for short-range state estimation. We acknowledge that the original manuscript did not clearly state this assumption, and we have now explicitly clarified it in the revised version. These configurations, integrated with adaptive control, motion planning, and parameter estimation, enable reliable execution of the autonomous aerial manipulation tasks presented.

Results

Flight design and electronic components

... For localization and estimation, we aim to balance accuracy, weight, and onboard computational load. In indoor experiments, an external optical motion-capture system provides sub-millimeter state estimation of both the aerial robot and the manipulated objects, enabling closed-loop integration with mission planning and adaptive control for autonomous operation. In outdoor experiments, we employ the Intel RealSense T261 tracking module, which weighs 26 g and offers a lightweight alternative to LiDAR-based localization systems while maintaining stable short-range state estimation (for example, within 20 m) suitable for experimental validation.

...

Interaction with humans

The compact size of HI-ARM enables it to adapt to spatially constrained environments in typical households, offering autonomous robotic services with great potential as an intelligent home assistant. We design a series of experiments that simulate daily scenarios to evaluate the possibility of HI-ARM in domestic environments (Fig. 6a). In this experiment, ~~the flying robot needs to execute multiple tasks in sequence, while interacting with a human.~~ **the positions and orientations of objects are acquired using a motion capture system, while the flying robot is required to sequentially perform multiple tasks involving human-robot interaction.** Figure 6b illustrates the continuous and smooth flight reference trajectory for multi-task operations generated by the aforementioned mission planner. Upon the arrival of the person, HI-ARM approaches the door, performs a fingertip grasp on an express box held by him, and delivers it to a storage bin (Fig. 6a ①①). The drone then flies to a table, uses a palm grasp to pick up a bottle of water, and hands it to the person (Fig. 6a ②③). While the person is drinking, the flying robot retrieves a boxed snack from the table (Fig. 6a ④⑤). Once the person finishes drinking, the empty bottle is handed to HI-ARM, which then deposits it into a trash can (Fig. 6a ⑥⑦). After completing services, HI-ARM flies to the coat rack and perches on it, entering standby mode (Fig. 6a ⑧). As shown in Figure 6d, HI-ARM can adaptively grasp different objects by adjusting the actuator's angle accordingly. During this experiment, HI-ARM encounters perturbation errors at each operation point (Fig. 6c), primarily caused by load variations introduced by the objects. Thanks to the multi-level adaptive controller, HI-ARM is able to accurately estimate external force interference (Fig. 6d) and effectively compensate for the disturbances, gradually reducing errors. As shown in Figure 6c, the mean absolute tracking error (ATE) of the single-axis trajectory during the experiment is less than 2 cm. This coherent process demonstrates HI-ARM's excellent ability to adaptively grasp different objects and move flexibly in domestic scenarios.

Furthermore, achieving precise, robust, and lightweight localization remains a formidable challenge. Although state estimation is not the primary focus of this work, we recognize it as a crucial step toward achieving higher levels of autonomy. We explicitly acknowledge this challenge and highlight lightweight and robust onboard localization and estimation as key directions for future research. A detailed discussion of this aspect has been incorporated into the revised *Discussion* section of the manuscript.

Discussion

In this study, we develop an innovative hand-inspired compact flying robot for manipulation, which integrates biomimetic grasping and aerial flying capabilities. We propose an efficient autonomous framework that enables the robot to perform precise and smooth aerial manipulation in real-world environments. The proposed robot demonstrates excellent performance across indoor and outdoor tasks, showing considerable potential as future smart home assistants, moving flying cameras, aerial teleoperation tools and flying delivery robots. **However, current state estimation depends on external localization, and the onboard visual localization module exhibits cumulative drift during long-range outdoor tasks. Achieving full autonomy requires not only accurate localization but also the ability to interpret the environment from visual inputs. End-to-end visual reinforcement learning approaches [3, 4], which directly map perception to control while adaptively correcting state estimation errors, offer a promising route to enhance system autonomy in the future.**

...

References

- [1] Wei Xu and Fu Zhang. Fast-lio: A fast, robust lidar-inertial odometry package by tightly-coupled iterated kalman filter. *IEEE Robotics and Automation Letters*, 6(2):3317–3324, 2021.
- [2] Haoyang Ye, Yuying Chen, and Ming Liu. Tightly coupled 3d lidar inertial odometry and mapping. *CoRR*, abs/1904.06993, 2019.
- [3] Minghuan Liu, Zixuan Chen, Xuxin Cheng, Yandong Ji, Ri-Zhao Qiu, Ruihan Yang, and Xiaolong Wang. Visual whole-body control for legged loco-manipulation, 2024.
- [4] Tianyue Wu, Yeke Chen, Tianyang Chen, Guangyu Zhao, and Fei Gao. Whole-body control through narrow gaps from pixels to action. In *2025 IEEE International Conference on Robotics and Automation (ICRA)*, pages 11317–11324, 2025.
- [5] John Ruben and Alan Feduccia. The origin and evolution of birds. *Bioscience*, 47, 06 1997.
- [6] Kenneth P Dial. Wing-assisted incline running and the evolution of flight. *Science*, 299(5605):402–404, 2003.
- [7] R Cano, C Pérez, F Pruano, A Ollero, and G Heredia. Mechanical design of a 6-dof aerial manipulator for assembling bar structures using uavs. In *2nd RED-UAS 2013 workshop on research, education and development of unmanned aerial systems*, volume 218, 2013.

- [8] Konstantin Kondak, Felix Huber, Marc Schwarzbach, Maximilian Laiacker, Dominik Sommer, Manuel Bejar, and Aníbal Ollero. Aerial manipulation robot composed of an autonomous helicopter and a 7 degrees of freedom industrial manipulator. In *2014 IEEE international conference on robotics and automation (ICRA)*, pages 2107–2112. IEEE, 2014.
- [9] Antonio E Jimenez-Cano, Jesús Martín, Guillermo Heredia, Aníbal Ollero, and Raul Cano. Control of an aerial robot with multi-link arm for assembly tasks. In *2013 IEEE International Conference on Robotics and Automation*, pages 4916–4921. IEEE, 2013.
- [10] Raphael Zufferey, Jesus Tormo-Barbero, Daniel Feliu-Talegón, Saeed Rafee Nekoo, José Ángel Acosta, and Anibal Ollero. How ornithopters can perch autonomously on a branch. *Nature Communications*, 13(1):7713, 2022.
- [11] Moju Zhao, Kei Okada, and Masayuki Inaba. Versatile articulated aerial robot dragon: Aerial manipulation and grasping by vectorable thrust control. *The International Journal of Robotics Research*, 42(4-5):214–248, 2023.
- [12] William RT Roderick, Mark R Cutkosky, and David Lentink. Bird-inspired dynamic grasping and perching in arboreal environments. *Science Robotics*, 6(61):eabj7562, 2021.
- [13] Samuel Ubellacker, Aaron Ray, James M Bern, Jared Strader, and Luca Carlone. High-speed aerial grasping using a soft drone with onboard perception. *npj Robotics*, 2(1):5, 2024.
- [14] William Stewart, Luca Guarino, Yegor Piskarev, and Dario Floreano. Passive perching with energy storage for winged aerial robots. *Advanced Intelligent Systems*, 5(4):2100150, 2023.
- [15] Emanuele Aucone, Steffen Kirchgeorg, Alice Valentini, Loïc Pellissier, Kristy Deiner, and Stefano Mintchev. Drone-assisted collection of environmental dna from tree branches for biodiversity monitoring. *Science robotics*, 8(74):eadd5762, 2023.
- [16] Moju Zhao, Tomoki Anzai, Fan Shi, Xiangyu Chen, Kei Okada, and Masayuki Inaba. Design, modeling, and control of an aerial robot dragon: A dual-rotor-embedded multilink robot with the ability of multi-degree-of-freedom aerial transformation. *IEEE Robotics and Automation Letters*, 3(2):1176–1183, 2018.
- [17] Fan Shi, Moju Zhao, Masaki Murooka, Kei Okada, and Masayuki Inaba. Aerial regrasping: Pivoting with transformable multilink aerial robot. In *2020 IEEE International Conference on Robotics and Automation (ICRA)*, pages 200–207. IEEE, 2020.
- [18] Moju Zhao, Tomoki Anzai, Fan Shi, Toshiya Maki, Takuzumi Nishio, Keita Ito, Naoki Kuromiya, Kei Okada, and Masayuki Inaba. Versatile multilinked aerial robot with tilted propellers: Design, modeling, control, and state estimation for autonomous flight and manipulation. *Journal of Field Robotics*, 38(7):933–966, 2021.
- [19] Nathan Bucki, Jerry Tang, and Mark W Mueller. Design and control of a midair-reconfigurable quadcopter using unactuated hinges. *IEEE Transactions on Robotics*, 39(1):539–557, 2022.
- [20] Na Zhao, Yudong Luo, Hongbin Deng, Yantao Shen, and Hao Xu. The deformable quad-rotor enabled and wasp-pedal-carrying inspired aerial gripper. In *2018 IEEE/RSJ International Conference on Intelligent Robots and Systems (IROS)*, pages 1–9. IEEE, 2018.

- [21] Davide Falanga, Kevin Kleber, Stefano Mintchev, Dario Floreano, and Davide Scaramuzza. The foldable drone: A morphing quadrotor that can squeeze and fly. *IEEE Robotics and Automation Letters*, 4(2):209–216, 2018.
- [22] Yuze Wu, Fan Yang, Ze Wang, Kaiwei Wang, Yanjun Cao, Chao Xu, and Fei Gao. Ring-rotor: A novel retractable ring-shaped quadrotor with aerial grasping and transportation capability. *IEEE Robotics and Automation Letters*, 8(4):2126–2133, 2023.
- [23] Mengxin Xu, Qixin De, Dafang Yu, An Hu, Zhe Liu, and Hesheng Wang. Biomimetic morphing quadrotor inspired by eagle claw for dynamic grasping. *IEEE Transactions on Robotics*, 2024.
- [24] Daniel Mellinger and Vijay Kumar. Minimum snap trajectory generation and control for quadrotors. In *2011 IEEE international conference on robotics and automation*, pages 2520–2525. IEEE, 2011.
- [25] Zhepei Wang, Xin Zhou, Chao Xu, and Fei Gao. Geometrically constrained trajectory optimization for multicopters. *IEEE Transactions on Robotics*, 38(5):3259–3278, 2022.
- [26] Shyam Sundar Kannan and Byung-Cheol Min. Autonomous drone delivery to your door and yard. In *2022 International Conference on Unmanned Aircraft Systems (ICUAS)*, pages 452–461, 2022.
- [27] Amr Afifi, Gianluca Corsini, Quentin Sable, Youssef Aboudorra, Daniel Sidobre, and Antonio Franchi. Physical human-aerial robot interaction and collaboration: Exploratory results and lessons learned. In *2023 International Conference on Unmanned Aircraft Systems (ICUAS)*, pages 956–962, 2023.
- [28] Suarez Alejandro, Gonzalez Antonio, Alvarez Carlos, and Ollero Anibal. Through-window home aerial delivery system with in-flight parcel load and handover: Design and validation in indoor scenario. *International Journal of Social Robotics*, 16:2109–2132, 2024.
- [29] Anibal Ollero, Marco Tognon, Alejandro Suarez, Dongjun Lee, and Antonio Franchi. Past, present, and future of aerial robotic manipulators. *IEEE Transactions on Robotics*, 38(1):626–645, 2022.
- [30] Guillermo Heredia, AE Jimenez-Cano, I Sanchez, Domingo Llorente, V Vega, J Braga, JA Acosta, and Anibal Ollero. Control of a multirotor outdoor aerial manipulator. In *2014 IEEE/RSJ international conference on intelligent robots and systems*, pages 3417–3422. IEEE, 2014.
- [31] Carmine Dario Bellicoso, Luca Rosario Buonocore, Vincenzo Lippiello, and Bruno Siciliano. Design, modeling and control of a 5-dof light-weight robot arm for aerial manipulation. In *2015 23rd Mediterranean Conference on Control and Automation (MED)*, pages 853–858. IEEE, 2015.
- [32] Suseong Kim, Seungwon Choi, and H Jin Kim. Aerial manipulation using a quadrotor with a two dof robotic arm. In *2013 IEEE/RSJ International Conference on Intelligent Robots and Systems*, pages 4990–4995. IEEE, 2013.
- [33] Justin Thomas, Joe Polin, Koushil Sreenath, and Vijay Kumar. Avian-inspired grasping for quadrotor micro uavs. In *International Design Engineering Technical Conferences and Computers and Information in Engineering Conference*, volume 55935, page V06AT07A014. American Society of Mechanical Engineers, 2013.
- [34] Meng Wang, Zeshuai Chen, Kexin Guo, Xiang Yu, Youmin Zhang, Lei Guo, and Wei Wang. Millimeter-level pick and peg-in-hole task achieved by aerial manipulator. *IEEE Transactions on Robotics*, 2023.

- [35] A Suarez, AE Jimenez-Cano, VM Vega, G Heredia, A Rodriguez-Castaño, and A Ollero. Lightweight and human-size dual arm aerial manipulator. In *2017 international conference on unmanned aircraft systems (ICUAS)*, pages 1778–1784. IEEE, 2017.
- [36] Matko Orsag, Christopher Korpela, Stjepan Bogdan, and Paul Oh. Valve turning using a dual-arm aerial manipulator. In *2014 international conference on unmanned aircraft systems (ICUAS)*, pages 836–841. IEEE, 2014.
- [37] Todd W Danko, Kenneth P Chaney, and Paul Y Oh. A parallel manipulator for mobile manipulating uavs. In *2015 IEEE international conference on technologies for practical robot applications (TePRA)*, pages 1–6. IEEE, 2015.
- [38] Ketao Zhang, Pisak Chermprayong, Feng Xiao, Dimos Tzoumanikas, Barrie Dams, Sebastian Kay, Basaran Bahadir Kocer, Alec Burns, Lachlan Orr, Talib Alhinai, et al. Aerial additive manufacturing with multiple autonomous robots. *Nature*, 609(7928):709–717, 2022.
- [39] Huazi Cao, Jiahao Shen, Cunjia Liu, Bo Zhu, and Shiyu Zhao. Motion planning for aerial pick-and-place based on geometric feasibility constraints. *arXiv preprint arXiv:2306.04970*, 2023.
- [40] Spencer B Backus, Lael U Odhner, and Aaron M Dollar. Design of hands for aerial manipulation: Actuator number and routing for grasping and perching. In *2014 IEEE/RSJ International Conference on Intelligent Robots and Systems*, pages 34–40. IEEE, 2014.
- [41] Paul EI Pounds, Daniel R Bersak, and Aaron M Dollar. Grasping from the air: Hovering capture and load stability. In *2011 IEEE international conference on robotics and automation*, pages 2491–2498. IEEE, 2011.
- [42] Daniel Mellinger, Quentin Lindsey, Michael Shomin, and Vijay Kumar. Design, modeling, estimation and control for aerial grasping and manipulation. In *2011 IEEE/RSJ International Conference on Intelligent Robots and Systems*, pages 2668–2673. IEEE, 2011.
- [43] Katie M Popek, Matthew S Johannes, Kevin C Wolfe, Rachel A Hegeman, Jessica M Hatch, Joseph L Moore, Kapil D Katyal, Bryanna Y Yeh, and Robert J Bamberger. Autonomous grasping robotic aerial system for perching (agrasp). In *2018 IEEE/RSJ International Conference on Intelligent Robots and Systems (IROS)*, pages 1–9. IEEE, 2018.
- [44] Markus Ryll, Giuseppe Muscio, Francesco Pierri, Elisabetta Cataldi, Gianluca Antonelli, Fabrizio Caccavale, Davide Bicego, and Antonio Franchi. 6d interaction control with aerial robots: The flying end-effector paradigm. *The International Journal of Robotics Research*, 38(9):1045–1062, 2019.
- [45] Antonio Franchi, Ruggero Carli, Davide Bicego, and Markus Ryll. Full-pose tracking control for aerial robotic systems with laterally bounded input force. *IEEE Transactions on Robotics*, 34(2):534–541, 2018.
- [46] Karen Bodie, Maximilian Brunner, Michael Pantic, Stefan Walser, Patrick Pfändler, Ueli Angst, Roland Siegwart, and Juan Nieto. Active interaction force control for contact-based inspection with a fully actuated aerial vehicle. *IEEE Transactions on Robotics*, 37(3):709–722, 2020.
- [47] Sangyul Park, Jeongseob Lee, Joonmo Ahn, Myungsin Kim, Jongbeom Her, Gi-Hun Yang, and Dongjun Lee. Odar: Aerial manipulation platform enabling omnidirectional wrench generation. *IEEE/ASME Transactions on mechatronics*, 23(4):1907–1918, 2018.

- [48] Hideyuki Tsukagoshi, Masahiro Watanabe, Takahiro Hamada, Dameitry Ashlih, and Ryuma Iizuka. Aerial manipulator with perching and door-opening capability. In *2015 IEEE international conference on robotics and automation (ICRA)*, pages 4663–4668. IEEE, 2015.
- [49] Georgios Darivianakis, Kostas Alexis, Michael Burri, and Roland Siegwart. Hybrid predictive control for aerial robotic physical interaction towards inspection operations. In *2014 IEEE international conference on robotics and automation (ICRA)*, pages 53–58. IEEE, 2014.
- [50] Karen Bodie, Maximilian Brunner, Michael Pantic, Stefan Walser, Patrick Pfändler, Ueli Angst, Roland Siegwart, and Juan Nieto. An omnidirectional aerial manipulation platform for contact-based inspection. *arXiv preprint arXiv:1905.03502*, 2019.
- [51] Rui Peng, Zehao Wang, and Peng Lu. Aecom: An aerial continuum manipulator with imu-based kinematic modeling and tendon-slacking prevention. *IEEE Transactions on Systems, Man, and Cybernetics: Systems*, 2023.
- [52] Pei Jiang, Ji Luo, Jiaying Li, Michael ZQ Chen, Yonghua Chen, Yang Yang, and Rui Chen. A novel scaffold-reinforced actuator with tunable attitude ability for grasping. *IEEE Transactions on Robotics*, 39(2):1164–1177, 2022.
- [53] Lydia Hingston, Jonathan Mace, Joao Buzzatto, and Minas Liarokapis. Reconfigurable, adaptive, lightweight grasping mechanisms for aerial robotic platforms. In *2020 IEEE International Symposium on Safety, Security, and Rescue Robotics (SSRR)*, pages 169–175. IEEE, 2020.
- [54] Mengxin Xu, Siyuan Huang, Ruokun He, Dafang Yu, and Hesheng Wang. Aerial shooting manipulator for distant grasping. *IEEE Robotics and Automation Letters*, 8(4):1991–1998, 2023.
- [55] Rui Peng, Yu Wang, Minghao Lu, and Peng Lu. A dexterous and compliant aerial continuum manipulator for cluttered and constrained environments. *Nature Communications*, 16(1):889, 2025.
- [56] Erik Bauer, Marc Blöchliger, Pascal Strauch, Arman Raayatsanati, Curdin Cavelti, and Robert K Katzschmann. An open-source soft robotic platform for autonomous aerial manipulation in the wild. *arXiv preprint arXiv:2409.07662*, 2024.
- [57] Pham H Nguyen, Karishma Patnaik, Shatadal Mishra, Panagiotis Polygerinos, and Wenlong Zhang. A soft-bodied aerial robot for collision resilience and contact-reactive perching. *Soft Robotics*, 10(4):838–851, 2023.
- [58] S Susan. Gray’s anatomy e-book: The anatomical basis of clinical practice, 2015.
- [59] James R Doyle. Anatomy of the finger flexor tendon sheath and pulley system. *The Journal of hand surgery*, 13(4):473–484, 1988.
- [60] Xin Zhou, Xiangyong Wen, Zhepei Wang, Yuman Gao, Haojia Li, Qianhao Wang, Tiankai Yang, Haojian Lu, Yanjun Cao, Chao Xu, et al. Swarm of micro flying robots in the wild. *Science Robotics*, 7(66):eabm5954, 2022.
- [61] Zhuohuan Wu, Sheng Cheng, Kasey A Ackerman, Aditya Gahlawat, Arun Lakshmanan, Pan Zhao, and Naira Hovakimyan. L1 adaptive augmentation for geometric tracking control of quadrotors. In *2022 International Conference on Robotics and Automation (ICRA)*, pages 1329–1336. IEEE, 2022.
- [62] Ezra Tal and Sertac Karaman. Accurate tracking of aggressive quadrotor trajectories using incremental nonlinear dynamic inversion and differential flatness. *IEEE Transactions on Control Systems Technology*, 29(3):1203–1218, 2020.

- [63] Xiaofeng Wang and Naira Hovakimyan. L1 adaptive controller for nonlinear time-varying reference systems. *Systems & Control Letters*, 61(4):455–463, 2012.

Revision Report for NCOMMS-25-21863A

Hand-like Autonomous Flying Robot for Airborne Grasping and Interaction

Yuze Wu, Fan Yang, Rui Jin, Yuhang Zhong, Junjie Wang, Xuankang Wu, and Fei Gao

The authors of the paper would like to thank the Editors and Reviewers for their timely handling of our manuscript and useful feedback. They have helped us improve our paper in several aspects, for which we are really grateful. We provide this document as the reply to the previous review comments and the revision of the submitted manuscript (“Hand-like Autonomous Flying Robot for Airborne Grasping and Interaction”, submission ID NCOMMS-25-21863A).

As instructed, the paper has been modified and extended according to the comments from editors and reviewers. Details on the revisions and specific answers to all queries are provided herein and can also be found in the table below for further clarification. Please note that, the comments of editor and reviewer are shown in red boxes. Author responses appear in normal font. Text of the revised manuscripts is shown in blue boxes with deletions marked and revised text in red. Text quoted from the manuscripts is underlined. Figure, table, and equation numbers in the Supplementary Information are preceded by an “S”, e.g., Fig. S1, Table S2, and Eq. S3. Elements appearing with an “R” prefix, e.g., Fig. R1 correspond to this response letter, while unmarked elements, e.g., Fig. 1, correspond to the main text. Herein, main text, Supplementary Information, and response letter are abbreviated as M.T., S.M., and R.L., respectively.

Table R1. Explanation of abbreviations, notations and annotations.

Abbreviation	Explanation
M.T.	Main Text
S.M.	Supplementary Information
R.L.	Response Letter
Notations and Annotations	Explanation
Normal Font	Author responses
red boxes	Reviewer comments
blue boxes	Revised text
deletions	Deletions
red	Revised text in red
underlined	Quoted text from the manuscripts
S	Supplementary Information figure, table, and equation numbers
R	Figure and table numbers in the Response Letter

Response to Comments by Editors

Comment 1

In the endurance flight tests (Comment 5), it is stated that the aerial robot can fly for 237 s in the open configuration, and 212 s in the close configuration due to the thrust loss. In both cases the endurance test is conducted for the 450 grams weight? **Please, provide the maximum flight time for the no load and full load cases.**

Response: We sincerely appreciate your valuable suggestion and have included the endurance time under both full-load and no-load conditions in the manuscript.

Revise

Endurance: In the open configuration (**no payload**), the aerial robot achieves a stable flight endurance of about 237 s, sufficient for performing representative tasks such as grasping, perching, door opening, and human–robot interaction. In the contracted configuration, airflow blockage causes an estimated power loss of about 10%, reducing the endurance to approximately 212 s. **Moreover, the endurance under full payload is approximately 117 s.**

Comment 2

The time metrics relative to the planning method (Comment 7) are quite low, in the range of ms and micro-second. For a computer board with a processor running at 1.8 GHz, I guess that the complexity of the trajectories is relatively low to achieve such low computational time, or rather the methods is very efficiently implemented.

Response: We sincerely appreciate your comment. As described in the Methods section, our trajectory planning primarily considers constraints such as dynamic feasibility, smoothness, and minimum-time optimization. These constraints are efficiently handled within a high-performance solving framework, enabling rapid computation. For further details, please refer to our previous work [1].

Comment 3

In the external load estimation (Comment 9) I think the plot of the external force is the same on pages 13 and 14 on the response letter. **I expected to see here the corrected estimation compensating the thrust loss to achieve the 1.5 N ground truth. Please, check also the manuscript.**

Response: We sincerely appreciate your valuable suggestions and have made corresponding revisions in the manuscript.

Palm grasping

Fig. R1. Hand-like grasping. (a) HI-ARM palm grasps, transports, and releases a water bottle (image mirrored horizontally for better readability). (b) State curves of HI-ARM during the water bottle grasping process, including position tracking, velocity tracking, estimated external forces, estimated external torques, motor speed, and actual total thrust. (c) Adaptive grasping of objects with various shapes by HI-ARM. (d) A sequence of HI-ARM performing fingertip grasping on a napkin. (e) Fingertip grasping of items with various shapes.

Comment 4

There is one remaining minor comment that should be clarified before final acceptance: **Immediately following Equation (4), the expression referring to "the module positions r_{mod_i} " lacks sufficient explanation.** It is not clear to the reader which figure or diagram provides the spatial definition or visualization of this symbol. A pointer to the appropriate figure (e.g., Fig. X) would greatly help readers.

Response: We sincerely appreciate your valuable suggestions and have made corresponding revisions in the manuscript.

Revise

Estimation of COG and inertia tensor

... The center of gravity \mathbf{r}_{COG} and inertia tensor \mathbf{J} are re-estimated in real time based on the updated component positions. This guarantees that the morphing-induced variations in mass distribution are consistently reflected in the dynamic model used for control. The robot is mainly composed of four brushless motors for flight, a servo motor, a main control board, two batteries, six linear slides and four main structural modules (the module refers to the finger or palm module, as shown in Fig. 3b(i), with the mass and COG of each module denoted as m_{mot_i} , \mathbf{r}_{mot_i}). To simplify the estimation of the inertia tensor, we focus our analysis on the mass-dominated components and ignore the less massive ones.

References

- [1] Zhepei Wang, Xin Zhou, Chao Xu, and Fei Gao. Geometrically constrained trajectory optimization for multicopters. *IEEE Transactions on Robotics*, 38(5):3259–3278, 2022.